# Learning Causally Invariant Representations for Out-of-Distribution Generalization on Graphs

**Yongqiang Chen**[1][*] **Yonggang Zhang**[2], **Yatao Bian**[3], **Han Yang**[1], **Kaili Ma**[1], **Binghui Xie**[1]
[1]The Chinese University of Hong Kong [2]Hong Kong Baptist University
{yqchen,hyang,klma,bhxie21,jcheng}@cse.cuhk.edu.hk yatao.bian@gmail.com

**Tongliang Liu**[4], **Bo Han**[2], **James Cheng**[1]
[3]Tencent AI Lab [4]TML Lab, The University of Sydney
tongliang.liu@sydney.edu.au {csygzhang,bhanml}@comp.hkbu.edu.hk

## Abstract

Despite recent success in using the invariance principle for out-of-distribution (OOD) generalization on Euclidean data (e.g., images), studies on graph data are still limited. Different from images, the complex nature of graphs poses unique challenges to adopting the invariance principle. In particular, distribution shifts on graphs can appear in a variety of forms such as attributes and structures, making it difficult to identify the invariance. Moreover, domain or environment partitions, which are often required by OOD methods on Euclidean data, could be highly expensive to obtain for graphs. To bridge this gap, we propose a new framework, called **C**ausality **I**nspired **I**nvariant **G**raph Le**A**rning (CIGA), to capture the invariance of graphs for guaranteed OOD generalization under various distribution shifts. Specifically, we characterize potential distribution shifts on graphs with causal models, concluding that OOD generalization on graphs is achievable when models focus *only* on subgraphs containing the most information about the causes of labels. Accordingly, we propose an information-theoretic objective to extract the desired subgraphs that maximally preserve the invariant intra-class information. Learning with these subgraphs is immune to distribution shifts. Extensive experiments on 16 synthetic or real-world datasets, including a challenging setting – DrugOOD, from AI-aided drug discovery, validate the superior OOD performance of CIGA[1].

## 1 Introduction

Graph representation learning with graph neural networks (GNNs) has gained great success in tasks involving relational information [45, 35, 99, 106, 107]. However, it assumes that the training and test graphs are drawn from the same distribution, which is often violated in reality [37, 47, 38, 40]. The mismatch between training and test distributions, i.e., *distribution shifts*, introduced by some underlying environmental factors related to data collection or processing, could seriously degrade the performance of deployed models [7, 24]. Such *out-of-distribution* (OOD) generalization failures become the major roadblock for practical applications of graph representation learning [40].

Meanwhile, enabling OOD generalization on regular Euclidean data has received surging attention and several solutions were proposed [4, 81, 10, 49, 23, 48, 2]. In particular, the invariance principle from causality is at the heart of those works [76, 74, 79]. The principle leverages the Independent Causal Mechanism (ICM) assumption [74, 77] and implies that, model predictions that only focus on the causes of the label can stay invariant to a large class of distribution shifts [76, 4, 2].

---

[*]Work done during an internship at Tencent AI Lab.
[1]Code is available at https://github.com/LFhase/CIGA.

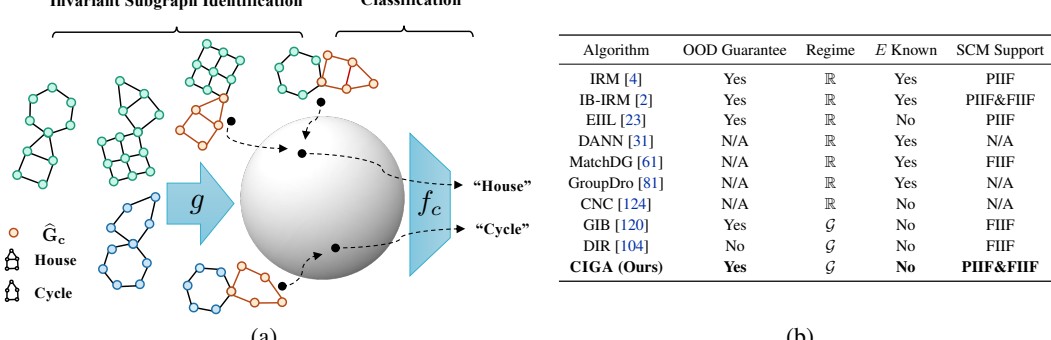

**Invariant Subgraph Identification**  **Classification**

| Algorithm | OOD Guarantee | Regime | $E$ Known | SCM Support |
|---|---|---|---|---|
| IRM [4] | Yes | $\mathbb{R}$ | Yes | PIIF |
| IB-IRM [2] | Yes | $\mathbb{R}$ | Yes | PIIF&FIIF |
| EIIL [23] | Yes | $\mathbb{R}$ | No | PIIF |
| DANN [31] | N/A | $\mathbb{R}$ | Yes | N/A |
| MatchDG [61] | N/A | $\mathbb{R}$ | Yes | FIIF |
| GroupDro [81] | N/A | $\mathbb{R}$ | Yes | N/A |
| CNC [124] | N/A | $\mathbb{R}$ | No | N/A |
| GIB [120] | Yes | $\mathcal{G}$ | No | FIIF |
| DIR [104] | No | $\mathcal{G}$ | No | FIIF |
| **CIGA (Ours)** | **Yes** | $\mathcal{G}$ | **No** | **PIIF&FIIF** |

(a)          (b)

Figure 1: (a) Illustration of **C**ausality **I**nspired In**v**ariant **G**raph Le**A**rning (CIGA): GNNs need to classify graphs based on the specific motif ("House" or "Cycle"). The featurizer $g$ will extract an (orange colored) subgraph $\widehat{G}_c$ from each input for the classifier $f_c$ to predict the label. The training objective of $g$ is implemented in a contrastive strategy where the distribution of $\widehat{G}_c$ at the latent sphere will be optimized to maximize the intra-class mutual information, hence predictions will be invariant to distribution shifts; (b) An overview of potential algorithms for OOD generalization on graphs.

Despite the success of the invariance principle on Euclidean data, the complex nature of graphs raises several new challenges that prohibit direct adoptions of the principle. First, distribution shifts on graphs are more complicated. They can happen at both attribute-level and structure-level, and be observed in multiple forms such as graph sizes, subgraph densities and homophily [113, 11, 102]. On the other hand, each of the shifts can spuriously correlate with labels in different modes [4, 71, 2]. Consequently, the entangled complex distribution shifts make it more difficult to identify and capture the invariance on graphs. Second, OOD algorithms developed and analyzed on Euclidean data often require additional environment (or domain) labels for distinguishing the sources of distribution shifts [4]. However, the environment labels could be highly expensive to obtain and thus often unavailable for graphs, as collecting the labels usually requires expert knowledge due to the abstraction of graphs [37]. These challenges render the problem studied in this paper even more challenging:

*How could one generalize the invariance principle to enable OOD generalization on graphs?*

To solve the above problem, we propose **C**ausality **I**nspired In**v**ariant **G**raph Le**A**rning (CIGA), a new framework for capturing the invariance of graphs to enable guaranteed OOD generalization under different distribution shifts. Specifically, we build three Structural Causal Models (SCMs) [74] to characterize the distribution shifts that could happen on graphs: one is to model the graph generation process, and the other two are to model two possible interactions between invariant and spurious features during the graph generation, i.e., Fully Informative Invariant Feature (FIIF) and Partially Informative Invariant Feature (PIIF) (Sec. 2.2). Then, we generalize the invariance principle to graphs for OOD generalization: GNN models are invariant to distribution shifts if they focus only on an invariant and critical subgraph $G_c$ that contains the most of the information in $G$ about the underlying causes of the label. Thus, the problem of achieving OOD generalization on graphs can be rephrased into two processes: invariant subgraph identification and label prediction. Accordingly, shown as Fig. 1(a), we introduce a prototypical invariant graph learning algorithm that decomposes a GNN into: a) a featurizer $g$ for identifying the underlying invariant subgraph $G_c$ from $G$; b) a classifier $f_c$ for making predictions based on $G_c$. To extract the desired subgraph $G_c$, we derive an information-theoretic objective for the featurizer to identify subgraphs that maximally preserves the invariant intra-class information across a set of different (unknown) environments. We theoretically show that this approach can provably identify the underlying $G_c$ under mild assumptions (Sec. 3).

Experiments on 16 synthetic and real-world datasets with various distribution shifts, including a challenging setting from AI-aided drug discovery [40], show that CIGA can significantly outperform all of existing methods up to $10\%$, demonstrating its promising OOD generalization ability (Sec. 4).

**Related Work.** We review existing methods that might improve the OOD generalization on graphs, summarize the main differences between our solution and them in Table 1(b), and leave thorough discussions to Appendix B.2. On Euclidean data, Invariant Learning [4, 23, 2], Group Distributionally Robust Optimization [49, 81, 124], Domain Adaption and Domain Generalization [31, 93, 52, 27, 61,

100] are three widely adopted approaches to enable OOD generalization. However, they all have their own limitations when being applied to graphs. First, previous invariant learning methods are mostly developed and analyzed for Euclidean data [4, 2, 23], or under specific SCM assumptions [4], making the theoretical results hardly able to generalize to the complicated graph data [80] that can have multiple types of distribution shifts [71]. Group Distributionally Robust Optimization that minimizes the gap between worst group risk and average risk [49, 81, 124], and Domain Adaption/Generalization methods that aim to learn class-conditional domain invariant representations [31, 93, 52, 27, 100], cannot guarantee a min-max optimal predictor without additional assumptions [126, 4, 2]. Moreover, most existing methods require environment labels that are however expensive to obtain in graphs, which limits their applications to graphs [4, 49, 2, 81, 31, 93, 27, 61]. In contrast, we aim to develop OOD algorithms for graphs that are provably generalizable under different types of distribution shifts.

Another line of relevant works is about GNN explainability that aims to find a subgraph of the input as the explanation for a GNN prediction [116, 122]. Although some may leverage causality to justify the generated explanation [53], they mostly focus on understanding the predictions of GNNs instead of for OOD generalization. The closest works to ours are two interpretable GNNs that aim to explicitly extract a subgraph for both predictions and explanations guided by information theory [120] and causality [104], respectively. However, they focus on graphs and shifts generated under a specific SCM. Although one of them can provide theoretical guarantee for OOD generalization [120] by using the information bottleneck criteria [2], they would inevitably fail to generalize to graphs generated under different SCMs. More discussions about the failure are deferred to Appendix D.4. Besides, Bevilacqua et al. [11] also discuss OOD generalization on graphs but limited to a specific graph family and graph size shifts. Wu et al. [103] propose OOD generalization algorithms on graphs for the task of node classification, also limited to graphs and shifts under a specific SCM.

To the best of our knowledge, there is no existing work that could handle more comprehensive graph distribution shifts than CIGA, while also achieving provable OOD generalization performance.

## 2 OOD Generalization on Graphs through the Lens of Causality

### 2.1 Problem Setup

In this work, we focus on OOD generalization in graph classification. Specifically, we are given a set of graph datasets $\mathcal{D} = \{\mathcal{D}^e\}_e$ collected from multiple environments $\mathcal{E}_{\text{all}}$. Samples $(G_i^e, Y_i^e) \in \mathcal{D}^e$ from the same environment are considered as drawn independently from an identical distribution $\mathbb{P}^e$. A GNN $\rho \circ h$ generically has an encoder $h : \mathcal{G} \to \mathbb{R}^h$ that learns a meaningful representation $h_G$ for each graph $G$ to help predict the label $\hat{Y}_G = \rho(h_G)$ with a downstream classifier $\rho : \mathbb{R}^h \to \mathcal{Y}$. The goal of OOD generalization on graphs is to train a GNN $\rho \circ h$ with data from training environments $\mathcal{D}_{\text{tr}} = \{\mathcal{D}^e\}_{e \in \mathcal{E}_{\text{tr}} \subseteq \mathcal{E}_{\text{all}}}$ that generalizes well to all (unseen) environments, i.e., to minimize $\max_{e \in \mathcal{E}_{\text{all}}} R^e$, where $R^e$ is the empirical risk of $\rho \circ h$ under environment $e$ [97, 4]. We leave more details about the background of GNN for graph classification and invariant learning in Appendix B.1.

It is known that OOD generalization is impossible without assumptions on the environments $\mathcal{E}_{\text{all}}$ [74, 2]. Thus, we will first formulate the data generation process with structural causal model and latent-variable model [74, 77, 50], to characterize the distribution shifts that could happen on graphs. Then, we investigate whether the existing methods are generalizable under these distribution shifts.

### 2.2 Graph Generation Process

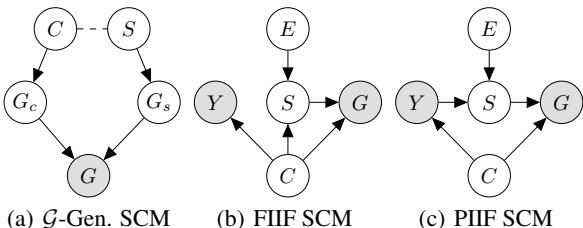

(a) $\mathcal{G}$-Gen. SCM    (b) FIIF SCM    (c) PIIF SCM

Figure 2: SCMs on graph distribution shifts.

We take a latent-variable model perspective on the graph generation process and assume that the graph is generated through a mapping $f_{\text{gen}} : \mathcal{Z} \to \mathcal{G}$, where $\mathcal{Z} \subseteq \mathbb{R}^n$ is the latent space and $\mathcal{G} = \cup_{N=1}^{\infty} \{0, 1\}^N \times \mathbb{R}^{N \times d}$ is the graph space. Let $E$ denote environments. Following previous works [50, 2], we partition the latent variable from $\mathcal{Z}$ into an invariant part $C \in \mathcal{C} = \mathbb{R}^{n_c}$ and a varying part

$S \in \mathcal{S} = \mathbb{R}^{n_s}$, s.t., $n = n_c + n_s$, according to whether they are affected by $E$ or not. Similarly in images, $C$ and $S$ can represent content and style while $E$ can refer to the locations where the images are taken [7, 125, 50]. Furthermore, $C$ and $S$ control the generation of the observed graphs (Assumption 2.1) and can have multiple types of interactions at the latent space (Assumptions 2.2, 2.3).

**Graph generation model.** We elaborate the SCM for the graph generation process in Assumption 2.1 and Fig. 2(a), where noises in the structural equations are omitted for simplicity [77].

**Assumption 2.1** (Graph Generation Structural Causal Model).

$$G_c := f_{\text{gen}}^{G_c}(C), \qquad G_s := f_{\text{gen}}^{G_s}(S), \qquad G := f_{\text{gen}}^{G}(G_c, G_s).$$

In Assumption 2.1, $f_{\text{gen}}$ is decomposed into $f_{\text{gen}}^{G_c}$, $f_{\text{gen}}^{G_s}$ and $f_{\text{gen}}^{G}$ to control the generation of $G_c$, $G_s$, and $G$, respectively. Among them, $G_c$ inherits the invariant information of $C$ that would not be affected by the interventions (or changes) of $E$ [74, 77]. For example, certain properties of a molecule can usually be described by a sub-molecule, or a functional group, which is invariant across different species or assays [12, 92, 40]. On the contrary, the generation of $G_s$ and $G$ will be affected by the environment $E$ through $S$. Thus, graphs collected from different environments (or domains) can have different distributions of structure-level properties (e.g., graph sizes [11, 102]) as well as feature-level properties (e.g., homophily [62, 17]). Therefore, the subgraph $G_s$ inherits the spurious feature about $Y$ [125]. In fact, Assumption 2.1 is compatible with many graph generation models by specifying the function classes of $f_{\text{gen}}^{G_c}$, $f_{\text{gen}}^{G_s}$ and $f_{\text{gen}}^{G}$ [89, 57, 117, 59]. Since our goal is to characterize the potential distribution shifts in Assumption 2.1, we focus on building a general SCM that is compatible to many graph families and leave graph family specifications and their implications to OOD generalization in future works. More discussions are provided in Appendix C.

**Interactions at latent space.** Following previous works [4, 2], we categorize the latent interactions between $C$ and $S$ into Fully Informative Invariant Features (FIIF, Fig. 2(b)) and Partially Informative Invariant Features (PIIF, Fig. 2(c))[2], depending on whether the latent invariant part $C$ is fully informative about label $Y$, i.e., $(S, E) \perp\!\!\!\perp Y | C$. Formal definitions of the corresponding SCMs are given as follows, where noises are omitted for simplicity [74, 77].

**Assumption 2.2** (FIIF Structural Causal Model). $Y := f_{\text{inv}}(C),\ S := f_{\text{spu}}(C, E),\ G := f_{\text{gen}}(C, S)$.

**Assumption 2.3** (PIIF Structural Causal Model). $Y := f_{\text{inv}}(C),\ S := f_{\text{spu}}(Y, E),\ G := f_{\text{gen}}(C, S)$.

In the two SCMs above, $f_{\text{gen}}$ corresponds to the graph generation process in Assumption 2.1, and $f_{\text{spu}}$ is the mechanism describing how $S$ is affected by $C$ and $E$ at the latent space. By definition, $S$ is directly controlled by $C$ in FIIF and indirectly controlled by $C$ through $Y$ in PIIF, which can exhibit different behaviors in the observed distribution shifts. In practice, performances of OOD algorithms can degrade dramatically if one of FIIF or PIIF is excluded [5, 71]. This issue can be more serious in graphs, since different distribution shifts can have different interaction modes at the latent space. Moreover, $f_{\text{inv}} : \mathcal{C} \to \mathcal{Y}$ indicates the labelling process, which assigns labels $Y$ for the corresponding $G$ merely based on $C$. Consequently, $\mathcal{C}$ is better clustered than $\mathcal{S}$ when given $Y$ [13, 15, 86, 87], which also serves as the necessary separation assumption for a classification task [69, 16, 65].

**Assumption 2.4** (Better Clustered Invariant Features). $H(C|Y) \leq H(S|Y)$.

### 2.3  Challenges of OOD Generalization on Graphs

Built upon the graph generation process, we can formally derive the desired GNN that is able to generalize to OOD graphs under different distribution shifts, which implies the invariant GNN below[3].

**Definition 2.5** (Invariant GNN). Given a set of graph datasets $\{\mathcal{D}^e\}_e$ and environments $\mathcal{E}_{\text{all}}$ that follow the same graph generation process in Sec. 2.2, considering a GNN $\rho \circ h$ that has a permutation invariant graph encoder $h : \mathcal{G} \to \mathbb{R}^h$ and a downstream classifier $\rho : \mathbb{R}^h \to \mathcal{Y}$, $\rho \circ h$ is an invariant GNN if it minimizes the worst case risk among all environments, i.e., $\min \max_{e \in \mathcal{E}_{\text{all}}} R^e$.

Can existing methods produce a desired invariant GNN model? We find the answers to be negative unfortunately. Based on the synthetic BAMotif graph classification task [58, 104] shown in Fig. 3,

---

[2]Note that FIIF and PIIF can be mixed as Mixed Informative Invariant Features (Appendix 6(d)) in several ways, while our analysis will focus on the axiom ones for the purpose of generality.

[3]A discussion on Def. 2.5 and its relation to the SCMs is provided in Appendix E.1.

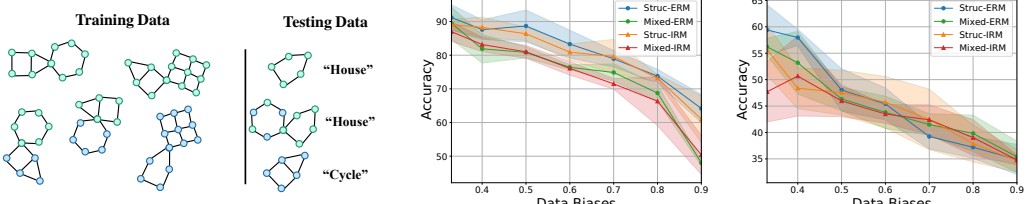

(a) Failure cases for existing methods. (b) Structure and attribute shifts. (c) Mixed with graph size shifts.

Figure 3: Failures of OOD generalization on graphs: (a) GNNs are required to classify whether the graph contains a "house" or "cycle" motif, where the colors represent node features. However, distribution shifts in the training data exist at both structure-level (from left to right: "house" mostly co-occur with a hexagon), attribute-level (from upper to lower: nodes are mostly colored green if the graph contains a "house", or colored blue if the graph contains a "cycle"), and graph sizes, making GNNs hard to capture the invariance. Consequently, *ERM can fail* for leveraging the shortcuts and predicting graphs that have a hexagon or have nodes mostly colored green as "house". *IRM can fail* as the test data are not sufficiently supported by the training data. (b) GCNs optimized with neither ERM nor IRM can generalize to OOD graphs under structure-level shifts (Struc-) or mixed with feature shifts (Mixed-). (c) When more complex shifts presented, GNNs can fail more seriously.

we theoretically and empirically analyze whether existing methods could produce an invariant GNN, through the investigation of the following aspects. More details and results are given in Appendix D.

**Can GNNs trained with ERM generalize to OOD graphs?** As shown in Fig. 3, we find that GNNs trained with the standard empirical risk minimization (ERM) algorithm [97] are not able to generalize to OOD graphs. As the data biases grows stronger, the performances of GNNs drop dramatically. Furthermore, when graph size shifts are mixed in the data, GNNs can have larger variance at low data biases, indicating the instability of learning the desired relationships for the task. The reason is that ERM tends to overfit to the shortcuts or spurious correlations presented in specific substructures or attributes in the graphs [33]. This phenomenon has also been shown to exist in GNNs equipped with more sophisticated architectures such as attention mechanisms [99], under graph size shifts [46].

**Can OOD objectives improve OOD generalization of GNNs?** Meanwhile, as shown in Fig. 3, OOD objectives primarily developed on Euclidean data such as invariant risk minimization (IRM) [4] also cannot alleviate the problem. On the contrary, IRM can fail catastrophically at non-linear regime if without sufficient support overlap for the test environments, i.e., $\cup_{e \in \mathcal{E}_{te}}\text{supp}(\mathbb{P}^e) \nsubseteq \cup_{e \in \mathcal{E}_{tr}}\text{supp}(\mathbb{P}^e)$ [80]. In addition to IRM, the failure would also happen for alternative objectives [49, 9, 2] as proved by Rosenfeld et al. [80]. Besides, different distribution shifts on graphs can be nested with each other where each one can have distinct spurious correlation type, e.g., FIIF or PIIF. OOD objectives will also fail seriously if either of the correlation types is not supported [5, 71]. Moreover, non-trivial environment partitions or labels are required for performance guarantee of these OOD objectives [4, 49, 81, 2]. However, collecting meaningful environment partitions of graphs requires expert knowledge about graph data. Thus, the environment labels can be expensive to obtain and are usually not available [67, 28, 37]. Alternative options such as random partitions tend not to alleviate the issue [23, 55], as it can be trivially deemed as mini-batching.

**Challenges of OOD generalization on graphs.** The aforementioned failure analysis reveals that existing methods or objectives fail to elicit an invariant GNN primarily due to the following two challenges: a) Distribution shifts on graphs are more complicated where different types of spurious correlations can be entangled via different graph properties; b) Environment labels are usually not available due to the abstraction of graphs. Despite these challenges, we are still highly motivated to address the following research question: *Would it be possible to learn an invariant GNN that is generalizable under various distribution shifts by lifting the invariance principle to the graph data?*

## 3 Invariance Principle for OOD Generalization on Graphs

We provide affirmative answers to the previous question by proposing a new framework, CIGA: **C**ausality **I**nspired Invariant **G**raph LeArning. Specifically, built upon the SCMs in Sec. 2.2, we generalize the invariance principle to graphs and instantiate the principle with theoretical guarantees.

## 3.1 Invariance for OOD Generalization on Graphs

Towards extending the invariance principle to graphs under SCMs in Sec. 2.2, we need to identify a set of variables that have stable causal relationship with $Y$ under both FIIF and PIIF (Assumption 2.2, 2.3). According to the ICM assumption [77], the labeling process $C \rightarrow Y$ is not informed nor influenced by other processes, implying that the conditional distribution $P(Y|C)$ remains invariant to the interventions on the environment latent variable $E$ [74]. Consequently, for a GNN with a permutation invariant encoder $h : \mathcal{G} \rightarrow \mathbb{R}^h$ and a downstream classifier $\rho : \mathbb{R}^h \rightarrow \mathcal{Y}$, if $h$ can recover the information of $C$ from $G$ in the learned graph representations, then the learning of $\rho$ resembles traditional ERM [97] and can achieve the desired min-max optimality required by an invariant GNN (Def. 2.5). However, recovering $C$ from $G$ is particularly difficult, since the generation of $G$ from $C$ involves two causal mechanisms $f_{\text{gen}}^{G_c}$ and $f_{\text{gen}}^{G}$ in Assumption 2.1. The unavailability of $E$ further adds up the difficulty of enforcing the independence between the learned representations and $E$.

## 3.2 Invariant Graph Learning Framework

**Causal algorithmic alignment.** To enable a GNN to learn to extract the information about $C$ from $G$, we propose the CIGA framework that *explicitly aligns with* the two causal mechanisms $f_{\text{gen}}^{G_c}$ and $f_{\text{gen}}^{G}$ in Assumption 2.1. The idea of alignment in CIGA is motivated by the algorithmic reasoning results that a neural network can learn a reasoning process better if its computation structure aligns with the process better [108, 110]. Specifically, we realize the alignment by decomposing a GNN into two sub-components[4]: a) a featurizer GNN $g : \mathcal{G} \rightarrow \mathcal{G}_c$ aiming to identify the desired $G_c$; b) a classifier GNN $f_c : \mathcal{G}_c \rightarrow \mathcal{Y}$ that predicts the label $Y$ based on the estimated $\widehat{G}_c$, where $\mathcal{G}_c$ refers to the space of subgraphs of $G$. Formally, the learning objectives of $f_c$ and $g$ can be formulated as:

$$\max_{f_c,\, g} I(\widehat{G}_c; Y), \text{ s.t. } \widehat{G}_c \perp\!\!\!\perp E, \ \widehat{G}_c = g(G), \tag{1}$$

where maximizing $I(\widehat{G}_c; Y)$ is equivalent to minimizing a variational upper bound of $R(f_c(\widehat{G}_c))$ [3, 120] that takes $\widehat{G}_c$ as inputs to predict label $Y$ for $G$ through $f_c$ and $g$, and $\widehat{G}_c$ is the estimated subgraph containing the information about $C$ and hence needs to be independent of $E$. Moreover, the extracted $G_c$ can either shares the same graph space with input $G$ or has its own space with latent node and edge features, depending on the specific graph generation process. In practice, architectures from the literature of interpretable GNNs are compatible with CIGA [122], hence can serve as practical choices for the implementation of CIGA. More details are given in Appendix F.

Although we can technically align with the two causal mechanisms with $g$ and $f_c$, trivially optimizing this architecture cannot satisfy $\widehat{G}_c \perp\!\!\!\perp E$. Formally, merely maximizing $I(\widehat{G}_c; Y)$ may include a subgraph from $G_s$ in $\widehat{G}_c$ since $G_s$ also shares certain mutual information with $Y$. Moreover, the unavailability of $E$ prevents the direct usage of $E$ in enforcing the independence that is often adopted by previous methods [4, 49, 81, 31, 93], making the identification of $G_c$ more challenging.

**Optimization objective.** To mitigate this issue, we need to find and translate other properties of $G_c$ into some differentiable and equivalent objectives to satisfy the independence constraint $\widehat{G}_c \perp\!\!\!\perp E$. *The goal of the desired objective.* We begin by considering a simplistic setting where all the invariant subgraphs $G_c$ have the same size $s_c$, i.e., $|G_c| = s_c$[5]. When maximizing $I(\widehat{G}_c; Y)$ in Eq. 1, both FIIF and PIIF can introduce part of $G_s$ into $\widehat{G}_c$. In FIIF (Fig. 2(b)), as $G_c$ already contains the maximal possible information in $G$ about $Y$, $G_c$ is a solution to $\max I(\widehat{G}_c; Y)$. However, some subgraph of $G_c$ can be replaced by some subgraph of $G_s$ that is equally informative about $Y$. In PIIF (Fig. 2(c)), there also exists some subgraph of $G_s$ that contains additional information about $Y$ than $G_c$, hence $\widehat{G}_c$ is more likely to involve some subgraph of $G_s$. Thus, the new objective needs to eliminate the auxiliary subgraphs of $\widehat{G}_c$ from $G_s$ such that the estimated $\widehat{G}_c$ can only contain $G_c$.

*An important property of $G_c$.* Under both FIIF and PIIF SCMs (Fig. 2), for $G_c^{e_1}, G_c^{e_2}$ that relate to the same causal factor $c$ under two environments $e_1$ and $e_2$, the desired $\widehat{G}_c^{e_1}, \widehat{G}_c^{e_2}$ in $e_1$ and $e_2$ tend to have high mutual information, i.e., $(G_c^{e_1}, G_c^{e_2}) \in \arg\max I(\widehat{G}_c^{e_1}; \widehat{G}_c^{e_2})$. While for $G_c^{e_1}$

---

[4]The encoder of the GNN in CIGA can be regarded as the composition of $g$ and the graph encoder in $f_c$.

[5]Throughout the paper, we use generalized set operators for the ease of understanding. They can have multiple implementations in terms of nodes, edges or attributes.

and another $G_{c'}^{e_1}$ corresponding to a different $c' \neq c$, under the same environment $e_1$, including any subgraph from $G_s^{e_1}$ in $\widehat{G}_c^{e_1}, \widehat{G}_{c'}^{e_1}$ will enlarge their mutual information, or in other words, $(G_c^{e_1}, G_{c'}^{e_1}) \in \arg\min I(\widehat{G}_c^{e_1}; \widehat{G}_{c'}^{e_1})$. Thus, we can derive an important property of $G_c$, that is, $\forall e_1, e_2 \in \mathcal{E}_{\text{all}}$,

$$G_c^{e_1} \in \arg\max_{\widehat{G}_c^{e_1}} I(\widehat{G}_c^{e_1}; \widehat{G}_c^{e_2} | C = c) - I(\widehat{G}_c^{e_1}; \widehat{G}_{c'}^{e_2} | C = c', c' \neq c), \tag{2}$$

where $\widehat{G}_c^{e_1}$ and $\widehat{G}_c^{e_2}$ are the estimated invariant subgraphs corresponding to the same causal factor $c$ under environment $e_1$ and $e_2$, respectively, while $\widehat{G}_{c'}^{e_2}$ corresponds to a different causal factor $c'$.

*Deriving CIGAv1 based on the identified property of $G_c$.* In practice, $C$ is not given. Nevertheless, since $C$ and $Y$ shares a stable causal relationship in both FIIF and PIIF SCMs, $Y$ can serve as a proxy of $C$ in Eq. 2. Moreover, as Eq. 2 holds for any $\forall e_1, e_2 \in \mathcal{E}_{\text{all}}$, the environment superscripts can be eliminated without affecting Eq. 2. Furthermore, when both $I(\widehat{G}_c^{e_1}; \widehat{G}_c^{e_2} | C = c)$ and $I(\widehat{G}_c; Y)$ are maximized, $I(\widehat{G}_c^{e_1}; \widehat{G}_{c'}^{e_1} | C = c', c' \neq c)$ is automatically minimized, otherwise all classes will collapse to trivial solutions which is contradictory given $I(\widehat{G}_c; Y)$ being maximized. Therefore, we can derive an alternative objective to Eq. 1 by leveraging Eq. 2 to replace the independence condition:

$$\text{(CIGAv1)} \qquad \max_{f_c, g} I(\widehat{G}_c; Y), \text{ s.t. } \widehat{G}_c \in \arg\max_{\widehat{G}_c = g(G), |\widehat{G}_c| \leq s_c} I(\widehat{G}_c; \widetilde{G}_c | Y), \tag{3}$$

where $\widetilde{G}_c = g(\widetilde{G})$ and $\widetilde{G} \sim \mathbb{P}(G|Y)$, i.e., $\widetilde{G}$ is sampled from training graphs that share the same label $Y$ as $G$. In Theorem 3.1, we show how Eq. 3 is equivalent to Eq. 1. Nevertheless, Eq. 3 requires a strong assumption on the size of $G_c$. However, the size of $G_c$ is usually unknown or changes for different $C$s. In this circumstance, maximizing Eq. 2 without additional constraints will lead to the presence of part of $G_s$ in $\widehat{G}_c$. For instance, $\widehat{G}_c = G$ is a trivial solution to Eq. 3 when $s_c = \infty$.

*Deriving CIGAv2 by resolving size constraint on $G_c$ in CIGAv1.* To this end, we further resort to the properties of $G_s$. In both FIIF and PIIF SCMs (Fig. 2), $G_s$ and $G_c$ can share certain overlapped information about $Y$. When maximizing $I(\widehat{G}_c; \widetilde{G}_c | Y)$ and $I(\widehat{G}_c; Y)$, the appearance of partial $G_s$ in $\widehat{G}_c$ will not affect the optimality. However, it can reduce the mutual information between the left part $\widehat{G}_s = G - \widehat{G}_c$ and $Y$, i.e., $I(\widehat{G}_s; Y)$. Therefore, by maximizing $I(\widehat{G}_s; Y)$, we can reduce including part of $G_s$ into $\widehat{G}_c$. Meanwhile, to avoid trivial solution that $G_c \subseteq \widehat{G}_s$ during maximizing $I(\widehat{G}_s; Y)$, we can leverage the better clustering property of $G_c$ implied by Assumption 2.4 to derive the constraint $I(\widehat{G}_s; Y) \leq I(\widehat{G}_c; Y)$. Thus, we can obtain a new objective CIGAv2 as follows:

$$\max_{f_c, g} I(\widehat{G}_c; Y) + I(\widehat{G}_s; Y), \text{ s.t. } \widehat{G}_c \in \arg\max_{\widehat{G}_c = g(G)} I(\widehat{G}_c; \widetilde{G}_c | Y),$$

$$\text{(CIGAv2)} \qquad\qquad I(\widehat{G}_s; Y) \leq I(\widehat{G}_c; Y), \; \widehat{G}_s = G - g(G), \tag{4}$$

where $\widehat{G}_c = g(G), \widetilde{G}_c = g(\widetilde{G})$ and $\widetilde{G} \sim \mathbb{P}(G|Y)$, i.e., $\widetilde{G}$ is sampled from training graphs that share the same label $Y$ as $G$. We also prove the equivalence between Eq. 4 and Eq. 1 in Theorem 3.1.

## 3.3 Theoretical Analysis and Practical Discussions

**Theorem 3.1** (CIGA Induces Invariant GNNs). *Given a set of graph datasets $\{\mathcal{D}^e\}_e$ and environments $\mathcal{E}_{\text{all}}$ that follow the same graph generation process in Sec. 2.2, assuming that (a) $f_{gen}^G$ and $f_{gen}^{G_c}$ in Assumption 2.1 are invertible, (b) samples from each training environment are equally distributed, i.e., $|\mathcal{D}_{\hat{e}}| = |\mathcal{D}_{\tilde{e}}|, \; \forall \hat{e}, \tilde{e} \in \mathcal{E}_{tr}$, then:*

*(i). If $\forall G_c, |G_c| = s_c$, then each solution to Eq. 3, elicits an invariant GNN (Def. 2.5).*

*(ii). Each solution to Eq. 4, elicits an invariant GNN (Def. 2.5).*

We prove Theorem 3.1 (i) and (ii) in Appendix E.2, E.3, respectively.

**Practical implementations of CIGA objectives.** After showing the power of CIGA, we introduce the practical implementations of CIGAv1 and CIGAv2 objectives. Specifically, an exact estimate of the second term $I(\widehat{G}_c; \widetilde{G}_c | Y)$ could be highly expensive [96, 8]. However, contrastive learning with supervised sampling provides a practical solution for the approximation [42, 20, 82, 96, 8]:

$$I(\widehat{G}_c; \widetilde{G}_c | Y) \approx \mathbb{E}_{\substack{\{\widehat{G}_c, \widetilde{G}_c\} \sim \mathbb{P}_g(G|\mathcal{Y}=Y) \\ \{G_c^i\}_{i=1}^M \sim \mathbb{P}_g(G|\mathcal{Y}\neq Y)}} \log \frac{e^{\phi(h_{\widehat{G}_c}, h_{\widetilde{G}_c})}}{e^{\phi(h_{\widehat{G}_c}, h_{\widetilde{G}_c})} + \sum_{i=1}^M e^{\phi(h_{\widehat{G}_c}, h_{G_c^i})}}, \tag{5}$$

Table 1: OOD generalization performance on structure and mixed shifts for synthetic graphs.

| | SPMotif-Struc[†] | | | SPMotif-Mixed[†] | | | |
| | BIAS=0.33 | BIAS=0.60 | BIAS=0.90 | BIAS=0.33 | BIAS=0.60 | BIAS=0.90 | AVG |
|---|---|---|---|---|---|---|---|
| ERM | 59.49 (3.50) | 55.48 (4.84) | 49.64 (4.63) | 58.18 (4.30) | 49.29 (8.17) | 41.36 (3.29) | 52.24 |
| ASAP | 64.87 (13.8) | 64.85 (10.6) | 57.29 (14.5) | 66.88 (15.0) | 59.78 (6.78) | 50.45 (4.90) | 60.69 |
| DIR | 58.73 (11.9) | 48.72 (14.8) | 41.90 (9.39) | 67.28 (4.06) | 51.66 (14.1) | 38.58 (5.88) | 51.14 |
| IRM | 57.15 (3.98) | 61.74 (1.32) | 45.68 (4.88) | 58.20 (1.97) | 49.29 (3.67) | 40.73 (1.93) | 52.13 |
| V-REX | 54.64 (3.05) | 53.60 (3.74) | 48.86 (9.69) | 57.82 (5.93) | 48.25 (2.79) | 43.27 (1.32) | 51.07 |
| EIIL | 56.48 (2.56) | 60.07 (4.47) | 55.79 (6.54) | 53.91 (3.15) | 48.41 (5.53) | 41.75 (4.97) | 52.73 |
| IB-IRM | 58.30 (6.37) | 54.37 (7.35) | 45.14 (4.07) | 57.70 (2.11) | 50.83 (1.51) | 40.27 (3.68) | 51.10 |
| CNC | 70.44 (2.55) | 66.79 (9.42) | 50.25 (10.7) | 65.75 (4.35) | 59.27 (5.29) | 41.58 (1.90) | 59.01 |
| **CIGAv1** | **71.07 (3.60)** | 63.23 (9.61) | 51.78 (7.29) | **74.35 (1.85)** | **64.54 (8.19)** | 49.01 (9.92) | **62.33** |
| **CIGAv2** | **77.33 (9.13)** | **69.29 (3.06)** | **63.41 (7.38)** | 72.42 (4.80) | **70.83 (7.54)** | **54.25 (5.38)** | **67.92** |
| ORACLE (IID) | | 88.70 (0.17) | | | 88.73 (0.25) | | |

[†]Higher accuracy and lower variance indicate better OOD generalization ability.

where positive samples $(\widehat{G}_c, \widetilde{G}_c)$ are the extracted subgraphs of graphs that share the same label as $G$, negative samples are those having different labels, $\mathbb{P}_g(G|\mathcal{Y} = Y)$ is the push-forward distribution of $\mathbb{P}(G|\mathcal{Y} = Y)$ by featurizer $g$, $\mathbb{P}(G|\mathcal{Y} = Y)$ refers to the distribution of $G$ given the label $Y$, $\mathbb{P}(G|\mathcal{Y} \neq Y)$ refers to the distribution of $G$ given the label that is different from $Y$, $h_{\widehat{G}_c}, h_{\widetilde{G}_c}, h_{G_c^i}$ are the graph presentations of the estimated subgraphs, and $\phi$ is the similarity metric for graph representations. As $M \to \infty$, Eq. 5 approximates $I(\widehat{G}_c; \widetilde{G}_c|Y)$, which can be regarded as a non-parameteric resubstitution entropy estimator via the von Mises-Fisher kernel density [1, 41, 101]. Thus, plugging it into Eq. 3 and Eq. 4 can relieve the issue of approximating $I(\widehat{G}_c; \widetilde{G}_c|Y)$ in practice.

To implement $I(\widehat{G}_s; Y)$ given the constraint $I(\widehat{G}_s; Y) \leq I(\widehat{G}_c; Y)$ in CIGAv2, a practical choice is to adopt hinge loss that implement the constrained $I(\widehat{G}_s; Y)$ as $\frac{1}{N} R_{\widehat{G}_s} \cdot \mathbb{I}(R_{\widehat{G}_c} \leq R_{\widehat{G}_s})$, where $N$ is the number of samples, $\mathbb{I}$ is an indicator function that outputs 1 when the inner condition is satisfied otherwise 0, and $R_{\widehat{G}_s}$ and $R_{\widehat{G}_c}$ are the empirical risk vector of the predictions for each sample based on the corresponding $\widehat{G}_s$ and $\widehat{G}_c$. More implementation details can be found in Appendix F.

**Discussions and implications of CIGA.** Although using contrastive learning to improve OOD generalization is not new in the literature [27, 61, 124], previous methods cannot yield OOD guarantees in graph circumstances due to the highly non-linearity and the unavailability of domain labels $E$. In particular, CIGA can *be reduced to directly applying contrastive learning* when without the decomposition for causal algorithmic alignment. However, in the experiments we found that merely using the contrastive objective, i.e., CNC [124], yields unsatisfactory OOD generalization performance, which further implies the necessity of the decomposition in CIGA.

Moreover, the architecture of CIGA can have multiple other implementations for both the featurizer and classifier, such as identifying $G_c$ at the latent space [86, 87]. Since we cannot enumerate every possible implementation, in this work we choose interpretable GNN architectures as a prototype validation for CIGA and leave more sophisticated architectures as future works. In particular, when optimized with ERM objective, CIGA can *be reduced to interpretable GNNs*. However, merely using interpretable GNNs such as ASAP [78], GIB [120] or DIR [104] cannot yield satisfactory OOD performance. As shown in Table 1(b) and discussed in Appendix. D.4, GIB can only work for FIIF, while DIR *cannot* yield OOD guarantees for neither FIIF and PIIF SCMs. These results are also empirically validated in the experiments. We provide more detailed discussions in Appendix B.

## 4 Empirical Studies

We conduct extensive experiments with 16 datasets to verify the effectiveness of CIGA.

**Datasets.** We use the SPMotif datasets from DIR [104] where artificial structural shifts and graph size shifts are nested (SPMotif-Struc). Besides, we construct a harder version mixed with attribute shifts (SPMotif-Mixed). To examine CIGA in real-world scenarios with more complicated relationships and distribution shifts, we also use DrugOOD [40] from AI-aided Drug Discovery with Assay, Scaffold, and Size splits, convert the ColoredMNIST from IRM [4] using the algorithm from Knyazev et al. [46] to inject attribute shifts, and split Graph-SST [122] to inject degree biases. To compare with previous specialized OOD methods for graph size shifts [113, 11], we use the datasets in Bevilacqua et al. [11] that are converted from TU benchmarks [67]. More details can be found in Appendix G.1.

**Baselines and our methods.** Besides the ERM, we also compare with SOTA interpretable GNNs, GIB [120], ASAP Pooling [78], and DIR [104], to validate the effectiveness of the optimization

Table 2: OOD generalization performance on complex distribution shifts for real-world graphs.

| DATASETS | DRUG-ASSAY | DRUG-SCA | DRUG-SIZE | CMNIST-SP | GRAPH-SST5 | TWITTER | AVG (RANK)[†] |
|---|---|---|---|---|---|---|---|
| ERM | 71.79 (0.27) | 68.85 (0.62) | 66.70 (1.08) | 13.96 (5.48) | 43.89 (1.73) | 60.81 (2.05) | 54.33 (6.00) |
| ASAP | 70.51 (1.93) | 66.19 (0.94) | 64.12 (0.67) | 10.23 (0.51) | 44.16 (1.36) | 60.68 (2.10) | 52.65 (8.33) |
| GIB | 63.01 (1.16) | 62.01 (1.41) | 55.50 (1.42) | 15.40 (3.91) | 38.64 (4.52) | 48.08 (2.27) | 47.11 (10.0) |
| DIR | 68.25 (1.40) | 63.91 (1.36) | 60.40 (1.42) | 15.50 (8.65) | 41.12 (1.96) | 59.85 (2.98) | 51.51 (9.33) |
| IRM | 72.12 (0.49) | 68.69 (0.65) | 66.54 (0.42) | 31.58 (9.52) | 43.69 (1.26) | 63.50 (1.23) | 57.69 (4.50) |
| V-REX | 72.05 (1.25) | 68.92 (0.98) | 66.33 (0.74) | 10.29 (0.46) | 43.28 (0.52) | 63.21 (1.57) | 54.01 (6.17) |
| EIIL | 72.60 (0.47) | 68.45 (0.53) | 66.38 (0.66) | 30.04 (10.9) | 42.98 (1.03) | 62.76 (1.72) | 57.20 (5.33) |
| IB-IRM | 72.50 (0.49) | 68.50 (0.40) | 66.64 (0.28) | **39.86 (10.5)** | 40.85 (2.08) | 61.26 (1.20) | 58.27 (5.33) |
| CNC | 72.40 (0.46) | 67.24 (0.90) | 65.79 (0.80) | 12.21 (3.85) | 42.78 (1.53) | 61.03 (2.49) | 53.56 (7.50) |
| **CIGAv1** | **72.71 (0.52)** | **69.04 (0.86)** | **67.24 (0.88)** | 19.77 (17.1) | **44.71 (1.14)** | **63.66 (0.84)** | 56.19 (2.50) |
| **CIGAv2** | **73.17 (0.39)** | **69.70 (0.27)** | **67.78 (0.76)** | 44.91 (4.31) | 45.25 (1.27) | 64.45 (1.99) | 60.88 (1.00) |
| ORACLE (IID) | 85.56 (1.44) | 84.71 (1.60) | 85.83 (1.31) | 62.13 (0.43) | 48.18 (1.00) | 64.21 (1.77) | |

[†] Averaged rank is also reported in the blankets because of dataset heterogeneity. Lower rank is better.

objective in CIGA. We use the same selection ratio (i.e., $s_c$) for all models. Moreover, to validate the effectiveness of the decomposition in CIGA, we compare CIGA with SOTA OOD objectives including IRM [4], v-Rex [49] and IB-IRM [2], for which we apply random environment partitions following [23]. We also compare CIGA with EIIL [23] and CNC [124] that do not require environment labels, where CNC [124] has a more sophisticated contrastive sampling strategy for combating subpopulation shifts. More implementation and comparison details are deferred to Appendix G.2.

**Evaluation.** We report the classification accuracy for all datasets, except for DrugOOD datasets where we use ROC-AUC following [40], and for TU datasets where we use Matthews correlation coefficient following [11]. We repeat the evaluation multiple times, select models based on the validation performances, and report the mean and standard deviation of the corresponding metric. For each dataset, we also report the "Oracle" performances that run ERM on the randomly shuffled data.

**OOD generalization performance on structure and mixed shifts.** In Table 1, we report the test accuracy of each method, where we omit GIB due to its poor convergence. Different biases indicate different strengths of the distribution shifts. Although the training accuracy of most methods converges to more than 99%, the test accuracy decreases dramatically as the bias increases and as more distribution shifts are mixed, which concurs with our discussions in Sec. 2.3 and Appendix D. Due to the simplicity of the task as well as the relatively high support overlap between training and test distributions, interpretable GNNs and OOD objectives can improve certain OOD performance, while they can have *high variance* since they donot have OOD generalization guarantees. In contrast, CIGAv1 and CIGAv2 outperform all of the baselines by a significant margin up to 10% with *lower variance*, which demonstrates the effectiveness and excellent OOD generalization ability of CIGA.

**OOD generalization performance on realistic shifts.** In Table 2 and Table 3, we examine the effectiveness of CIGA in real-world data and more complicated distribution shifts. Both averaged accuracy and ranks are reported because of the dataset heterogeneity. Since the tasks are harder than synthetic ones, interpretable GNNs and OOD objectives perform similar to or even under-perform the ERM baselines, which is also consistent to the observations in non-linear benchmarks [34, 40]. However, both CIGAv1 and CIGAv2 con-

Table 3: OOD generalization performance on graph size shifts for real-world graphs in terms of Matthews correlation coefficient.

| DATASETS | NCI1 | NCI109 | PROTEINS | DD | AVG |
|---|---|---|---|---|---|
| ERM | 0.15 (0.05) | 0.16 (0.02) | 0.22 (0.09) | 0.27 (0.09) | 0.20 |
| ASAP | 0.16 (0.10) | 0.15 (0.07) | 0.22 (0.16) | 0.21 (0.08) | 0.19 |
| GIB | 0.13 (0.10) | 0.16 (0.02) | 0.19 (0.08) | 0.01 (0.18) | 0.12 |
| DIR | 0.21 (0.06) | 0.13 (0.05) | 0.25 (0.14) | 0.20 (0.10) | 0.20 |
| IRM | 0.17 (0.02) | 0.14 (0.01) | 0.21 (0.09) | 0.22 (0.08) | 0.19 |
| V-REX | 0.15 (0.04) | 0.15 (0.04) | 0.22 (0.06) | 0.21 (0.07) | 0.18 |
| EIIL | 0.14 (0.03) | 0.16 (0.02) | 0.20 (0.05) | 0.23 (0.10) | 0.19 |
| IB-IRM | 0.12 (0.04) | 0.15 (0.06) | 0.21 (0.06) | 0.15 (0.13) | 0.16 |
| CNC | 0.16 (0.04) | 0.16 (0.04) | 0.19 (0.08) | 0.27 (0.13) | 0.20 |
| WL KERNEL | **0.39 (0.00)** | 0.21 (0.00) | 0.00 (0.00) | 0.00 (0.00) | 0.15 |
| GC KERNEL | 0.02 (0.00) | 0.00 (0.00) | 0.29 (0.00) | 0.00 (0.00) | 0.08 |
| $\Gamma_{1\text{-HOT}}$ | 0.17 (0.08) | **0.25 (0.06)** | 0.12 (0.09) | 0.23 (0.08) | 0.19 |
| $\Gamma_{GIN}$ | 0.24 (0.04) | 0.18 (0.04) | 0.29 (0.11) | **0.28 (0.06)** | 0.25 |
| $\Gamma_{RPGIN}$ | 0.26 (0.05) | 0.20 (0.04) | 0.25 (0.12) | 0.20 (0.05) | 0.23 |
| **CIGAv1** | 0.22 (0.07) | **0.23 (0.09)** | **0.40 (0.06)** | 0.29 (0.08) | **0.29** |
| **CIGAv2** | **0.27 (0.07)** | 0.22 (0.05) | **0.31 (0.12)** | 0.26 (0.08) | **0.27** |
| ORACLE (IID) | 0.32 (0.05) | 0.37 (0.06) | 0.39 (0.09) | 0.33 (0.05) | |

sistently and significantly outperform previous methods, including previous specialized methods $\Gamma$ GNNs [11] for combating graph size shifts, demonstrating the generality and superiority of CIGA.

**Comparisons with advanced ablation variants.** As discussed in Sec. 3.3, CIGA can be reduced to interpretable GNNs and contrastive learning approaches. However, across all experiments, we can observe that neither the advanced interpretable GNNs (DIR) nor sophisticated contrastive objectives with specialized sampling strategy (CNC) can yield satisfactory OOD performance, which serves as *strong evidence* for the necessities of the decomposition as well as the objective in CIGA. Furthermore, although CIGAv1 can outperform CIGAv2 when we may have a relatively accurate

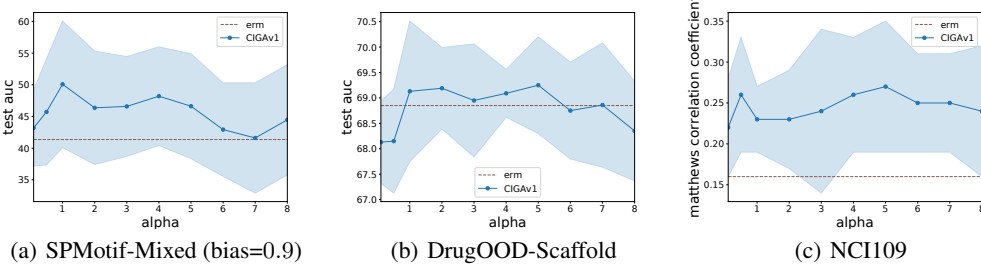

(a) SPMotif-Mixed (bias=0.9)     (b) DrugOOD-Scaffold     (c) NCI109

Figure 4: Hyperparameter sensitivity analysis on the coefficient of contrastive loss ($\alpha$).

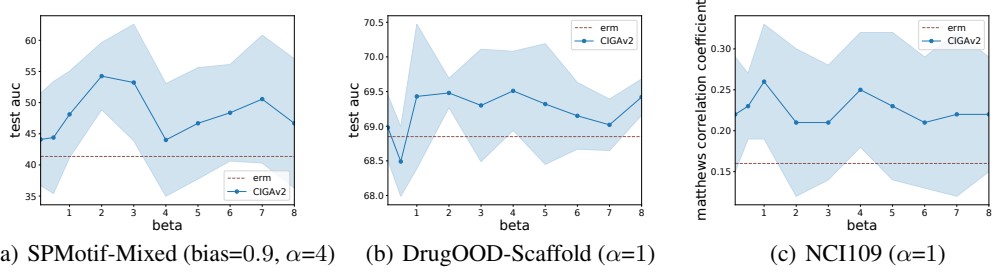

(a) SPMotif-Mixed (bias=0.9, $\alpha$=4)     (b) DrugOOD-Scaffold ($\alpha$=1)     (c) NCI109 ($\alpha$=1)

Figure 5: Hyperparameter sensitivity analysis on the coefficient of hinge loss ($\beta$).

$s_c$, the improvements in CIGAv1 are not as stable as CIGAv2 or even unsatisfactory when the assumption is violated. This phenomenon also reveals the superiority of CIGAv2 in practice.

**Hyperparameter sensitivity analysis.** To examine how sensitive CIGA is to the hyperparamters $\alpha$ and $\beta$ for contrastive loss and hinge loss, respectively. We conduct experiments based on the hardest datasets from each table (i.e., SPMotif-Mixed with the bias of 0.9, DrugOOD-Scaffold and the NCI109 datasets from Table 1, Table 2, and Table 3, respectively.) with different $\alpha$ and $\beta$. When changing the value of $\beta$, we fix the $\alpha$ to a specific value under which the model has a relatively good performance (but not the best, to fully examine the robustness of CIGA in practice).

The results are shown in Fig. 4 and Fig. 5. It can be found that both CIGAv1 and CIGAv2 are robust to different values of $\alpha$ and $\beta$, respectively, across different datasets and distribution shifts. Besides, the results also reflect the effects of the additional penalty terms in CIGA. For example, in Fig. 16, when $\alpha$ is too small, the invariance of the identified invariant subgraphs $\widehat{G}_c$ may not be guaranteed, resulting worse performances. Similarly, as shown in Fig. 17, when $\beta$ becomes too small, some part of the spurious subgraph may still appear in the estimated invariant subgraphs, which yields worse performances. Besides, when $\alpha$ and $\beta$ become too large, the optimization of CIGA can be affected due to their intrinsic conflicts with ERM, hence a better optimization scheme for CIGA can be a promising future direction [18]. We provide more details and additional analysis on the efficiency of CIGA and single environment OOD generalization performance of CIGA in Appendix G.4, as well as the visualization examples of the identified invariant subgraph in Appendix G.5.

## 5 Conclusions

We studied the OOD generalization on graphs via graph classification, and propose a new solution CIGA through the lens of causality. By modeling potential distribution shifts on graphs with SCMs, we generalized and instantiated the invariance principle to graphs, which was shown to have promising theoretical and empirical OOD generalization ability under a variety of distribution shifts.

## Acknowledgments and Disclosure of Funding

We thank the reviewers for their valuable comments. This work was supported by GRF 14208318 from the RGC of HKSAR and CUHK direct grant 4055146. TL was partially supported by Australian Research Council Projects DP180103424, DE-190101473, IC-190100031, DP-220102121, and FT-220100318. YZ and BH were supported by the RGC Early Career Scheme No. 22200720, NSFC Young Scientists Fund No. 62006202, Guangdong Basic and Applied Basic Research Foundation No. 2022A1515011652, and Tencent AI Lab Rhino-Bird Gift Fund.

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
