# Appendix of CIGA

## Contents

# A Broader Impacts

Considering the wide applications and high sensitivity of GNNs to distribution shifts and spurious correlations, it is important to develop GNNs that are able to generalize to OOD data, especially for realistic scenarios such as AI-aided Drug Discovery where OOD data are ubiquitous. By formulating OOD generalization problem on graphs using causality, our work can serve as an initiate step towards tackling OOD generalization problem on graphs, with the hope to empower GNNs for broader applications and social benefits. Besides, this paper does not raise any ethical concerns. This study does not involve any human subjects, practices to data set releases, potentially harmful insights, methodologies and applications, potential conflicts of interest and sponsorship, discrimination/bias/fairness concerns, privacy and security issues, legal compliance, and research integrity issues.

# B  More Discussions on Related Work and Future Directions

## B.1  More backgrounds

We give more background introduction about GNNs and Invariant Learning in this section.

**Graph Neural Networks.** Let $G = (A, X)$ denote a graph with $n$ nodes and $m$ edges, where $A \in \{0, 1\}^{n \times n}$ is the adjacency matrix, and $X \in \mathbb{R}^{n \times d}$ is the node feature matrix with a node feature dimension of $d$. In graph classification, we are given a set of $N$ graphs $\{G_i\}_{i=1}^N \subseteq \mathcal{G}$ and their labels $\{Y_i\}_{i=1}^N \subseteq \mathcal{Y} = \mathbb{R}^c$ from $c$ classes. Then, we train a GNN $\rho \circ h$ with an encoder $h : \mathcal{G} \to \mathbb{R}^h$ that learns a meaningful representation $h_G$ for each graph $G$ to help predict their labels $y_G = \rho(h_G)$ with a downstream classifier $\rho : \mathbb{R}^h \to \mathcal{Y}$. The representation $h_G$ is typically obtained by performing pooling with a READOUT function on the learned node representations:

$$h_G = \text{READOUT}(\{h_u^{(K)} | u \in V\}), \tag{6}$$

where the READOUT is a permutation invariant function (e.g., SUM, MEAN) [107, 115, 70, 107, 19, 68], and $h_u^{(K)}$ stands for the node representation of $u \in V$ at $K$-th layer that is obtained by neighbor aggregation:

$$h_u^{(K)} = \sigma(W_K \cdot a(\{h_v^{(K-1)}\} | v \in \mathcal{N}(u) \cup \{u\})), \tag{7}$$

where $\mathcal{N}(u)$ is the set of neighbors of node $u$, $\sigma(\cdot)$ is an activation function, e.g., ReLU, and $a(\cdot)$ is an aggregation function over neighbors, e.g., MEAN.

**Invariant Learning.** Invariant learning typically considers a supervised learning setting based on the data $\mathcal{D} = \{\mathcal{D}^e\}_e$ collected from multiple environments $\mathcal{E}_{\text{all}}$, where $\mathcal{D}^e = \{G_i^e, y_i^e\}$ is the dataset from environment $e \in \mathcal{E}_{\text{all}}$. $(G_i^e, y_i^e)$ from a single environment $e$ are considered as drawn independently from an identical distribution $\mathbb{P}^e$. The goal of OOD generalization is to train a GNN $\rho \circ h : \mathcal{G} \to \mathcal{Y}$ with data from training environments $\mathcal{D}_{\text{tr}} = \{\mathcal{D}^e\}_{e \in \mathcal{E}_{\text{tr}} \subseteq \mathcal{E}_{\text{all}}}$, and generalize well to all (unseen) environments, i.e., to minimize:

$$\min_{\rho, h} \max_{e \in \mathcal{E}_{\text{all}}} R^e(\rho \circ h), \tag{8}$$

where $R^e$ is the empirical risk under environment $e$ [97, 76, 4]. More details can be referred in [2].

## B.2  Detailed related work

**GNN Explainability.** Works in GNN explainability aim to find a subgraph of the input graph as the explanation for the prediction of a GNN model [116, 122]. Although some may leverage causality in explanation generation [53], they mostly focus on understanding the predictions of GNNs in a post-hoc manner instead of OOD generalization. Recently there are two works aiming to provide robust explanations under distribution shifts, i.e., GIB [120] and DIR [104], and both of them focus on tackling FIIF spurious correlations (Assumption C.2). The theoretical guarantees of GIB follows the theory of information bottleneck [95], while GIB can not solve PIIF spurious correlations (Assumption C.3). As both FIIF and PIIF widely exist in realistic scenarios, failing to solve either of them could result in severe performance degradation in practice [4, 2, 5, 71]. While for DIR, though as a generalization of Chang et al. [14] to graphs, can not provide any theoretical guarantees under FIIF spurious correlations as shown in Appendix D.4, nor under PIIF spurious correlations.

**GNN Extrapolation.** Recently there is a surge of attention in improving the extrapolation ability of GNNs and apply them to various applications, such as mathematical reasoning [84, 85], physics [6, 83], and graph algorithms [94, 98, 108, 105]. Xu et al. [110] study the neural network extrapolation ability from a geometrical perspective. Han et al. [36] improve OOD drug discovery by mitigating the overconfident misprediction issue. Knyazev et al. [46], Yehudai et al. [113] focus on the extrapolation of GNNs in terms of graph sizes, while making additional assumptions on the knowledge about ground truth attentions and access to test inputs. Bevilacqua et al. [11] study the graph size extrapolation problem of GNNs through a causal lens, while the induced invariance principle is built upon assumptions on the specific family of graphs. Different from these works, we consider the GNN extrapolation as a causal problem, establish generic SCMs that are compatible with several graph generation models, as well as, more importantly, different types of distribution shifts. Hence, the induced the invariance principle and provable algorithms built upon the SCMs in our work can generalize to multiple graph families and distribution shifts.

Additionally, Wu et al. [103] propose causal models as well as specialized objectives to extrapolate nodes with different neighbors. However, their formulation is limited to node classification task and specific spurious correlation type. In contrast, the induced invariance principle in Wu et al. [103], can be seen as a extension of CIGA for node classification, where we cab identify an invariant subgraph from the $K$-hop neighbor graph of each node, and making predictions based on it, i.e., $Y \perp\!\!\!\perp E | G_c^{\text{ego}} \subseteq G_u^{\text{ego}}$ for node $u$. We leave specific formulation and implementation to future works.

**Causality and OOD Generalization.** Causality comes to the stage for demystifying and improving the huge success of machine learning algorithms to further advances [75, 86, 87]. One of the most widely applied concept from causality is the Independent Causal Mechanism (ICM) that assumes conditional distribution of each variable given its causes (i.e., its mechanism) does not inform or influence the other conditional distributions [74, 77]. The invariance principle is also induced from the ICM assumption. Once proper assumptions about the underlying data generation process via Structural Causal Models (SCM) are established, it is promising to apply the invariance principle to machine learning models for finding an invariant representation about the causal relationship between the underlying causes and the label [76, 4]. Consequently, models built upon the invariant representation can generalize to unseen environments or domains with guaranteed performance [76, 79, 4, 81, 10, 48, 34, 49, 23, 2]. The arguably first formulation of invariance principle was introduced by Peters et al. [76]. Arjovsky et al. [4] propose a novel formulation of learning causal invariance in representation learning, i.e., IRM, show how it connects with existing areas such as distributional robust optimization [72] and generalization [123], and prove its effectiveness in addressing PIIF spurious correlations (Assumption C.3). However, in practice, both PIIF and FIIF (Assumption C.2) can appear in data, while IRM can fail in these cases [5, 71]. Ahuja et al. [2] then propose to add information bottleneck criteria into the IRM formulation to address the issue. However, their results are restricted to linear regime and also require environment partitions to distinguish the sources of distribution shifts. Recently, Creager et al. [23] and Lin et al. [55] propose new OOD objectives to relieve the needs for environment partitions, but limited to PIIF spurious types and linear regime. Besides, Lin et al. [54] identify the overfitting problem as a key challenge when applying IRM on large neural networks. Zhou et al. [127] propose to alleviate this problem by imposing sparsity constrain.

In parallel invariant learning approaches, Sagawa* et al. [81] propose to regularize the worst group in group distributionally robust optimization (GroupDro). Zhang et al. [124] propose a contrastive approach to tackle GroupDro when the group partitions are not available. However, minimizing the gap between worst group risk and averaged risk can not yield a OOD generalizable predictors in our circumstances. Besides, traditional approaches to tackle OOD generalization also include Domain Adaption, Transfer Learning and Domain Generalization[79, 21, 31, 93, 52, 27, 61, 100], which aim to learn the class conditional invariant representation shared across source domain and target domain. However, they all require a stronger assumption on the availability of target domain data or the ground truth predictors [34, 2], hence are not able to yield predictors with OOD generalization guarantees. We refer interested readers to Pearl [75], Schölkopf [86], Schölkopf et al. [87] for an in-depth understanding, and Gulrajani and Lopez-Paz [34], Ahuja et al. [2] for a thorough overview.

### B.3 More discussions on connections of CIGA with existing work

Although primarily serving for graph OOD generalization problem, our theory complements the identifiability study on graphs through contrastive learning, and aligns with the discoveries in the image domain that contrastive learning learns to isolate the content ($C$) and style ($S$) [128, 50]. Moreover, our results also partially explain the success of graph contrastive learning [118, 60, 119], where GNNs may implicitly learn to identify the underlying invariant subgraphs for prediction.

**On expressivity of graph encoder in CIGA.** The expressivity of CIGA is essentially constrained by the encoders embedded for learning graph representations. During isolating $G_c$ from $G$, if the encoder can not differentiate two isomorphic graphs $G_c$ and $G_c \cup G_s^p$ where $G_s^p \subseteq G_s$, then the featurizer will fail to identify the underlying invariant subgraph. Moreover, the classifier will also fail if the encoder can not differentiate two non-isomorphic $G_c$s from different classes. Thus, adopting more powerful graph representation encoders into CIGA can improve the OOD generalization.

**On CIGA and graph information bottleneck.** Under the FIIF assumption on latent interaction, the independence condition derived from causal model can also be rewritten as $Y \perp\!\!\!\perp S|C$ (similar to that in DIR [104] as they also focus on FIIF), which further implies $Y \perp\!\!\!\perp S|\widehat{G}_c$. Hence it is natural to use Information Bottleneck (IB) objective [95] to solve for $G_c$:

$$
\begin{aligned}
\min_{f_c, g} & \ R_{G_c}(f_c(\widehat{G}_c)), \\
\text{s.t. } G_c = & \underset{\widehat{G}_c = g(G) \subseteq G}{\arg\max} \ I(\widehat{G}_c, Y) - I(\widehat{G}_c, \mathcal{G}),
\end{aligned}
\tag{9}
$$

which explains the success of many existing works in finding predictive subgraph through IB [120]. However, the estimation of $I(\widehat{G}_c, G)$ is notoriously difficult due to the complexity of graph, which can lead to unstable convergence as observed in our experiments. In contrast, optimization with contrastive objective in CIGA as Eq. 5 induces more stable convergence.

**On CIGA for node classifications.** As the task of node classification can be viewed as graph classification based on the ego-graphs of a node, our analysis and discoveries can generalize to node classification. More specifically, the invariance principle for node classification can be implemented by identifying an invariant subgraph from the $K$-hop neighbor graph of each node, and making predictions based on it, i.e., $Y \perp\!\!\!\perp E|G_c^{\text{ego}} \subseteq G_u^{\text{ego}}$ for node $u$ [103].

### B.4 Discussions on limitations of CIGA and future directions

**Better graph generation modeling.** Compared to Bevilacqua et al. [11], we do not specify a specific graph family in the SCM for graph generation process. Since our focus is to describe the potential distribution shifts with SCMs, in Assumption 2.1, we aim to build a SCM that is compatible to many graph generation processes [89, 57, 117, 59]. However, it is often the case that practitioners have certain inductive knowledge about the graph generation process, which may imply useful leads and invariance in modeling the generation process [111, 30, 56]. In Appendix C.1, we provide an example about incorporating the graphon [57] knowledge into the SCMs, which derives similar solutions as in the literature [113, 11]. Therefore, we believe it is promising to leverage more additional knowledge for more precise graph generation modeling and better OOD generalization on graphs.

**Better contrastive sampling.** Typical contrastive or graph contrastive learning approaches leverage augmentation techniques as well as sophisticated sampling strategies during the positive or negative pairs selection [20, 82, 96, 118, 119]. A better augmentation or sampling strategy can benefit the OOD generalization in general as shown by Kügelgen et al. [50] and Zhang et al. [124]. Since our implementation of CIGA in this work aims to verify the theoretical findings, we do not apply sophisticated augmentation or sampling during the sampling while simply using the supervised contrastive approach [42]. Nevertheless, it is promising to leverage better augmentation and contrastive strategy to improve the generalization ability in CIGA [121].

**More sophisticated architectures/parameter tunning.** The CIGA framework introduced in Sec. 3 can have multiple implementations. We choose interpretable architectures in our experiments for the purpose of concept verification. Essentially, different architectures can have different advantages and limitations. For the interpretable GNNs used in our experiments, it can provide interpretability for the results (as shown in Appendix G.5), but still requires more training time (as shown in Appendix G.4).

Therefore, it may not be applicable to some resource-limited scenarios such as Edge-AI. Besides, the approximation may also be limited to the chosen architectures. More sophisticated architectures can be incorporated, such as identifying and disentangling $G_c$ at the latent space [86, 87]. Moreover, as shown in Appendix G.4, CIGA still requires certain additional tunning efforts for the objectives. Hence we believe it is also a promising future direction to reduce the parameter tunning by leveraging better optimization techiniques [88, 18]

## C  Full Structural Causal Models on Graph Generation

Due to the space constraints in the main paper, we make some simplifications when giving the SCMs on the graph generation process. Hence in this section, supplementary to the graph generation process in Sec. 2.2, we provide full SCMs on the graph generation process in this section as shown in Fig. 6. Formal descriptions are given as Assumptions C.1, C.2, C.3, C.4.

To begin with, we take a latent-variable model perspective on the graph generation process and assume that the graph is generated through a mapping $f_{\text{gen}} : \mathcal{Z} \to \mathcal{G}$, where $\mathcal{Z} \subseteq \mathbb{R}^n$ is the latent space and $\mathcal{G} = \cup_{N=1}^{\infty} \{0,1\}^N \times \mathbb{R}^{N \times d}$ is the graph space. Let $E$ denote environments. Following previous works [50, 2], we partition the latent variable from $\mathcal{Z}$ into an invariant part $C \in \mathcal{C} = \mathbb{R}^{n_c}$ and a varying part $S \in \mathcal{S} = \mathbb{R}^{n_s}$, s.t., $n = n_c + n_s$, according to whether they are affected by $E$. Similarly in images, $C$ and $S$ can represent content and style while $E$ can refer to the locations where the images are taken [7, 125, 50]. While in graphs, $C$ can be the latent variable that controls the generation of functional groups in a molecule, which can not be affected by the changes of environments, such as species (or scaffolds), experimental environment for examining the chemical property (or assays) [40]. On the contrary, the other latent variable $S$ inherits environment-specific information thus can further affect the finally generated graphs. Besides, $C$ and $S$ can have multiple types of interactions at the latent space with environments $E$ and labels $Y$, which will generate different types of spurious correlations [2].

**Assumption C.1** (Graph generation SCM)**.**

$$(Z_A^c, Z_X^c) := f_{\text{gen}}^{(A,X)^c}(C), \ G_c := f_{\text{gen}}^{G_c}(Z_A^c, Z_X^c),$$
$$(Z_A^s, Z_X^s) := f_{\text{gen}}^{(A,X)^s}(S), \ G_s := f_{\text{gen}}^{G_s}(Z_A^s, Z_X^s),$$
$$G := f_{\text{gen}}^{G}(G_c, G_s).$$

Specifically, the graph generation process is shown as Fig. 6(a). The generation mapping $f_{\text{gen}}$ is decomposed into $f_{\text{gen}}^{(A,X)^c}, f_{\text{gen}}^{G_c}, f_{\text{gen}}^{(A,X)^s}, f_{\text{gen}}^{G_s}$ and $f_{\text{gen}}^{G}$ to control the generation of $(Z_A^c, Z_X^c)$, $G_c$, $(Z_A^s, Z_X^s)$, $G_s$, and $G$, respectively. Given the variable partitions $C$ and $S$ at the latent space $\mathcal{Z}$, they control the generation of the adjacency matrix and features for the invariant subgraph $G_c$ and spurious subgraph $G_s$ through two pairs of latent variables $(Z_A^c, Z_X^c)$ and $(Z_A^s, Z_X^s)$, respectively. $Z_A^c$ and $Z_A^s$ will control the structure-level properties in the generated graphs, such as degrees, sizes, and subgraph densities. While $Z_X^c$ and $Z_X^s$ mainly control the attribute-level properties in the generated graphs, such as homophily. Then, $G_c$ and $G_s$ are entangled into the observed graph $G$ through $f_{\text{gen}}^{G}$. It can be a simply JOIN of a $G_c$ with one or multiple $G_s$, or more complex generation processes controlled by the latent variables [89, 57, 117, 59, 11]. Note that since our focus is to describe the potential distribution shifts with SCMs, in Assumption 2.1, we aim to build a SCM that is compatible to many graph generation processes [89, 57, 117, 59]. In fact, in Appendix C.1, we showcase how our SCMs can generalize to specific graph families studied in the literature [11, 104, 103], when given more additional knowledge about the graph generation process. Nevertheless, we believe integrating specific graph generation processes and their implications to improving OOD generalization on graphs would be a promising future direction, as discussed in Appendix B.4.

Due to the correlation between $E$ and $G$, graphs collected from different environments can have different structure-level properties such as degrees, graph sizes, and subgraph densities, as well as feature-level properties such as homophily [46, 113, 11, 17]. Meanwhile, all of them can spuriously correlated with the labels depending on how the underlying latent variables are interacted with each others. The interaction types can be further divided into two axiom types FIIF and PIIF, as well as the mixed one MIIF. Previous OOD methods such as GIB [120] and DIR [104] mainly focus on FIIF case, while others such as IRM [4] mainly focuses on the PIIF case. Evidences show that

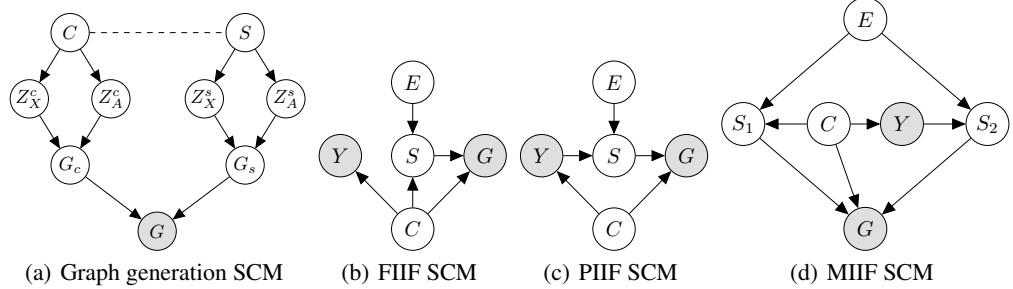

| (a) Graph generation SCM | (b) FIIF SCM | (c) PIIF SCM | (d) MIIF SCM |

Figure 6: Full SCMs on Graph Distribution Shifts.

failing to model either of them when developing the OOD objectives can have serious performance degenerations in practice [5, 71]. That is why we aim to model both of them in our solution.

**Assumption C.2** (FIIF SCM).

$$Y := f_{\text{inv}}(C),\ S := f_{\text{spu}}(C, E),\ G := f_{\text{gen}}(C, S).$$

**Assumption C.3** (PIIF SCM).

$$Y := f_{\text{inv}}(C),\ S := f_{\text{spu}}(Y, E),\ G := f_{\text{gen}}(C, S).$$

**Assumption C.4** (MIIF SCM).

$$Y := f_{\text{inv}}(C),\ S_1 := f_{\text{spu}}(C, E),\ S_2 := f_{\text{spu}}(Y, E),\ G := f_{\text{gen}}(C, S_1, S_2).$$

As for the interactions between $C$ and $S$ at the latent space, we categorize the interaction modes into Fully Informative Invariant Features (FIIF, Fig. 6(b)), and Partially Informative Invariant Features (PIIF, Fig. 6(c)), depending on whether the latent invariant part $C$ is fully informative about label $Y$, i.e., $(S, E) \perp\!\!\!\perp Y | C$. It is also possible that FIIF and PIIF are entangled into a Mixed Informative Invariant Features (MIIF,Fig. 6(d)). We follow Arjovsky et al. [4], Ahuja et al. [2] to formulate the SCMs for FIIF and PIIF, where we omit noises for simplicity [74, 77]. Since MIIF is built upon FIIF and PIIF, we will focus on the axiom interaction modes (FIIF and PIIF) in this paper, while most of our discussions can be extended to MIIF or more complex interactions built upon FIIF and PIIF.

Among all of the interaction modes, $f_{\text{gen}}$ corresponds to the graph generation process in Assumption C.1. $f_{\text{spu}}$ is the mechanism describing how $S$ is affected by $C$ and $E$ at the latent space. In FIIF, $S$ is directly controlled by $C$ while in PIIF, indirectly controlled by $C$ through $Y$, which can exhibit different behaviors in practice [2, 71]. Additionally, in MIIF, $S$ is further partitioned into $S_1$ and $S_2$ depending on whether it is directly or indirectly controlled by $C$, respectively. Moreover, $f_{\text{inv}} : \mathcal{C} \to \mathcal{Y}$ indicates the labeling process, which assigns labels $Y$ for the corresponding $G$ merely based on $C$. Consequently, $\mathcal{C}$ is better clustered than $\mathcal{S}$ when given $Y$ [13, 15, 86, 87], which also serves as the necessary separation assumption for a classification task [69, 16, 65].

**Assumption C.5** (Latent Separability). $H(C|Y) \leq H(S|Y)$.

### C.1 Discussions on specific cases of the SCMs

Although our primary focus in this work is to characterize general graph distribution shifts that could happen in practice without any additional knowledge about the underlying graph family, and derive the corresponding solutions, our SCMs (Fig. 6) can generalize to specific cases studied in previous works, when incorporating more inductive biases about the underlying graph family [11, 104, 103]. Specifically, we illustrate the specialized SCMs in Fig. 7 for the SCM studied in [11] which assumes the graphs are generated following a graphon model [57].

When with the additional knowledge about the underlying graph generative model, the graph generation SCM (Fig. 6(a)) and the FIIF SCM (Fig. 6(b)) together generalizes to the graphon SCM studied in [11]. We now give a brief description in the below.

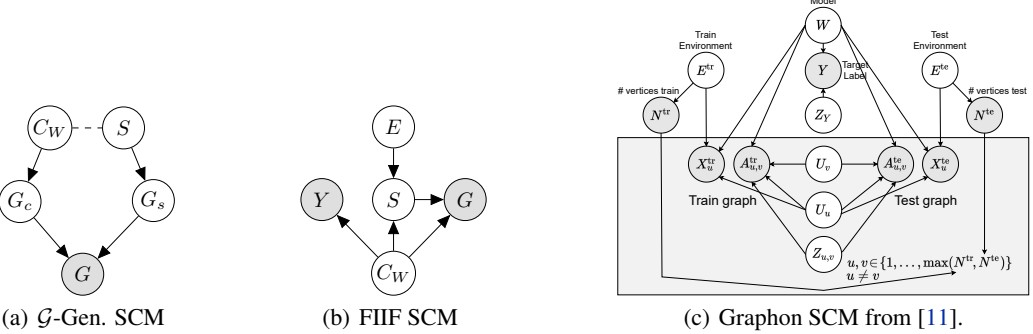

(a) $\mathcal{G}$-Gen. SCM       (b) FIIF SCM       (c) Graphon SCM from [11].

Figure 7: Specialized graph generation SCMs when incorporating additional knowledge.

Specifically, shown as in Fig. 7(a), $C$ now is instantiated as a graphon model $C_W \sim \mathbb{P}(C_W)$, where $C_W : [0,1]^2 \to [0,1]$ is a random symmetric measurable function sampled from the set of all symmetric measurable functions [57]. Besides, the label $Y$ is determined according to $C_W$. Then, $C_W$ will further control the generation of the adjacency matrix $G_c = A^c$ through graphon generative process:

$$A_{u,v}^c := \mathbb{I}(Z_{u,v} > C_W(U_u, U_v)), \ \forall u, v \in V,$$

where $Z_{u,v}$ is an independent uniform noises on $[0,1]$ for each possible edge $(u,v)$ in the graph. Bascially, $Z$ and $U$ are inherited from the graphon SCM as Fig. 7(c).

On the other hand, as $S$ does not imply any information about $Y$ in this case, it resembles the FIIF SCM (Fig. 6(b)). In other words, $(S, E) \perp\!\!\!\perp Y | C$ still holds. Moreover, the node attributes $G_s = X^s$ are generated jointly influenced by the environment $E$ and the graphon $C_W$ through $S$:

$$X_v := f_{\text{gen}}^s(S), \ S := f_{\text{spu}}(E, C_W),$$

which resembles the attribute generation in Fig. 7(c).

Then, both $G_c$ and $G_s$ are concatenated together. In a simplistic case intuitively, we can regard $G_c$ only contains the edges in $G$ and $G_s$ only contains the node attributes. Since the graphon model mainly controls the edge connection, the edge connection patterns, e.g., motif appearance frequency or subgraph densities, acts as a informative indicator for the label $Y$. In contrast, the node attributes and its numbers would be affected by the environments. A GNN model is prone to the changes of the environments if it overfits to some spurious patterns about the graph sizes or the attributes. While if the GNN model can leverage the connection patterns to make predictions, it remain invariant to the changes of environments, or the spurious patterns such as graph sizes and node attributes, which resembles the solutions derived in [113, 11]. Besides, it also partially explains why CIGA can generalize to OOD graphs studied in these works [113, 11].

In addition to the graphon SCM, essentially, the SCM studied in [104] resembles the FIIF SCM, and that of [103] resembles PIIF SCM, which also serves as partial evidence for the superiority OOD generalization performances of CIGA.

## D    More Details about Failure Case Studies in Sec. 2.3

In this section, we provide details on failure case studies in Sec. 2.3. We first elaborate the empirical evaluation setting where we construct a synthetic graph datasets to probe the behaviors of existing methods in OOD generalization on graphs.

### D.1    More empirical details about failure case study in Sec. 2.3

To begin with, we construct 3-class synthetic datasets based on BAMotif [58] and follow Wu et al. [104] to inject spurious correlations between motif graph and base graph during the generation. In this graph classification task, the model needs to tell which motif the graph contains, e.g., "House" or "Cycle" motif, as shown in Fig. 8. We inject the distribution shifts in the training data while

keeping the test data and validation data without the biases. For structure-level shifts, we introduce the artificial bias based on FIIF, where the motif and the base graph are spuriously correlated with a probability of various bias. For mixed shifts, we additionally introduced attribute-level shifts based on FIIF, where all of the node features are spuriously correlated with a probability of various bias. The number of training graphs is 600 for each class and the number of graphs in validation and test set is 200 for each class. More construction details are given in Appendix G.

For the GNN encoders, by default, we use 3-layer GCN [45] with mean readout, a hidden dimension of 64, and JK jump connections [106] at the last layer. During training, we use a batch size of 32, learning rate of $1e-3$ with Adam optimizer [43], and batch normalization between hidden layers [39]. Meanwhile, to stabilize the training, we also use dropout [91] of 0.1 and early stop the training when the validation accuracy does not increase till 5 epoch after first 20 epochs. All of the experiments are repeated 5 times, and the mean accuracy as well as variance are reported and plotted. When using IRM objective [4], as the environment partitions are not available, we generate 2 environments with random partitions.

### D.2 More discussions about failure case study in Sec. 2.3

In Fig. 9, 10, 11, 12, we investigate whether existing training objectives (ERM and IRM), adding more message passing, as well as using expressive GNNs, can improve the OOD generalization ability on graphs. Here we also provide a additional discussion in complementary to the discussions on OOD generalization performance of ERM and IRM objectives in Sec. 2.3.

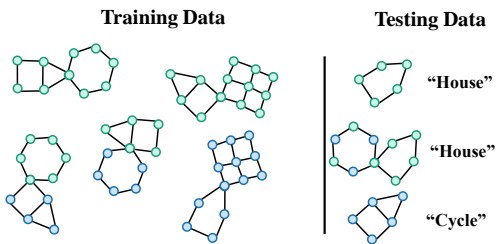

*Can better architectures improve OOD generalization of GNNs?*

**Adding more message passing turns.** It is a common practice in GNNs to denoise the signals by aggregating more neighbors with higher layers, or enhance the expressive power with more powerful readout functions [106, 107, 112]. Aggregating neighbor information with more layers to denoise the input signal, or enhancing the expressivity with more powerful readout functions, are two common choices in GNNs to improve the generalization ability [106, 51, 107, 112]. However, in the experiments next, we empirically found that GCNs with more layers and more powerful readout operations are still sensitive to distribution shifts. In particular, stacking more layers helps denoising certain shifts, while

Figure 8: Failure cases of existing methods. GNNs are required to classify whether the graph contains a "house" or "cycle", where the colors represent node features. However, distribution shifts in the training exists at both structure level (From left to right: "house" mostly co-occur with a hexagon), attribute level (From upper to lower: graphs nodes are mostly green colored if they contain "house", or blued colored if they contain "cycle"), and graph sizes, making GNNs hard to capture the invariance. *ERM can fail* for leveraging the shortcuts and predict graphs that have a hexagon or have mostly green nodes as "house". *IRM can fail* when test data is not sufficiently supported by the training data.

the OOD performance would drop more sharply when the bias increases. Intuitively, if the spurious features from nodes cannot be eliminated by the denoising property of a deeper GNN, they would spread among the whole graph more widely, which in turn leads to stronger spurious correlations. Besides, the spurious correlations would be more difficult to be disentangled if there are distribution shifts at both structure-level and attribute-level. Since the node representations from hidden layers can also encode graph topology features [107], distribution shifts introduced through $Z_A^s$ and $Z_X^s$ will doubly mix at the learned features. In the worst case, the information about $Z_A^c$ and $Z_X^c$ could be partially covered by or even replaced by $Z_A^s$ and $Z_X^s$. This will make OOD generalization of message passing GNNs trained through ERM much more difficult or even impossible. Besides, as the node representations of $1 \le i \le k$-th layer can also encode graph topology features [107], which, if spuriously correlated with labels through $Z_A^s$ and entangled with part of invariant node features, i.e., $Z_X^c$, in the worst case, can greatly improve the difficulty or even make the OOD generalization impossible for neighbor aggregation GNNs trained with ERM.

**Using more expressive GNNs.** Previous results on the expressivity of GNNs show that GNNs are limited to distinguish isomorphic graphs at most as 1-WL/2-WL test can distinguish [107]. After

that, many follow-up variants are proposed to improve the expressivity of GNNs [68]. However, if the labels are spuriously correlated with certain subgraphs, even the GNN has high expressivity can still be prone to distribution shifts. In a idealistic case, when classifying a graph with a highly expressive GNN, it reduces to the linear or discrete feature case on the Euclidean regime. In this case, there exists many evidences showing that neural networks can fail to generalize to OOD data without a proper objective [7, 24, 4, 81, 10, 49, 23, 48, 2]. Empirically, we use $k$-GNNs [66] to verify the intuition and observe similar failures for this provably more expressive GNN as basic GNN variants.

### D.3 More empirical results about failure case study in Sec. 2.3

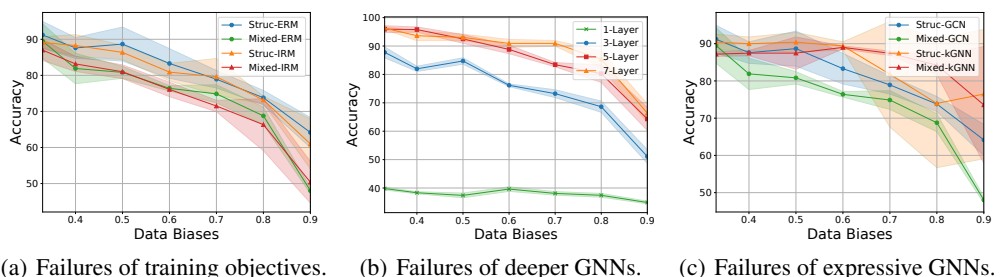

(a) Failures of training objectives.  (b) Failures of deeper GNNs.  (c) Failures of expressive GNNs.

Figure 9: Failure of existing methods on SPMotif with FIIF attribute shifts.

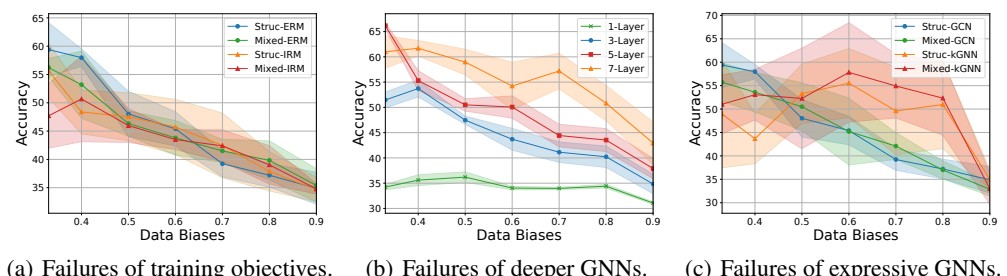

(a) Failures of training objectives.  (b) Failures of deeper GNNs.  (c) Failures of expressive GNNs.

Figure 10: Failure of existing methods on SPMotif with FIIF attribute shifts and graph size shifts.

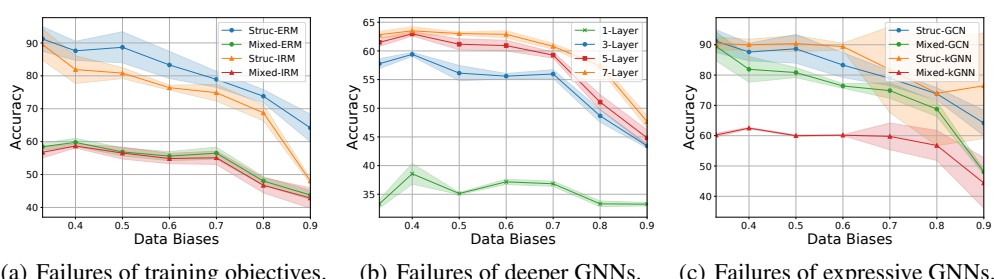

(a) Failures of training objectives.  (b) Failures of deeper GNNs.  (c) Failures of expressive GNNs.

Figure 11: Failure of existing methods on SPMotif with PIIF attribute shifts.

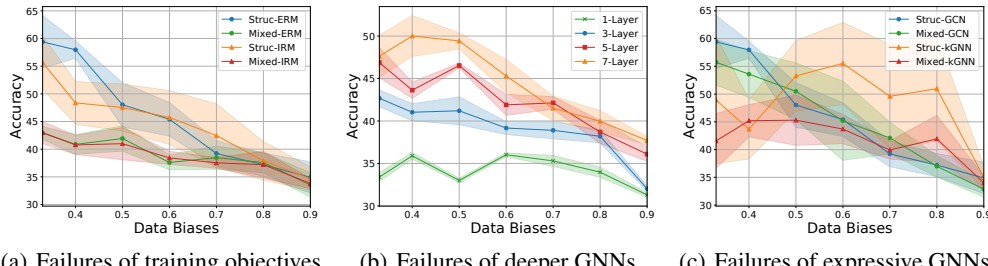

(a) Failures of training objectives.  (b) Failures of deeper GNNs.  (c) Failures of expressive GNNs.

Figure 12: Failure of existing methods on SPMotif PIIF attribute shifts with graph size shifts.

To explore the behaviors of aforementioned methods against complicated distribution shifts on graphs, we first modify construction method in Wu et al. [104] to construct dataset for Fig. 9, where only FIIF structure-level spurious correlations are injected. Then we also inject FIIF attribute-level shifts, by setting the node attributes to constant vectors which is spuriously correlated with the labels. Furthermore, in Fig. 10, graph size shifts are added, which is exactly the SPMotif datasets used in DIR [104]. Besides, in Fig. 11, we can also change the FIIF attribute-level shifts to PIIF attribute-level shifts, where we flip the labels by a probability of $5\%$ and let the flipped label to be spuriously correlated with the node features, following the PIIF SCM in Fig. 6. Graph size shifts can also be injected in this case, shown as Fig. 12. Next, we summarize our findings from the experiments.

**Observation I: All existing methods are sensitive to distribution shifts.** From the Fig. 9, 10, 11, 12, we can observe that *all* GNNs are sensitive to distribution shifts. As the intensity of spurious correlation grows, GNNs are more likely to overfit to shortcuts presented either in the structure-level or attribute-level, which is similar to general deep learning models [33].

**Observation II: Higher variance also indicates unstable OOD performance.** Although GNNs show certain robustness against single distribution shifts, e.g., performances do not decrease sharply at the beginning in Fig. 9, when the spurious correlation grows stronger, the OOD performance become more *unstable*, e.g., higher variance. The reason is that, GNNs sometimes can directly learn about the desired information at some random initializations, since the task is relatively simple compared to reality. Hence the performance will be highly sensitive to the quality of initialized points at the beginning. Consequently, the performances from multiple runs would exhibit high variance. However, when the task becomes more difficult, GNNs will consistently be prone to distribution shifts, and the variance will be smaller, as shown in experiments (Sec. 4).

**Observation III: Entangling more distribution shifts can degenerate more GNN performance.** As implied by the graph generation SCMs in Fig. 6, distribution shifts can happen at both structure-level and attribute-level, and each of them can have different type of spurious correlation with the label. In Fig. 9, we can find that, when the attribute-level distribution shifts are mixed, the performance will be worse and more unstable. When the graph size shifts are mixed, this phenomenon will be more obvious, as shown in Fig. 10. This phenomenon also verifies the observations in Knyazev et al. [46] that attention mechanism in GNN is also sensitive to graph size shifts and can hardly learn the desired attention distributions without further guidance. Moreover, when the structure-level and attribute-level shifts have different spurious correlation types, i.e., when FIIF structure-level shifts and PIIF attribute-level shifts are both presented, the performance drop will be more serious, by comparing Fig. 9 to Fig. 11, as well as Fig. 10 to Fig. 12.

**Observation IV: Using more powerful architectures can not improve the OOD performance.** From the sub-figures (b) and (c) in Fig. 9, 10, 11, 12, we can also observe that neither adding more message passing turns nor using more expressive GNN architectures can be immune to distribution shifts. On the contrary, they also exhibit similar behaviors like basic GNN architectures. Specifically, adding more message passing runs show certain robustness against distribution shifts since they are more likely to learn the desired information during the optimization [109]. However, when the intensity of spurious correlation grows stronger, deeper GNNs are more likely to overfit to shortcuts hence their performances will drop more sharply. On the other hand, using provably more expressive GNN architectures can not improve the OOD performance, either. In Fig. 9, 10, 11, 12 we use 1-2-3-GNN following the algorithm of $k$-GNNs which is provably more expressive than 2-WL test [66]. When there are no graph size shifts, $k$-GNNs will have higher performance at the beginning. When

there are graph size shifts, $k$-GNNs will have a lower initial performance at the beginning. Then, as the spurious strength grows, $k$-GNNs can suddenly become seriously unstable, though $k$-GNNs can have higher averaged performance, which reflects unsatisfactory OOD performance as Observation II implies. When the intensity of spurious correlations grows even stronger, similar to deeper GNNs, OOD performances of $k$-GNNs will be more unstable and go down to similar level as that of normal GNN architectures. Hence, it calls for better optimization objectives as well as a suitable architectures to help improve the OOD generalization performance.

Beyond the empirical studies in previous section, we aim to accompany more formal discussions for explaining the failures of existing optimization objectives and architectures in the next sections.

### D.4 Theoretical discussions for failure case study in Sec. 2.3

**A motivating example.** To begin with, we follow Ahuja et al. [2] to introduce a formal example on the failures of GNNs optimized with ERM or IRM [97, 4] via a linear binary classification problem:

**Definition D.1** (Linear classification structural equation model (FIIF))**.**

$$Y := (w_{\text{inv}}^* \cdot C) \oplus N, \ N \sim \text{Ber}(q), \ N \perp (C, S),$$
$$X \leftarrow S(C, S),$$

where $w_{\text{inv}}^* \in \mathbb{R}^{n_c}$ with $\|w_{\text{inv}}^*\| = 1$ is the labeling hyperplane, $C \in \mathbb{R}^{n_c}$, $S \in \mathbb{R}^{n_s}$ are the corresponding invariant and varying latent variables, $N$ is Bernoulli binary noise with a parameter of $q$ and identical across all environments, $\oplus$ is the XOR operator, $S$ is invertible.

Given data generation process as Assumption C.1, and latent space interaction as Assumption C.2 or C.3, and strictly separable invariant features 2.4, consider a $k$-layer linearized GNN $\rho \circ h$ using mean as READOUT for binary graph classification, if $\cup_{e \in \mathcal{E}_{\text{te}}} \text{supp}(\mathbb{P}^e) \not\subseteq \cup_{e \in \mathcal{E}_{\text{tr}}} \text{supp}(\mathbb{P}^e)$:

(i) For graphs features generated as Definition D.1, $\rho \circ h$ optimized with ERM or IRM will fail to generalize OOD (Eq. 8) almost surely;

(ii) For graphs with more than two nodes, globally same node features generated as Definition D.1, and graph labels that are the same as global node labels, $\rho \circ h$ optimized with ERM or IRM will fail to generalize OOD (Eq. 8) almost surely;

For graph classification, if the number of nodes is fixed to one, it covers the linear classification as above. When $\cup_{e \in \mathcal{E}_{\text{te}}} \text{supp}(\mathbb{P}^e) \not\subseteq \cup_{e \in \mathcal{E}_{\text{tr}}} \text{supp}(\mathbb{P}^e)$, it implies the $S$ from training environments $\mathcal{E}_{\text{tr}}$ does not cover $S$ from testing environments, while $C$ can be covered. Moreover, the condition of strictly separable training data now can be formulated as $\min_{C \in \cup_{e \in \mathcal{E}_{\text{tr}}}(C \subseteq G^e)} \text{sgn}(w_{\text{inv}}^* \cdot C)(w_{\text{inv}}^* \cdot C) > 0$. Recall that ERM trains the model by minimizing the empirical risk (e.g., 0-1 loss) over all training data, and IRM formulates OOD generalization as:

$$\min_{\theta, f_c} \frac{1}{|\mathcal{E}_{\text{tr}}|} \sum_{e \in \mathcal{E}_{\text{tr}}} R^e(\rho \circ h)$$
$$\text{s.t. } \rho \in \arg\min_{\hat{\rho}} R^e(\hat{\rho} \circ h), \ \forall e \in \mathcal{E}_{\text{tr}}. \tag{10}$$

However, both ERM and IRM can not enable OOD generalization, i.e., finding the ground truth $w_{\text{inv}}^*$, following the Theorem 3 from Ahuja et al. [2]:

**Theorem D.2** (Insufficiency of ERM and IRM)**.** *Suppose each $e \in \mathcal{E}_{all}$ follows Definition. D.1, $C$ are strictly separable, bounded and satisfy the support overlap between $\mathcal{E}_{tr}$ and $\mathcal{E}_{te}$, and $S$ are bounded, if $S$ does not support the overlap, then both ERM and IRM fail at solving the OOD generalization problem.*

The reason is that, when $C$ from all environments are strictly separable, there can be infinite many Bayes optimal solutions given training data $\{G^e, y^e\}_{e \in \mathcal{E}_{\text{tr}}}$, while there is only one optimal solution that does not rely on $S$. Hence, the probability of generalization to OOD (finding the optimal solution) tends to be 0 in probability.

As for case (ii), when the GNN uses mean readout to classify more than one node graphs, assuming the graph label is determined by the node label and all of the nodes have the same label that are determined as Definition D.1, then GNN optimized with ERM and IRM will also fail because of the same reasons as case (i).

**Discussions on the failures of previous OOD related solutions.** First of all, for IRM or similar objectives [81, 49, 2, 9] that require environment information or non-trivial data partitions, they can hardly be applied to graphs due to the lack of such information. The reason is that obtaining such information can be expensive due to the abstraction of graphs. Moreover, as proved in Theorem 5.1 of Rosenfeld et al. [80], when there is not sufficient support overlap between training environments and testing environments, the IRM or similar objectives can fail catastrophically when being applied to non-linear regime. The only OOD objective EIIL [23] that does not require environment labels, also rely on similar assumptions on the support overlap. We also empirically verify their failing behaviors in our experiments.

Moreover, since part of explainability works also try to find a subset of the inputs for interpretable prediction robustly against distribution shifts. Here we also provide a discussion for these works. The first work following this line is INVRAT [14], which develops an information-theoretic objective (we re-formulate it to suit with OOD generalization problem on graphs):

$$\min_{g, f_c} \max_{f_s} R(f_c \circ g, Y) + \lambda h(R(f_c \circ g, Y) - R_e(f_s \circ g, Y, E)). \tag{11}$$

However, it also requires extra environment labels for optimization that are often unavailable in graphs. Besides, the corresponding assumption on the data generation for guaranteed performance is essentially PIIF if applied to our case, while it can not provide any theoretical guarantee on FIIF.

We also notice a recent work, DIR [104], as a generalization of INVRAT to graphs while studying FIIF spurious correlations, that proposes an alternative objective which does not require environment label:

$$\min \mathbb{E}_s[R(h, Y | \text{do}(S = s))] + \lambda \text{Var}_s(\{R(h, Y | \text{do}(S = s))\}). \tag{12}$$

However, the theoretical justification established for DIR (Theorem 1 to Corollary 1 in Wu et al. [104]) essentially depends on the quality of the generator $g$ which can be prone to spurious correlations. Thus, DIR can hardly provide any theoretical guarantees when applied to our case, neither for FIIF nor PIIF. In experiments, we empirically find the unstable and relatively high sensitivity of DIR to spurious correlations, which verifies our finding. More details about empirical behaviors of DIR can be found in Appendix G.

In contrast to DIR, GIB [120] that focuses on discovering a informative subgraph for explanation, essentially can provide theoretical guarantees for FIIF spurious correlations. Theoretically, (we copy the discussion in Appendix F here to provide an overview of relationships between GIB and DIR.) Under the FIIF assumption on latent interaction, the independence condition derived from causal model can also be rewritten as $Y \perp\!\!\!\perp S|C$ (similar to that in DIR [104] as they also focus on FIIF), which further implies $Y \perp\!\!\!\perp S|\widehat{G}_c$. Hence it is natural to use Information Bottleneck (IB) objective [95] to solve for $G_c$:

$$\min_{f_c, g} R_{G_c}(f_c(\widehat{G}_c)),$$
$$\text{s.t. } G_c = \arg\max_{\widehat{G}_c = g(G) \subseteq G} I(\widehat{G}_c, Y) - I(\widehat{G}_c, \mathcal{G}), \tag{13}$$

which explains the success of many existing works in finding predictive subgraph through IB [120]. However, the estimation of $I(\widehat{G}_c, G)$ is notoriously difficult due to the complexity of graph, which can lead to unstable convergence as observed in our experiments. In contrast, optimization with contrastive objective in CIGA as Eq. 5 induces more stable convergence.

### D.5 Challenges of OOD generalization on graphs.

From the aforementioned analysis, we can summarize some key challenges revealed by the failures of both existing optimization objectives and GNN architectures. In particular, we are facing two main challenges a) Distribution shifts on graphs are more complicated where different types of spurious correlations can be entangled via different graph properties; b) Environment labels are usually not available due to the abstract graph data structure.

## E    Theory and Discussions

In this section, we provide proofs for propositions and theorems mentioned in the main paper.

### E.1 More discussions on Definition 2.5 for Invariant GNNs

Definition 2.5 is motivated by applying the invariance principle to the established SCMs in Sec. 2.2, following the literature of invariant learning [76]. In this section, we will present Proposition E.2 and Proposition E.3 to illustrate how satisfying the minmax objective in Definition E.1 is equivalent to identifying the underlying invariant subgraph $G_c$ that contains all of the information about causal factor $C$ in $G$, under both FIIF and PIIF SCMs (Fig. 2(b) and Fig. 2(c)).

**Definition E.1** (Invariant GNN). Given a set of graph datasets $\{\mathcal{D}^e\}_e$ and environments $\mathcal{E}_{\text{all}}$ that follow the same graph generation process in Sec. 2.2, considering a GNN $\rho \circ h$ that has a permutation invariant graph encoder $h : \mathcal{G} \to \mathbb{R}^h$ and a downstream classifier $\rho : \mathbb{R}^h \to \mathcal{Y}$, $\rho \circ h$ is an invariant GNN if it minimizes the worst case risk among all environments, i.e., $\min \max_{e \in \mathcal{E}_{\text{all}}} R^e$.

First, we show that using the invariant subgraphs $G_c$ to predict $Y$ can satisfy the minmax objective $\min \max_{e \in \mathcal{E}_{\text{all}}} R^e$ in Proposition E.2.

**Proposition E.2.** *Let $\mathcal{G}_c$ denote the subgraph space for $G_c$, given a set of graphs with their labels $\mathcal{D} = \{G^{(i)}, y^{(i)}\}_{i=1}^N$ and $\mathcal{E}_{all}$ that follow the graph generation process in Sec. 2.2 (or Sec. C), a GNN $\rho \circ h : \mathcal{G}_c \to \mathcal{Y}$ that takes $G_c$ of $G$ as the input to predict $Y$, and solves the following objective can generalize to OOD graphs, i.e., solving the minmax objective in Def. E.1:*

$$\min_{\theta} R_{\mathcal{G}_c}(\rho \circ h),$$

*where $R_{\mathcal{G}_c}$ is the empirical risk over $\{G_c^{(i)}, y^{(i)}\}_{i=1}^N$ and $G_c^{(i)}$ is the underlying invariant subgraph $G_c$ for $G^{(i)}$.*

*Proof.* We establish the proof with independent causal mechanism (ICM) assumption in SCM [74, 77]. In particular, given the data generation assumption, i.e., for both FIIF (Assumption 2.2) and PIIF (Assumption 2.3), we have: $\forall e$,

$$
\begin{aligned}
P(Y|C) &= P(Y|C, E = e) \\
P(Y|G_c) \sum_{G_c} P(G_c|C) &= P(Y|G_c) \sum_{G_c} P(G_c|C, E = e) \\
P(Y|G_c) \sum_{G_c} P(G_c|C) &= P(Y|G_c, E = e) \sum_{G_c} P(G_c|C) \\
P(Y|G_c) &= P(Y|G_c, E = e),
\end{aligned}
\tag{14}
$$

where we use ICM for the first three equalities. From Eq. 14, it suffices to know $P(Y|G_c)$ is invariant across different environments. Hence, a GNN predictor $\rho \circ h : \mathcal{G}_c \to \mathcal{Y}$ optimized with empirical risk given $G_c$, essentially minimizes the empirical risk across all environments, i.e., $\min R_{\mathcal{G}_c} = \min \max R^e$. Thus, if $\rho \circ h$ solves $\min R_{\mathcal{G}_c}$, it also solves $\min \max R^e$, hence it elicits a invariant GNN predictor according to Definition. E.1. □

Besides, we show in Proposition E.3 that only using the underlying invariant subgraphs $G_c$ to make predictions can satisfy the minmax objectives. Or equivalently, a GNN predictor solving the minmax objective can only rely on the underlying invariant subgraph $G_c$ to predict $Y$.

**Proposition E.3.** *Given a set of graph datasets $\{\mathcal{D}^e\}_e$ and environments $\mathcal{E}_{all}$ that follow the same graph generation process in Sec. 2.2, considering a GNN $\rho \circ h$ that has a permutation invariant graph encoder $h : \mathcal{G} \to \mathbb{R}^h$ and a downstream classifier $\rho : \mathbb{R}^h \to \mathcal{Y}$, $\rho \circ h$ that minimizes the worst case risk among all environments, i.e., $\min \max_{e \in \mathcal{E}_{all}} R^e$, can not rely on any part of $G_s$, i.e., $\rho \circ h(G) \perp\!\!\!\perp G_s$.*

*Proof.* The proof for Proposition E.3 is straightforward. Assuming that $\rho \circ h(G) \not\perp\!\!\!\perp G_s$, as $E$ is influenced by the changes of $E$ through $S$ in both FIIF and PIIF SCMs (Fig. 2(b) and Fig. 2(c)), then $\rho \circ h(G) \not\perp\!\!\!\perp E$ as well. Consequently, there exists some graph $G$ corresponding to $G_c, G_s^e$ and $\rho \circ h(G) = Y$ under an environment $e$, such that we can always find a proper $e'$ to make $\rho \circ h(G) \neq Y$. In contrast, the prediction of a GNN that satisfies $\rho \circ h(G) \perp\!\!\!\perp G_s$ remains invariant against arbitrary changes of environments. Thus, it leads to a contradiction to the condition that $\min \max_{e' \in \mathcal{E}_{\mathrm{all}}} R^{e'}$. Therefore, a GNN that solves $\min \max_{e \in \mathcal{E}_{\mathrm{all}}} R^e$ must satisfy $\rho \circ h(G) \perp\!\!\!\perp G_s$. $\qquad\square$

Combining Proposition E.2 and Proposition E.3, we are highly motivated to find the underlying invariant subgraphs to make predictions about the original graphs, which converges to Eq. 1. Tackling Eq. 1 under the unavailability of $E$ brings us two variants of CIGA solutions, as illustrated in Section 3.

### E.2   Proof for theorem 3.1 (i)

**Theorem E.4** (CIGAv1 Induces Invariant GNNs). *Given a set of graph datasets $\{\mathcal{D}^e\}_e$ and environments $\mathcal{E}_{\mathrm{all}}$ that follow the same graph generation process in Sec. 2.2, assuming that* (a) $f_{gen}^G$ *and* $f_{gen}^{G_c}$ *in Assumption 2.1 are invertible,* (b) *samples from each training environment are equally distributed, i.e.,* $|\mathcal{D}_{\hat{e}}| = |\mathcal{D}_{\tilde{e}}|$, $\forall \hat{e}, \tilde{e} \in \mathcal{E}_{tr}$, if $\forall G_c, |G_c| = s_c$, *then a GNN* $f_c \circ g$ *solves Eq. 4, is an invariant GNN (Def. 2.5).*

*Proof.* We re-write the objective as follows:

$$\max_{f_c, g} \; I(\widehat{G}_c; Y), \text{ s.t. } \widehat{G}_c \in \underset{\widehat{G}_c = g(G), |\widehat{G}_c| \leq s_c}{\arg\max} \; I(\widehat{G}_c; \widetilde{G}_c | Y), \tag{15}$$

where $\widehat{G}_c = g(G)$, $\widetilde{G}_c = g(\widetilde{G})$ and $\widetilde{G} \sim \mathbb{P}(G|Y)$, i.e., $\widetilde{G}$ and $G$ have the same label.

The proof of Theorem E.4 is essentially to show the estimated $\widehat{G}_c$ through Eq. 15 is the underlying $G_c$, then the maximizer of $I(\widehat{G}_c; Y)$ in Eq. 15 can produce most informative and stable predictions about $Y$ based on $G$, hence is an invariant GNN (Definition. E.1).

In the next, we are going to take an information-theoretic view of the first term $I(\widehat{G}_c; Y)$ and the second term $I(\widehat{G}_c; \widetilde{G}_c | Y)$ to conclude the proof. We begin by introducing the following lemma:

**Lemma E.5.** *Given the same conditions as Thm. E.4, $I(\widehat{G}_c; Y)$ is maximized if and only if $I(\widehat{G}_c; Y | E = e)$ is maximized, $\forall e \in \mathcal{E}_{tr}$.*

The proof for Lemma E.5 is straightforward, given the condition that samples from each training environment are equally distributed, i.e., $|\mathcal{D}_{\hat{e}}| = |\mathcal{D}_{\tilde{e}}|$, $\forall \hat{e}, \tilde{e} \in \mathcal{E}_{tr}$. Obviously, $\widehat{G}_c = G_c$ is a maximizer of $I(\widehat{G}_c; Y) = I(C; Y) = H(Y)$, since $f_{gen}^c : \mathcal{C} \to \mathcal{G}_c$ is invertible and $C$ causes $Y$. However, there might be some subset $G_s^p \subseteq G_s$ from the underlying $G_s$ that entail the same information about label, i.e., $I(G_c^p \cup G_s^p; Y) = I(G_c; Y)$ where $\widehat{G}_c = G_c^p \cup G_s^p$ and $G_c^p = G_c \cap \widehat{G}_c$. For FIIF (Assumption 6(b)), it can not happen, otherwise, let $G_c^l = G_c - G_c^p$, then we have:

$$\begin{aligned}
I(\widehat{G}_c; Y) = I(G_c^p \cup G_s^p; Y) &= I(G_c^p \cup G_c^l; Y) = I(G_c; Y) \\
I(G_c^p; Y) + I(G_s^p; Y | G_c^p) &= I(G_c^p; Y) + I(G_c^l; Y | G_c^p) \\
I(G_s^p; Y | G_c^p) &= I(G_c^l; Y | G_c^p) \\
H(Y | G_c^p) - H(Y | G_c^p, G_s^p) &= H(Y | G_c^p) - H(Y | G_c^p, G_c^l) \\
H(Y | G_c^p) - H(Y | G_c^p, G_s^p) &= H(Y | G_c^p), \\
H(Y | G_c^l, G_s^p) &= 0,
\end{aligned} \tag{16}$$

where the second last equality is due to $C \to Y$ and the invertibility of $f_{gen}^c : \mathcal{C} \to \mathcal{G}_c$ in FIIF, i.e., $H(Y|C) = H(Y|G_c) = H(Y|G_c^p, G_c^l) = 0$. However, in PIIF, it can hold since conditioning on $G_c^p, G_s^p$ can not determine $Y$, as $S \not\perp\!\!\!\perp Y | C$. In other words, $G_s \not\perp\!\!\!\perp Y | G_c$, which means $G_s$ can imply some information about $Y$ that is equivalent to $I(G_c^l; Y | G_c^p)$.

To avoid the presence of spuriously correlated $G_s$ in $\widehat{G}_c$, we will use the second term to eliminate it:

$$\max_{f_c, g} I(\widehat{G}_c; \widetilde{G}_c | Y),$$

$$= H(\widehat{G}_c | Y) - H(\widehat{G}_c | \widetilde{G}_c, Y), \tag{17}$$

where $\widehat{G}_c = g(G)$, $\widetilde{G}_c = g(\widetilde{G})$ are two positive samples drawn from the same class (i.e., condition on the same $Y$). Since the all of the training environments are equally distributed, maximizing $I(\widehat{G}_c; \widetilde{G}_c | Y)$ is essentially maximizing $I(\widehat{G}_c, E = \hat{e}; \widetilde{G}_c, E = \tilde{e} | Y)$, $\forall \hat{e}, \tilde{e} \in \mathcal{E}_{\text{tr}}$. Hence, we have:

$$\max_{f_c, g} I(\widehat{G}_c; \widetilde{G}_c | Y),$$

$$= I(\widehat{G}_c, E = \hat{e}; \widetilde{G}_c, E = \tilde{e} | Y) \tag{18}$$

$$= H(\widehat{G}_c, E = \hat{e} | Y) - H(\widehat{G}_c, E = \hat{e} | \widetilde{G}_c, E = \tilde{e}, Y).$$

We claim Eq. 18 can eliminate any potential subsets from $G_s$ in the estimated $\widehat{G}_c$.

Otherwise, suppose there are some subsets $\widehat{G}_s^p \subseteq \widehat{G}_s$ and $\widetilde{G}_s^p \subseteq \widetilde{G}_s$ contained in the estimated $\widehat{G}_c, \widetilde{G}_c$, where $\widehat{G}_s, \widetilde{G}_s$ be the corresponding underlying $G_s$s for $\widehat{G}_c, \widetilde{G}_c$. Let $\widehat{G}_c^*$ and $\widetilde{G}_c^*$ be the ground truth invariant subgraph $G_c$s of $\widehat{G}$ and $\widetilde{G}$, $\widehat{G}_c^l = \widehat{G}_c^* - \widehat{G}_c$ and $\widetilde{G}_c^l = \widetilde{G}_c^* - \widetilde{G}_c$ be the **l**eft (un-estimated) subsets from corresponding ground truth $G_c$s, and $\widehat{G}_c^p = \widehat{G}_c^* - \widehat{G}_c^l$ and $\widetilde{G}_c^p = \widetilde{G}_c^* - \widetilde{G}_c^l$ be the complement, or equivalently, the **p**artial $\widehat{G}_c^*, \widetilde{G}_c^*$ that are estimated in $\widehat{G}_c, \widetilde{G}_c$, respectively. We can also define similar counterparts for $G_s$: $\widehat{G}_s^p, \widetilde{G}_s^p$ are the partial $\widehat{G}_s, \widetilde{G}_s$s contained in the estimated $\widehat{G}_c, \widetilde{G}_c$ while $\widehat{G}_s^l, \widetilde{G}_s^l$ are the left subsets $\widehat{G}_s, \widetilde{G}_s$, respectively.

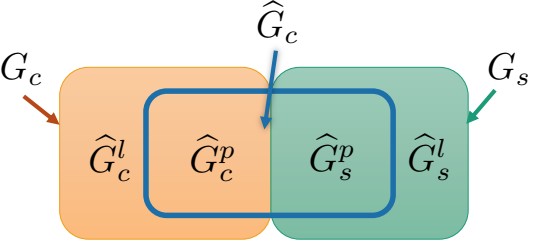

Figure 13: Illustration of the notation. $G_c$ and $G_s$ are two disjoint sets. $\widehat{G}_c$ may contain certain subsets from $G_c$ and $G_s$. The subsets from $G_c$ and $G_s$ contained in $\widehat{G}_c$ are denoted as $\widehat{G}_c^p$ and $\widehat{G}_s^p$, respectively. While the left subsets in $G_c$ and $G_s$ are denoted as $\widehat{G}_c^l$ and $\widehat{G}_s^l$, respectively.

Recall the constraint that $|G_c| = s_c$, hence if $\widehat{G}_s^p \subseteq \widehat{G}_c$, then a corresponding $\widehat{G}_c^l = \widehat{G}_c^* - \widehat{G}_c^p$ will be replaced by $\widehat{G}_s^p$ in $\widehat{G}_c$. In this case, we have:

$$H(\widehat{G}_c, E = \hat{e} | Y) = H(E = \hat{e} | \widehat{G}_c, Y) + H(\widehat{G}_c | E = \hat{e}, Y)$$

$$= H(\widehat{G}_c^p \cup \widehat{G}_s^p | E = \hat{e}, Y) \tag{19}$$

$$= H(\widehat{G}_c^p | E = \hat{e}, Y) + H(\widehat{G}_s^p | \widehat{G}_c^p, E = \hat{e}, Y)$$

where the second equality is due to $E = \hat{e}$ is determined so that $H(E = \hat{e} | \widehat{G}_c, Y) = 0$. Compared Eq. 19 to that when $\widehat{G}_c = \widehat{G}_c^*$, we have the entropy change as:

$$\Delta H(\widehat{G}_c, E = \hat{e} | Y) = H(\widehat{G}_c, E = \hat{e} | Y) - H(\widehat{G}_c^*, E = \hat{e} | Y),$$

$$= H(\widehat{G}_s^p | \widehat{G}_c^p, E = \hat{e}, Y) - H(\widehat{G}_c^l | \widehat{G}_c^p, E = \hat{e}, Y). \tag{20}$$

Let $\epsilon = H(\widehat{G}_s^p | \widehat{G}_c^p, E = \hat{e}, Y)$. In a idealistic setting, when the noise of the generation process $S := f_{\text{spu}}(Y, E)$ in PIIF tends to be 0, i.e., $\epsilon \to 0$, $S$ is determined conditioned on $E, Y$, hence $G_s$ and any subsets of $G_s$ are all determined. Then, it suffices to know that in Eq. 20, $H(\widehat{G}_s^p | \widehat{G}_c^p, E = \hat{e}, Y) = 0$ while $H(\widehat{G}_c^l | \widehat{G}_c^p, E = \hat{e}, Y) > 0$ since $\widehat{G}_c^l$ can not be determined when given $\widehat{G}_c^p, E = \hat{e}, Y$. Thus, when some subset from $G_s$ is included in $\widehat{G}_c$, it will minimize $H(\widehat{G}_c, E = \hat{e} | Y)$.

However in practice, it is usual that $\epsilon > 0$. Therefore, in the next, we will show how $\epsilon = H(\widehat{G}_s^p | \widehat{G}_c^p, E = \hat{e}, Y)$ can be cancelled thus leading to a smaller $H(\widehat{G}_c, E = \hat{e} | Y)$, by considering the second term $H(\widehat{G}_c, E = \hat{e} | \widetilde{G}_c, E = \tilde{e}, Y)$.

As for $H(\widehat{G}_c, E = \hat{e}|\widetilde{G}_c, E = \tilde{e}, Y)$, without loss of generality, we can divide all of the possible cases into two:

(i) One of $\widehat{G}_c$ and $\widetilde{G}_c$ contains some subset of $G_s$, i.e., $\widehat{G}_c$ contains some $\widehat{G}_s^p \subseteq \widehat{G}_s$;

(ii) Both $\widehat{G}_c$ and $\widetilde{G}_c$ contain some $\widehat{G}_s^p \subseteq \widehat{G}_s$ and $\widetilde{G}_s^p \subseteq \widetilde{G}_s$, respectively.

For (i), we have:

$$
\begin{aligned}
H(\widehat{G}_c, E = \hat{e}|\widetilde{G}_c, E = \tilde{e}, Y) &= H(\widehat{G}_c^p, \widehat{G}_s^p, E = \hat{e}|\widetilde{G}_c, E = \tilde{e}, Y) \\
&= H(\widehat{G}_s^p|\widetilde{G}_c, E = \tilde{e}, Y, \widehat{G}_c^p, E = \hat{e}) + H(\widehat{G}_c^p, E = \hat{e}|\widetilde{G}_c, E = \tilde{e}, Y),
\end{aligned}
\tag{21}
$$

Thus, we can write the change of $H(\widehat{G}_c, E = \hat{e}|\widetilde{G}_c, E = \tilde{e}, Y)$ between $\widehat{G}_c = \widehat{G}_c^p \cup \widehat{G}_s^p$ and $\widehat{G}_c = \widehat{G}_c^*$ as:

$$
\begin{aligned}
\Delta H(\widehat{G}_c, E = \hat{e}|\widetilde{G}_c, E = \tilde{e}, Y) &= H(\widehat{G}_c, E = \hat{e}|\widetilde{G}_c, E = \tilde{e}, Y) - H(\widehat{G}_c^*, E = \hat{e}|\widetilde{G}_c, E = \tilde{e}, Y), \\
&= H(\widehat{G}_s^p|\widetilde{G}_c, E = \tilde{e}, Y, \widehat{G}_c^p, E = \hat{e}) \\
&\quad - H(\widehat{G}_c^l|\widetilde{G}_c, E = \tilde{e}, Y, \widehat{G}_c^p, E = \hat{e}).
\end{aligned}
\tag{22}
$$

Combing $\Delta H(\widehat{G}_c, E = \hat{e}|Y)$, we have:

$$
\begin{aligned}
\Delta I(\widehat{G}_c, E = \hat{e}; \widetilde{G}_c, E = \tilde{e}|Y) &= \Delta H(\widehat{G}_c, E = \hat{e}|Y) - \Delta H(\widehat{G}_c, E = \hat{e}|\widetilde{G}_c, E = \tilde{e}, Y) \\
&= \left\{ H(\widehat{G}_s^p|\widehat{G}_c^p, E = \hat{e}, Y) - H(\widehat{G}_s^p|\widetilde{G}_c, E = \tilde{e}, Y, \widehat{G}_c^p, E = \hat{e}) \right\} \\
&\quad + \left\{ -H(\widehat{G}_c^l|\widehat{G}_c^p, E = \hat{e}, Y) + H(\widehat{G}_c^l|\widetilde{G}_c, E = \tilde{e}, Y, \widehat{G}_c^p, E = \hat{e}) \right\}, \\
&= -H(\widehat{G}_c^l|\widehat{G}_c^p, E = \hat{e}, Y) + H(\widehat{G}_c^l|\widetilde{G}_c, E = \tilde{e}, Y, \widehat{G}_c^p, E = \hat{e}),
\end{aligned}
\tag{23}
$$

where the last equality is because of the independence of $\widehat{G}_s^p$ between $\widetilde{G}_c, E = \tilde{e}$ conditioned on $Y, E = \hat{e}$. Since conditioning will lower the entropy for both discrete and continuous variables [22, 114], we have:

$$
\Delta I(\widehat{G}_c, E = \hat{e}; \widetilde{G}_c, E = \tilde{e}|Y) < 0,
\tag{24}
$$

which implies the existence of $\widehat{G}_s^p$ in $\widehat{G}_c$ will lower down the second term in Eq. 15 for the case (i).

For (ii), we have:

$$
\begin{aligned}
H(\widehat{G}_c, E = \hat{e}|\widetilde{G}_c, E = \tilde{e}, Y) &= H(\widehat{G}_c^p, \widehat{G}_s^p, E = \hat{e}|\widetilde{G}_c^p, \widetilde{G}_s^p, E = \tilde{e}, Y) \\
&= H(\widehat{G}_s^p|\widetilde{G}_c^p, \widetilde{G}_s^p, E = \tilde{e}, Y, \widehat{G}_c^p, E = \hat{e}) \\
&\quad + H(\widehat{G}_c^p, E = \hat{e}|\widetilde{G}_c^p, \widetilde{G}_s^p, E = \tilde{e}, Y),
\end{aligned}
\tag{25}
$$

Similar to (i), $H(\widehat{G}_s^p|\widetilde{G}_c^p, \widetilde{G}_s^p, E = \tilde{e}, Y, \widehat{G}_c^p, E = \hat{e})$ can be cancelled out with $H(\widehat{G}_s^p|\widehat{G}_c^p, E = \hat{e}, Y)$. Then, we have:

$$
\begin{aligned}
\Delta I(\widehat{G}_c, E = \hat{e}; \widetilde{G}_c, E = \tilde{e}|Y) &= \Delta H(\widehat{G}_c, E = \hat{e}|Y) - \Delta H(\widehat{G}_c, E = \hat{e}|\widetilde{G}_c, E = \tilde{e}, Y) \\
&= -H(\widehat{G}_c^l|\widehat{G}_c^p, E = \hat{e}, Y) + H(\widehat{G}_c^l|\widetilde{G}_c^p, \widetilde{G}_s^p, E = \tilde{e}, \widehat{G}_c^p, Y, E = \hat{e}).
\end{aligned}
\tag{26}
$$

Since additionally conditioning on $\widehat{G}_s^p$ in $H(\widehat{G}_c^l, E = \hat{e}|\widetilde{G}_c^p, \widetilde{G}_s^p, E = \tilde{e}, Y)$ can not lead to new information about $\widehat{G}_c^l$, we have:

$$
\begin{aligned}
H(\widehat{G}_c^l|\widetilde{G}_c^p, \widetilde{G}_s^p, E = \tilde{e}, \widehat{G}_c^p, Y, E = \hat{e}) &= H(\widehat{G}_c^l|\widetilde{G}_c^p, E = \tilde{e}, \widehat{G}_c^p, Y, E = \hat{e}) \\
&< H(\widehat{G}_c^l|\widehat{G}_c^p, Y, E = \hat{e}),
\end{aligned}
\tag{27}
$$

which follows that $\Delta I(\widehat{G}_c, E = \hat{e}; \widetilde{G}_c, E = \tilde{e}|Y) < 0$.

To summarize, the ground truth $G_c$ is the only maximizer of the objective (Eq. 15), hence solving for the objective (Eq. 15) can elicit an invariant GNN.

### E.3 Proof for theorem 3.1 (ii)

**Theorem E.6** (CIGAv2 Induces Invariant GNNs). *Given a set of graph datasets $\{\mathcal{D}^e\}_e$ and environments $\mathcal{E}_{all}$ that follow the same graph generation process in Sec. 2.2, assuming that* (a) $f_{gen}^G$ *and* $f_{gen}^{G_c}$ *in Assumption 2.1 are invertible,* (b) *samples from each training environment are equally distributed, i.e.,$|\mathcal{D}_{\hat{e}}| = |\mathcal{D}_{\tilde{e}}|$, $\forall \hat{e}, \tilde{e} \in \mathcal{E}_{tr}$, a GNN $f_c \circ g$ solves Eq. 4, is an invariant GNN (Def. 2.5).*

*Proof.* We re-write the objective as follows:

$$\max_{f_c, g} I(\widehat{G}_c; Y) + I(\widehat{G}_s; Y), \text{ s.t. } \widehat{G}_c \in \underset{\widehat{G}_c = g(G), \widetilde{G}_c = g(\widetilde{G})}{\arg\max} I(\widehat{G}_c; \widetilde{G}_c | Y),$$

$$I(\widehat{G}_s; Y) \leq I(\widehat{G}_c; Y), \ \widehat{G}_s = G - g(G). \tag{28}$$

where $\widehat{G}_c = g(G), \widetilde{G}_c = g(\widetilde{G})$ and $\widetilde{G} \sim \mathbb{P}(G|Y)$, i.e., $\widetilde{G}$ and $G$ have the same label.

Similar to the proof for Theorem E.4, to prove Theorem E.6 is essentially to show the estimated $\widehat{G}_c$ through Eq. 28 is the underlying $G_c$, hence the minimizer of Eq. 28 elicits an invariant GNN predictor (Definition. E.1).

In the next, we also begin with a lemma:

**Lemma E.7.** *Given data generation process as Theorem E.6, for both FIIF and PIIF, we have:*

$$I(C; Y) \geq I(S; Y),$$

*hence $I(G_c; Y) \geq I(G_s; Y)$.*

*Proof for Lemma E.7.* For both FIIF and PIIF, Assumption 2.4 implies that $H(C|Y) \leq H(S|Y)$. It follows that $I(C; Y) = H(Y) - H(C|Y) \geq H(Y) - H(S|Y) = I(S; Y)$. Then, since $f_{gen}^{G_c} : \mathcal{C} \to \mathcal{G}_c$ is invertible, we have $I(G_c; Y) = I(C; Y) \geq I(S; Y) \geq I(G_s; Y)$. $\square$

Given Lemma E.7, we know $\widehat{G}_c$ at least contains some subset of the underlying $G_c$, otherwise the constraint $I(\widehat{G}_s; Y) \leq I(\widehat{G}_c; Y)$ will be violated since $G_c \subseteq \widehat{G}_s$ in this case.

Assuming there are some subset of $G_s$ contained in $\widehat{G}_c$, without loss of generality, we can divide all of the possible cases about $\widehat{G}_c$ into two:

   (i) $\widehat{G}_c$ only contains a subset of the underlying $G_c$;

   (ii) $\widehat{G}_c$ contains a subset of the underlying $G_c$ as well as part of the underlying $G_s$;

Before the discussion, let us inherit the notations of subsets of $G_c, G_s$ from the proof for Theorem E.4: Let $\widehat{G}_c^*$ and $\widetilde{G}_c^*$ be the ground truth invariant subgraph $G_c$s of $\widehat{G}$ and $\widetilde{G}$, $\widehat{G}_c^l = \widehat{G}_c^* - \widehat{G}_c$ and $\widetilde{G}_c^l = \widetilde{G}_c^* - \widetilde{G}_c$ be the **l**eft (un-estimated) subsets from corresponding ground truth $G_c$s, and $\widehat{G}_c^p = \widehat{G}_c^* - \widehat{G}_c^l$ and $\widetilde{G}_c^p = \widetilde{G}_c^* - \widetilde{G}_c^l$ be the complement, or equivalently, the **p**artial $\widehat{G}_c^*, \widetilde{G}_c^*$ that are estimated in $\widehat{G}_c, \widetilde{G}_c$, respectively. Similarly, $\widehat{G}_s^p, \widetilde{G}_s^p$ are the partial $\widehat{G}_s, \widetilde{G}_s$s contained in the estimated $\widehat{G}_c, \widetilde{G}_c$ while $\widehat{G}_s^l, \widetilde{G}_s^l$ are the left subsets $\widehat{G}_s, \widetilde{G}_s$, respectively.

First of all, case (i) cannot hold because, when maximizing $I(\widehat{G}_c; \widetilde{G}_c | Y)$, if $\exists \widehat{G}_c^l = \widehat{G}_c^* - \widehat{G}_c$, as shown in the proof for Theorem E.4, including $\widehat{G}_c^l$ into $\widehat{G}_c$ can always enlarge $I(\widehat{G}_c; \widetilde{G}_c | Y)$, while not affecting the optimality of $I(\widehat{G}_s; Y) +$

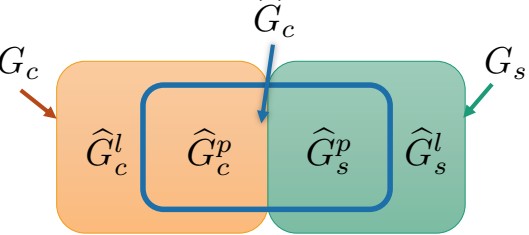

Figure 14: Illustration of the notation for estimated $\widehat{G}_c$ from $G$. $G_c$ and $G_s$ are two disjoint sets. $\widehat{G}_c$ may contain certain subsets from $G_c$ and $G_s$. The subsets from $G_c$ and $G_s$ contained in $\widehat{G}_c$ are denoted as $\widehat{G}_c^p$ and $\widehat{G}_s^p$, respectively. While the left subsets in $G_c$ and $G_s$ are denoted as $\widehat{G}_c^l$ and $\widehat{G}_s^l$, respectively. Similar notations are also applicable for the estimated $\widetilde{G}_c$ from $\widetilde{G}$.

$I(\widehat{G}_c; Y)$ by re-distributing $\widehat{G}_c^l$ from $\widehat{G}_s$ to $\widehat{G}_c$. Consequently, $\widehat{G}_c^*$ must be included in $\widehat{G}_c$, i.e., $\widehat{G}_c^* \subseteq \widehat{G}_c$.

As for case (ii), recall that, by the condition of equally distributed training samples from each training environment, maximizing $I(\widehat{G}_c; \widetilde{G}_c|Y)$ is essentially maximizing $I(\widehat{G}_c, E = \hat{e}; \widetilde{G}_c, E = \tilde{e}|Y)$, $\forall \hat{e}, \tilde{e} \in \mathcal{E}_{\text{tr}}$, hence, we have:

$$\max_{g, f_c} I(\widehat{G}_c; \widetilde{G}_c|Y),$$
$$= I(\widehat{G}_c, E = \hat{e}; \widetilde{G}_c, E = \tilde{e}|Y) \tag{29}$$
$$= H(\widehat{G}_c, E = \hat{e}|Y) - H(\widehat{G}_c, E = \hat{e}|\widetilde{G}_c, E = \tilde{e}, Y).$$

We claim Eq. 29 can eliminate any potential subsets in the estimated $\widehat{G}_c$. Similarly, we have:

$$\begin{aligned} H(\widehat{G}_c, E = \hat{e}|Y) &= H(E = \hat{e}|\widehat{G}_c, Y) + H(\widehat{G}_c|E = \hat{e}, Y) \\ &= H(\widehat{G}_c^* \cup \widehat{G}_s^p|E = \hat{e}, Y) \\ &= H(\widehat{G}_c^*|E = \hat{e}, Y) + H(\widehat{G}_s^p|\widehat{G}_c^*, E = \hat{e}, Y) \\ &= H(\widehat{G}_c^*|Y) + H(\widehat{G}_s^p|\widehat{G}_c^*, E = \hat{e}, Y) \end{aligned} \tag{30}$$

where the second equality is due to $E = \hat{e}$ is determined. Compared to the case that $\widehat{G}_c = \widehat{G}_c^*$, we have:

$$\begin{aligned} \Delta H(\widehat{G}_c, E = \hat{e}|Y) &= H(\widehat{G}_c, E = \hat{e}|Y) - H(\widehat{G}_c^*, E = \hat{e}|Y), \\ &= H(\widehat{G}_s^p|\widehat{G}_c^*, E = \hat{e}, Y). \end{aligned} \tag{31}$$

Then, as for $H(\widehat{G}_c, E = \hat{e}|\widetilde{G}_c, E = \tilde{e}, Y)$, without loss of generality, we can divide all of the possible cases into two:

(a) $\widehat{G}_c$ contains some $\widehat{G}_s^p \subseteq \widehat{G}_s$;

(b) Both $\widehat{G}_c$ and $\widetilde{G}_c$ contain some $\widehat{G}_s^p \subseteq \widehat{G}_s$ and $\widetilde{G}_s^p \subseteq \widetilde{G}_s$, respectively.

For (a), we have:

$$\begin{aligned} H(\widehat{G}_c, E = \hat{e}|\widetilde{G}_c, E = \tilde{e}, Y) &= H(\widehat{G}_c^*, \widehat{G}_s^p, E = \hat{e}|\widetilde{G}_c, E = \tilde{e}, Y) \\ &= H(\widehat{G}_s^p|\widetilde{G}_c, E = \tilde{e}, Y, \widehat{G}_c^*, E = \hat{e}) + H(\widehat{G}_c^*, E = \hat{e}|\widetilde{G}_c, E = \tilde{e}, Y), \end{aligned} \tag{32}$$

Similarly to the proof for Theorem E.4, when considering $\Delta I(\widehat{G}_c; \widetilde{G}_c|Y)$, the effects of $H(\widehat{G}_s^p|\widetilde{G}_c, E = \tilde{e}, Y, \widehat{G}_c^*, E = \hat{e})$ is cancelled out by $H(\widehat{G}_s^p|\widehat{G}_c^*, E = \hat{e}, Y)$. Hence, we have:

$$\Delta I(\widehat{G}_c; \widetilde{G}_c|Y) = 0.$$

For (b), we have:

$$\begin{aligned} H(\widehat{G}_c, E = \hat{e}|\widetilde{G}_c, E = \tilde{e}, Y) &= H(\widetilde{G}_c^*, \widetilde{G}_s^p, E = \hat{e}|\widetilde{G}_c^*, \widetilde{G}_s^p, E = \tilde{e}, Y) \\ &= H(\widehat{G}_s^p|\widetilde{G}_c^*, \widetilde{G}_s^p, E = \tilde{e}, Y, \widehat{G}_c^*, E = \hat{e}) \\ &\quad + H(\widehat{G}_c^*|\widetilde{G}_c^*, \widetilde{G}_s^p, E = \tilde{e}, Y, E = \hat{e}), \end{aligned} \tag{33}$$

Similarly, $H(\widehat{G}_s^p|\widetilde{G}_c^*, \widetilde{G}_s^p, E = \tilde{e}, Y, \widehat{G}_c^*, E = \hat{e}) = 0$ can also be cancelled out by $H(\widehat{G}_s^p|\widehat{G}_c^*, E = \hat{e}, Y)$. Moreover, for $H(\widehat{G}_c^*|\widetilde{G}_c^*, \widetilde{G}_s^p, E = \tilde{e}, Y, E = \hat{e})$, $\widetilde{G}_s^p$ can not bring no additional information about $\widehat{G}_c^*$, when conditioning on $\widetilde{G}_c^*, Y, E = \tilde{e}$. Hence, we also have:

$$\Delta I(\widehat{G}_c; \widetilde{G}_c|Y) = 0.$$

To summarize, when maximizing $I(\widehat{G}_c; \widetilde{G}_c|Y)$, including any $\widehat{G}_s^p \subseteq \widehat{G}_s^*$ can not bring additional benefit while affecting the optimality of $I(\widehat{G}_s; Y) + I(\widehat{G}_c; Y)$. More specifically, when considering the changes to $I(\widehat{G}_s; Y) + I(\widehat{G}_c; Y)$, $\forall G_s^p \subseteq G_s$, we have

$$I(G - \widehat{G}_c^* - G_s^p; Y) \le I(G - \widehat{G}_c^*; Y), \ \forall G_s^p \subseteq G_s,$$

while $I(Y; \widehat{G}_c^*, G_s^p) = I(Y; \widehat{G}_c^*) + I(Y; \widehat{G}_s^p | \widehat{G}_c^*)$, $\forall e \in \mathcal{E}_{\text{tr}}$. Consequently,

$$
\begin{aligned}
\Delta I(\widehat{G}_s; Y) + I(\widehat{G}_c; Y) &= -I(\widehat{G}_s^p; Y | \widehat{G}_s^l) + I(\widehat{G}_s^p; Y | \widehat{G}_c^*) \\
&= -I(\widehat{G}_s^p; Y) + I(\widehat{G}_s^p; Y | \widehat{G}_c^*) \le 0.
\end{aligned}
\tag{34}
$$

Hence, only the underlying $G_c$ is the solution to Eq. 28, which implies that solving for the objective (Eq. 28) can elicit an invariant GNN.

## F Details of Prototypical CIGA Implementation

In fact, the CIGA framework introduced in Sec. 3 can have multiple implementations. We choose interpretable architectures in our experiments for the purpose of concept verification. More sophisticated architectures can be incorporated. Experimental results in Sec. 4 also demonstrates that, even equipped with basic GNN architectures, CIGA already has the excellent OOD generalization ability, hence it is promising to incorporate more advanced architectures from the prosperous GNN literature.

We now introduce the details of the architectures used in our experiments. Recall that CIGA decomposes a GNN model for graph classification into two modules, i.e., a featurizer: $g : \mathcal{G} \to \mathcal{G}_c$ and a classifier $f_c : \mathcal{G}_c \to \mathcal{Y}$. Specifically, for the implementation of Featurizer, we choose one of the common practices GAE [44] for calculating the sampled weights for each edge. More formally, the soft mask is predicted through the following equation:

$$
Z = \text{GNN}(G) \in \mathbb{R}^{n \times h}, \; M = \text{a}(Z, A) \in \mathbb{R}^{n \times n},
$$

where $a$ calculates the sampling weights for each edge using a MLP: $M_{ij} = \text{MLP}([Z_i, Z_j])$.

If a sampling ratio $s_c$ is predetermined, we sample $s_c$ of total edges with the largest predicted weights as a soft estimation of $\widehat{G}_c$. Then, the estimated $\widehat{G}_c$ will be forwarded to the classifier $f_c$ for predicting the labels of the original graph. Although Theorem E.4 assumes $s_c$ is known, in real applications we do not know the specific $s_c$. Hence, in experiments, we select $s_c$ according to the validation performance. To thoroughly study the effects of $I(\widehat{G}_s; Y)$ comparing to CIGAv1, we stick to using the same $s_c$ and sampling process for CIGAv2, while CIGAv2 essentially requires less specific knowledge about ground truth $r_c$ hence achieving better empirical performance. Moreover, once the sampled edges are determined, the classifier GNN can take either the original feature of the input graph or the learned feature from the featurizer as the new node attributes for $\widehat{G}_c$. We select the architecture according to the validation performance from some random runs.

For the implementation of the information theoretic objectives, we will use CIGAv2 for elaboration while the implementation of CIGAv1 can be obtained via removing the third term from CIGAv2. Recall that CIGAv2 has the following formulation:

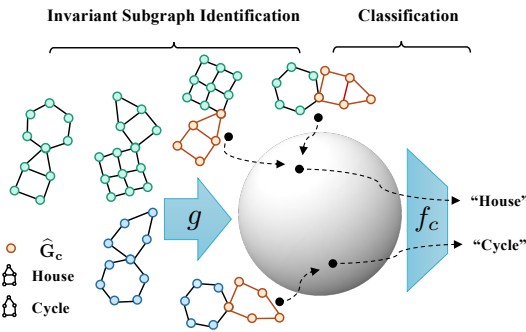

Figure 15: Illustration of **C**ausality **I**nspired Invariant **G**raph Le**A**rning (CIGA): GNNs need to classify graphs based on the specific motif ("House" or "Cycle"). The featurizer $g$ will extract an (orange colored) subgraph $\widehat{G}_c$ from each input for the classifier $f_c$ to predict the label. The training objective of $g$ is implemented in a contrastive strategy where the distribution of $\widehat{G}_c$ at the latent sphere will be optimized to maximize the intra-class mutual information. With the identified invariant subgraph $G_c$, the predictions made by classifier $f_c$ based on $G_c$ are invariant to distribution shifts;

$$
\max_{f_c, g} I(\widehat{G}_c; Y) + I(\widehat{G}_s; Y), \; \text{s.t.} \; \widehat{G}_c \in \underset{\widehat{G}_c = g(G), \widetilde{G}_c = g(\widetilde{G})}{\arg \max} I(\widehat{G}_c; \widetilde{G}_c | Y),
$$
$$
I(\widehat{G}_s; Y) \le I(\widehat{G}_c; Y), \; \widehat{G}_s = G - g(G).
\tag{35}
$$

where $\widehat{G}_c = g(G), \widetilde{G}_c = g(\widetilde{G})$ and $\widetilde{G} \sim P(G|Y)$, i.e., $\widetilde{G}$ and $G$ have the same label. In Sec. 3.3, we introduce a contrastive approximation for $I(\widehat{G}_c; \widetilde{G}_c|Y)$:

$$I(\widehat{G}_c; \widetilde{G}_c|Y) \approx \mathbb{E}_{\substack{\{\widehat{G}_c, \widetilde{G}_c\} \sim \mathbb{P}_g(G|\mathcal{Y}=Y) \\ \{G_c^i\}_{i=1}^M \sim \mathbb{P}_g(G|\mathcal{Y} \neq Y)}} \log \frac{e^{\phi(h_{\widehat{G}_c}, h_{\widetilde{G}_c})}}{e^{\phi(h_{\widehat{G}_c}, h_{\widetilde{G}_c})} + \sum_i^M e^{\phi(h_{\widehat{G}_c} h_{G_c^i})}}, \tag{36}$$

where positive samples $(\widehat{G}_c, \widetilde{G}_c)$ are the extracted subgraphs of graphs that have the same label of $G$, negative samples are those with different labels, $\mathbb{P}_g(G|\mathcal{Y} = Y)$ is the pushforward distribution of $\mathbb{P}(G|\mathcal{Y} = Y)$ by featurizer $g$, $\mathbb{P}(G|\mathcal{Y} = Y)$ refers to the distribution of $G$ given the label $Y$, $h_{\widehat{G}_c}, h_{\widetilde{G}_c}, h_{G_c^i}$ are the graph presentations of the estimated subgraphs, and $\phi$ is the similarity metric for the graph presentations. As $M \to \infty$, Eq. 36 approximates $I(\widehat{G}_c; \widetilde{G}_c|Y)$ which can be regarded as a non-parameteric resubstitution entropy estimator via the von Mises-Fisher kernel density [1, 41, 101].

While for the third term $I(\widehat{G}_s; Y)$ given the constraint $I(\widehat{G}_s; Y) \leq I(\widehat{G}_c; Y)$, a straightforward implementation is to imitate the hinge loss:

$$\frac{1}{N} R_{\widehat{G}_s} \cdot \mathbb{I}(R_{\widehat{G}_s} \leq R_{\widehat{G}_c}), \tag{37}$$

where $N$ is the number of samples, $\mathbb{I}$ is a indicator function that outputs 1 when the interior condition is satisfied otherwise 0, and $R_{\widehat{G}_s}$ and $R_{\widehat{G}_c}$ are the empirical risk vector of the predictions for each sample based on $\widehat{G}_s$ and $\widehat{G}_c$ respectively. One can also formulate Eq. 35 from game-theoretic perspective [14].

Finally, we can derive the specific loss for the optimization of CIGAv2 combining Eq. 36 and Eq. 37:

$$R_{\widehat{G}_c} + \alpha \mathbb{E}_{\substack{\{\widehat{G}_c, \widetilde{G}_c\} \sim \mathbb{P}_g(G|\mathcal{Y}=Y) \\ \{G_c^i\}_{i=1}^M \sim \mathbb{P}_g(G|\mathcal{Y} \neq Y)}} \log \frac{e^{\phi(h_{\widehat{G}_c}, h_{\widetilde{G}_c})}}{e^{\phi(h_{\widehat{G}_c}, h_{\widetilde{G}_c})} + \sum_i^M e^{\phi(h_{\widehat{G}_c} h_{G_c^i})}}$$
$$+ \beta \frac{1}{N} R_{\widehat{G}_s} \cdot \mathbb{I}(R_{\widehat{G}_c} \leq R_{\widehat{G}_s}), \tag{38}$$

where $R_{\widehat{G}_c}, R_{\widehat{G}_s}$ are the empirical risk when using $\widehat{G}_c, \widehat{G}_s$ to predict $Y$ through the classifier. Typically, we use a additional MLP downstream classifier $\rho_s$ for $\widehat{G}_s$ in the classifier GNN. $h_{\widehat{G}_c}$ is the graph representation of $\widehat{G}_c$ which can be induced from the GNN encoder either in the featurizer or in the classifier. $\alpha, \beta$ are the weights for $I(\widehat{G}_c; \widetilde{G}_c|Y)$ and $I(\widehat{G}_s; Y)$, and $\phi$ is implemented as cosine similarity. The optimization loss for CIGAv1 merely contains the first two terms in Eq. 38.

The detailed algorithm for CIGA is given in the Algorithm 1, assuming the $h_{\widehat{G}_c}$ is obtained via the graph encoder in $f_c$. Fig. 15 also shows a illustration of the working procedure of CIGA.

# G    Detailed Experimental Settings

In this section, we provide more details about our experimental settings in Sec. 4, including the dataset preparation, dataset statistics, implementations of baselines, selection of models and hyperparameters as well as evaluation protocols.

## G.1    Details about the datasets

We provide more details about the motivation and construction method of the datasets that are used in our experiments. Statistics of the datasets are presented in Table 4.

**SPMotif datasets.** We construct 3-class synthetic datasets based on BAMotif [116, 58] following [104], where the model needs to tell which one of three motifs (House, Cycle, Crane) that the graph contains. For each dataset, we generate 3000 graphs for each class at the training set, 1000 graphs for each class at the validation set and testing set, respectively. During the construction, we merely inject the distribution shifts in the training data while keep the testing data and validation data without the biases. For structure-level shifts (**SPMotif-Struc**), we introduce the bias based

---
**Algorithm 1** Pseudo code for the CIGA framework.

---
**Input:** Training graphs and labels $\mathcal{D}_{\text{tr}} = \{G_i, Y_i\}_{i=1}^{N}$; learning rate $l$; loss weights $\alpha, \beta$ required by Eq. 38; number of training epochs $e$; batch size $b$;
Randomly initialize parameters of $g, f_c, \rho_s$;
**for** $i = 1$ **to** $e$ **do**

    Sample a batch of graphs $\{G^j, Y^j\}_{j=1}^{b}$;

    Estimate the invariant subgraph for the batch: $\{\widehat{G}_c^j\}_{j=1}^{b} = g(\{G^j, Y^j\}_{j=1}^{b})$;

    Make predictions based the estimated invariant subgraph: $\{\widehat{Y}^j\}_{j=1}^{b} = f_c(\{\widehat{G}_c^j\}_{j=1}^{b})$;

    Calculate the empirical loss $R_{\widehat{G}_c}$ with $\{\widehat{Y}^j\}_{j=1}^{b}$;

    Fetch the graph representations of invariant subgraphs from $f_c$ as $\{h_{\widehat{G}_c^j}\}_{j=1}^{b}$;

    Calculate the contrastive loss $R_c$ with Eq. 36, where positive samples and negative samples are constructed from the batch;

    Obtain $\widehat{G}_s$ for the batch: $\{\widehat{G}_s^j\}_{j=1}^{b} = \{G^j - \widehat{G}_c^j\}_{j=1}^{b}$;

    Make predictions based on the $\widehat{G}_s$: $\{\widehat{Y}_s^j\}_{j=1}^{b} = \rho_s(\{\widehat{G}_s^j\}_{j=1}^{b})$;

    Calculate the empirical loss $R_{\widehat{G}_s}$ with $\{\widehat{Y}_s^j\}_{j=1}^{b}$, and weighted as Eq. 37;

    Update parameters of $g, f_c, \rho_s$ with respect to $R_{\widehat{G}_c} + \alpha R_c + \beta R_{\widehat{G}_s}$ as Eq. 38;

  **end for**

---

Table 4: Information about the datasets used in experiments. The number of nodes and edges are taking average among all graphs. MCC indicates the Matthews correlation coefficient.

| DATASETS | # TRAINING | # VALIDATION | # TESTING | # CLASSES | # NODES | # EDGES | METRICS |
|---|---|---|---|---|---|---|---|
| SPMOTIF | 9,000 | 3,000 | 3,000 | 3 | 44.96 | 65.67 | ACC |
| PROTEINS | 511 | 56 | 112 | 2 | 39.06 | 145.63 | MCC |
| DD | 533 | 59 | 118 | 2 | 284.32 | 1,431.32 | MCC |
| NCI1 | 1,942 | 215 | 412 | 2 | 29.87 | 64.6 | MCC |
| NCI109 | 1,872 | 207 | 421 | 2 | 29.68 | 64.26 | MCC |
| SST5 | 6,090 | 1,186 | 2,240 | 5 | 19.85 | 37.70 | ACC |
| TWITTER | 3,238 | 694 | 1,509 | 3 | 21.10 | 40.20 | ACC |
| CMNIST-SP | 40,000 | 5,000 | 15,000 | 2 | 56.90 | 373.85 | ACC |
| DRUGOOD-ASSAY | 34,179 | 19,028 | 19,032 | 2 | 32.27 | 70.25 | ROC-AUC |
| DRUGOOD-SCAFFOLD | 21,519 | 19,041 | 19,048 | 2 | 29.95 | 64.86 | ROC-AUC |
| DRUGOOD-SIZE | 36,597 | 17,660 | 16,415 | 2 | 30.73 | 66.90 | ROC-AUC |

on FIIF, where the motif and one of the three base graphs (Tree, Ladder, Wheel) are artificially (spuriously) correlated with a probability of various biases, and equally correlated with the other two. Specifically, given a predefined bias $b$, the probability of a specific motif (e.g., House) and a specific base graph (Tree) will co-occur is $b$ while for the others is $(1 - b)/2$ (e.g., House-Ladder, House-Wheel). We use random node features for SPMotif-Struc, in order to study the influences of structure level shifts. Moreover, to simulate more realistic scenarios where both structure level and topology level have distribution shifts, we also construct **SPMotif-Mixed** for mixed distribution shifts. We additionally introduced FIIF attribute-level shifts based on SPMotif-Struc, where all of the node features are spuriously correlated with a probability of various biases by setting to the same number of corresponding labels. Specifically, given a predefined bias $b$, the probability that all of the node features of a graph has label $y$ (e.g., $y = 0$) being set to $y$ (e.g., $\boldsymbol{X} = \boldsymbol{0}$) is $b$ while for the others is $(1 - b)/2$ (e.g., $P(\boldsymbol{X} = \boldsymbol{1}) = P(\boldsymbol{X} = \boldsymbol{2}) = (1 - b)/2$). More complex distribution shift mixes can be studied following our construction approach, which we will leave for future works.

**TU datasets.** To study the effects of graph sizes shifts, we follow Yehudai et al. [113], Bevilacqua et al. [11] to study the OOD generalization abilities of various methods on four of TU datasets [67], i.e., **PROTEINS, DD, NCI1, NCI109**. Specifically, we use the data splits generated by Yehudai et al. [113] and use the Matthews correlation coefficient as evaluation metric following [11] due to the class imbalance in the splits. The splits are generated as follows: Graphs with sizes smaller than the 50-th percentile are assigned to training, while graphs with sizes larger than the 90-th percentile are assigned to test. A validation set for hyperparameters tuning consists of $10\%$ held out examples from training. We also provide a detailed statistics about these datasets in table 5.

Table 5: Detailed statistics of selected TU datasets. Table from Yehudai et al. [113], Bevilacqua et al. [11].

| | NCI1 | | | NCI109 | | |
|---|---|---|---|---|---|---|
| | ALL | SMALLEST 50% | LARGEST 10% | ALL | SMALLEST 50% | LARGEST 10% |
| CLASS A | 49.95% | 62.30% | 19.17% | 49.62% | 62.04% | 21.37% |
| CLASS B | 50.04% | 37.69% | 80.82% | 50.37% | 37.95% | 78.62% |
| NUM OF GRAPHS | 4110 | 2157 | 412 | 4127 | 2079 | 421 |
| AVG GRAPH SIZE | 29 | 20 | 61 | 29 | 20 | 61 |

| | PROTEINS | | | DD | | |
|---|---|---|---|---|---|---|
| | ALL | SMALLEST 50% | LARGEST 10% | ALL | SMALLEST 50% | LARGEST 10% |
| CLASS A | 59.56% | 41.97% | 90.17% | 58.65% | 35.47% | 79.66% |
| CLASS B | 40.43% | 58.02% | 9.82% | 41.34% | 64.52% | 20.33% |
| NUM OF GRAPHS | 1113 | 567 | 112 | 1178 | 592 | 118 |
| AVG GRAPH SIZE | 39 | 15 | 138 | 284 | 144 | 746 |

**Graph-SST datasets.** Inspired by the data splits generation for studying distribution shifts on graph sizes, we split the data curated from sentiment graph data [122], that converts sentiment sentence classification datasets **SST5** and **SST-Twitter** [90, 26] into graphs, where node features are generated using BERT [25] and the edges are parsed by a Biaffine parser [32]. Our splits are created according to the averaged degrees of each graph. Specifically, we assign the graphs as follows: Those that have smaller or equal than 50-th percentile averaged degree are assigned into training, those that have averaged degree large than 50-th percentile while smaller than 80-th percentile are assigned to validation set, and the left are assigned to test set. For SST5 we follow the above process while for Twitter we conduct the above split in an inversed order to study the OOD generalization ability of GNNs trained on large degree graphs to small degree graphs.

**CMNIST-sp.** To study the effects of PIIF shifts, we select the ColoredMnist dataset created in IRM [4]. We convert the ColoredMnist into graphs using super pixel algorithm introduced by Knyazev et al. [46]. Specifically, the original Mnist dataset are assigned to binary labels where images with digits $0-4$ are assigned to $y = 0$ and those with digits $5-9$ are assigned to $y = 1$. Then, $y$ will be flipped with a probability of $0.25$. Thirdly, green and red colors will be respectively assigned to images with labels 0 and 1 an averaged probability of $0.15$ (since we do not have environment splits) for the training data. While for the validation and testing data the probability is flipped to $0.9$.

**DrugOOD datasets.** To evaluate the OOD performance in realistic scenarios with realistic distribution shifts, we also include three datasets from DrugOOD benchmark. DrugOOD is a systematic OOD benchmark for AI-aided drug discovery, focusing on the task of drug target binding affinity prediction for both macromolecule (protein target) and small-molecule (drug compound). The molecule data and the notations are curated from realistic ChEMBL database [63]. Complicated distribution shifts can happen on different assays, scaffolds and molecule sizes. In particular, we select `DrugOOD-lbap-core-ic50-assay`, `DrugOOD-lbap-core-ic50-scaffold`, and `DrugOOD-lbap-core-ic50-size`, from the task of Ligand Based Affinity Prediction which uses `ic50` measurement type and contains `core` level annotation noises. For more details, we refer interested readers to Ji et al. [40].

### G.2 Training and Optimization in Experiments

During the experiments, we do not tune the hyperparameters exhaustively while following the common recipes for optimizing GNNs. Details are as follows.

**GNN encoder.** For fair comparison, we use the same GNN architecture as graph encoders for all methods. By default, we use 3-layer GNN with Batch Normalization [39] between layers and JK residual connections at last layer [106]. For the architectures we use the GCN with mean readout [45] for all datasets except Proteins where we empirically observe better validation performance with a GIN and max readout [107], and for DrugOOD datasets where we follow the backbone used in the paper [40], i.e., 4-layer GIN with sum readout. The hidden dimensions are fixed as 32 for SPMotif, TU datasets, CMNIST-sp, and 128 for SST5, Twitter and DrugOOD datasets.

**Optimization and model selection.** By default, we use Adam optimizer [43] with a learning rate of $1e-3$ and a batch size of 32 for all models at all datasets. Except for DrugOOD datasets, we use a

batch size of 128 following the original paper [40]. To avoid underfitting, we pretrain models for 20 epochs for all datasets, except for CMNIST and Twitter where we pretrain 5 epochs and for SST5 we pretrain 10 epochs, because of the dataset size and the difficulty of the task. To avoid overfitting, we also employ an early stopping of 5 epochs according to the validation performance. Meanwhile, dropout [91] is also adopted for some datasets. Specifically, we use a dropout rate of 0.5 for CMNIST, SST5, Twitter, DrugOOD-Assay and DurgOOD-Scaffold, 0.1 for DrugOOD-Size according to the validation performance, and 0.3 for TU datasets following the practice of Bevilacqua et al. [11].

**Implementations of baselines.** For implementations of the interpretable GNNs, we use the author released codes [120, 78], where we use the codes provided by the authors[6] for DIR c[104] which is the same as the author released codes. During the implementation, we use the same $s_c$ for all interpretable GNN baselines, chosen from $\{0.1, 0.2, 0.25, 0.3, 0.4, 0.5, 0.6, 0.7, 0.8, 0.9\}$ according to the validation performances, and set to 0.25 for SPMotif following Wu et al. [104], 0.3 for Proteins and DD, 0.6 for NCI1, 0.7 for NCI109, 0.8 for CMNIST-sp, 0.5 for SST5 and Twitter, and 0.8 for DrugOOD datasets, respectively. Empirically, we observe that the optimization process in GIB can be unstable during its nested optimization for approximating the mutual information of the predicted subgraph and the input graph. We use a larger batch size of 128 or reduce the nested optimization steps to be lower than 20 for stabilizing the performance. If the optimization failed due to the instability during training, we will select the results with best validation accuracy as the final outcomes. Although SPMotif-Struc is also evaluated in DIR, we find the results are inconsistent to the results reported by the author, because DIR adopts `Last Epoch Model Selection` which is *different* from the claim that they select models according to `the validation performance`, i.e., `line 264` to `line 278` in `train/spmotif_dir.py` from the commit `4b975f9b3962e7820d8449eb4abbb4cc30c1025d` of `https://github.com/Wuyxin/DIR-GNN`. We select the hyperparamter for the proposed DIR regularization from $\{0.01, 0.1, 1, 10\}$ according to the validation performances at the datasets, while we stick to the authors claimed hyperparameters for the datasets they also experimented with.

For invariant learning, we refer to the implementations in DomainBed [34] for IRM [4], V-Rex [49] and IB-IRM [2]. Since the environment information is not available, we perform random partitions on the training data to obtain two equally large environments for these objectives. Moreover, we select the weights for the corresponding regularization from $\{0.01, 0.1, 1, 10, 100\}$ for these objectives according to the validation performances of IRM and stick to it for others, since we empirically observe that they perform similarly with respect to the regularization weight choice. For EIIL [23], we use the author released implementations about assigning different samples the weights for being put in each environment and calculating the IRM loss.

Besides, for CNC [124], we follow the algorithm description to modify the sampling strategy in supervised contrastive loss [42] based on a pretrained GNN optimized with ERM, and choose the weight for contrastive loss using the same grid search as for CIGA.

**Implementations of CIGA.** For fair comparison, CIGA uses the same GNN architecture for GNN encoders as the baseline methods. We did not do exhaustive hyperparameters tuning for the loss Eq. 38. By default, we fix the temperature to be 1 in the contrastive loss, and merely search $\alpha$ from $\{0.5, 1, 2, 4, 8, 16, 32\}$ and $\beta$ from $\{0.5, 1, 2, 4\}$ according to the validation performances. For CMNIST-sp, we find larger $\beta$ are required to get rid of intense spurious node features hence we expand the search range for $\beta$ to $\{0.5, 1, 2, 4, 16, 32\}$, For Graph-SST datasets, we search $\alpha$ from $\{0.5, 1, 2, 4\}$ as we empirically find that increasing $\alpha$ does not help increase the performance with few random runs. Besides, we also have various implementation options for obtaining the features in $\widehat{G}_c$, for obtaining $h_{\widehat{G}_c}$, as well as for obtaining predictions based on $\widehat{G}_s$. By default, we feed the graph representations of featurizer GNN to the classifier GNN, as well as to the contrastive loss. For classifying $G$ based on $\widehat{G}_s$, we use a separate MLP downstream classifier in the classifier GNN $f_c$. The only exception is for the CMNIST-sp dataset where the spurious correlation is stronger than the invariant signal. Directly feeding the graph representations from the featurizer GNN can easily overfit to the shortcuts hence we instead feed the original features to the downstream classifier GNN. There can be more other options, such as using separate graph convolutions on $\widehat{G}_s$ or $\widehat{G}_c$, which we leave for future work.

**Evaluation protocol.** We run each experiment 10 on TU datasets and 5 times for others where the random seeds start from 1 to the number of total repeated times. During each run, we select the

---

[6]`https://anonymous.4open.science/r/DIR/`

model according to the validation performance and report the mean and standard deviation of the corresponding metrics.

## G.3 Software and Hardware

We implement our methods with PyTorch [73] and PyTorch Geometric [29]. We ran our experiments on Linux Servers with 40 cores Intel(R) Xeon(R) Silver 4114 CPU @ 2.20GHz, 256 GB Memory, and Ubuntu 18.04 LTS installed. GPU environments are varied from 4 NVIDIA RTX 2080Ti graphics cards with CUDA 10.2, 2 NVIDIA RTX 2080Ti and 2 NVIDIA RTX 3090Ti graphics cards with CUDA 11.3, and NVIDIA TITAN series with CUDA 11.3.

## G.4 Additional Analysis

**Hyperparameter sensitivity analysis.** To examine how sensitive CIGA is to the hyperparamters $\alpha$ and $\beta$ for contrastive loss and hinge loss, respectively, under different distribution shifts. We conduct experiments based on the hardest datasets from each table (i.e., SPMotif-Mixed with the bias of 0.9, DrugOOD-Scaffold and the NCI109 datasets from Table 1, Table 2, and Table 3, respectively.) To increase the difficulty, we search for more fine-grained spaces for both parameters, i.e., $\{0.1, 0.5, 1, 2, 3, 4, 5, 6, 7, 8\}$. During changing the value of $\beta$, we will fix the $\alpha$ to a specific value under which the model has a relatively good performance (but not the best, to fully examine the robustness of CIGA in practice). During the sensitivity tests, we follow the evaluation protocol as that used for the main experiments. The results are shown in Fig. 16 and Fig. 17.

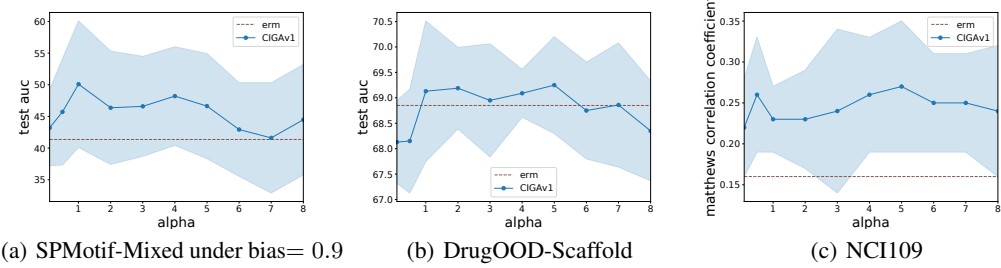

(a) SPMotif-Mixed under bias= 0.9     (b) DrugOOD-Scaffold     (c) NCI109

Figure 16: Hyperparameter sensitivity analysis on the coefficient of contrastive loss ($\alpha$).

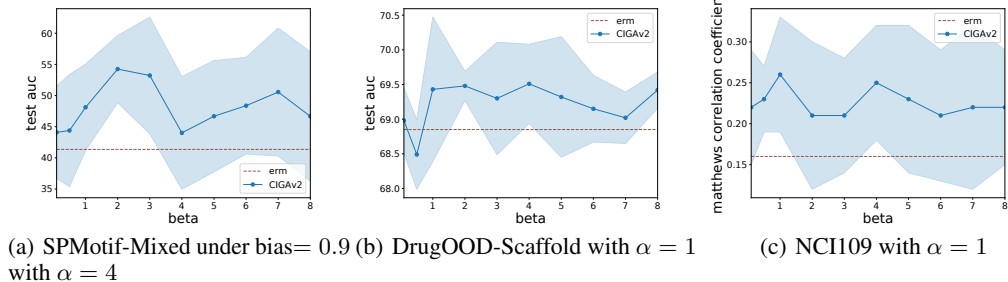

(a) SPMotif-Mixed under bias= 0.9 (b) DrugOOD-Scaffold with $\alpha = 1$     (c) NCI109 with $\alpha = 1$
with $\alpha = 4$

Figure 17: Hyperparameter sensitivity analysis on the coefficient of hinge loss ($\beta$).

From the results above, we can see that both CIGAv1 and CIGAv2 are robust to different values of $\alpha$ and $\beta$, respectively, across different datasets and distribution shifts. Notably, in Fig. 16, when the coefficient $\alpha$ for the contrastive loss become too small, the invariance of the identified invariant subgraphs $\widehat{G}_c$ may not be guaranteed, resulting worse performances. Moreover, when $\alpha$ becomes too large, it may affect the optimization and yield worse performances. In SPMotif datasets, the worse performances can be observed via the large variances as well. Similarly for $\beta$, as shown in Fig. 17, when $\beta$ becomes too small, some part from the spurious subgraph may still be contained in

the estimated invariant subgraphs. While if $\beta$ becomes too large, there might be part of $\widehat{G}_c$ being eliminated. Although both CIGAv1 and CIGAv2 are robust to the changes of $\alpha$ and $\beta$, the intrinsic difficult optimization in OOD generalization algorithms including the proposed CIGA in our work, still require a more proper and smooth optimization process [18].

Table 6: Averaged training time (sec.) per epoch of various methods on DrugOOD-Scaffold.

| METHODS | ERM | ASAP | GIB | DIR | IRM | EIIL | CNC | CIGAv1 | CIGAv2 |
|---|---|---|---|---|---|---|---|---|---|
| RUNNING TIME | 8.055 | 15.578 | 300.304 | 106.919 | 8.73 | 69.664 | 9.795 | 40.065 | 46.181 |
| OOD PERFORMANCE | 68.85 | 66.19 | 62.01 | 63.91 | 68.69 | 68.45 | 67.24 | 69.04 | 69.7 |
| AVG. RANK | 2 | 5.5 | 9 | 8 | 3 | 6 | 4.5 | 3.5 | 3.5 |

**Running time analysis.** To examine how much computational overhead is induced by the architecture and the additional objectives in CIGA, we analyze and compare the averaged training time of different methods on DrugOOD-Scaffold. Factors that could affect the running time such as GNN backbone, batch size, and the running devices (NVIDIA RTX 2080Ti, Linux Servers with 40 cores Intel(R) Xeon(R) Silver 4114 CPU @ 2.20GHz, 256 GB Memory, and Ubuntu 18.04 LTS), are fixed the same during the testing. The results are shown as in Table. 6. It can be found that CIGA is the only OOD method that outperforms ERM by a non-trivial margin with a relatively low additional computational overhead.

Table 7: Performances of different methods on Drug-Assay under single environment OOD generalization (i).

| METHODS | ERM | ASAP | GIB | DIR | CIGAv1 | CIGAv2 | ORACLE (IID) |
|---|---|---|---|---|---|---|---|
| OOD PERFORMANCE | 63.29(2.67) | 63.41(0.70) | 62.72(0.59) | 62.56(0.79) | **63.86 (0.57)** | **64.31 (0.92)** | 84.71 (1.60) |
| RANK | 5 | 4 | 8 | 9 | 2 | 1 | |

Table 8: Performances of different methods on Drug-Assay under single environment OOD generalization (ii).

| METHODS | ERM | IRM | V-REX | EIIL | IB-IRM | CNC | CIGAv1 | CIGAv2 | ORACLE (IID) |
|---|---|---|---|---|---|---|---|---|---|
| OOD PERFORMANCE | 63.29(2.67) | 63.25(1.45) | 62.18(1.71) | 62.95(1.37) | 61.95(1.72) | 63.61(0.96) | **63.86 (0.57)** | **64.31 (0.92)** | 84.71 (1.60) |
| RANK | 5 | 6 | 10 | 7 | 11 | 3 | 2 | 1 | |

**Single environment OOD generalization.** The theory of invariant learning fundamentally assume the presence of multiple environments [76, 4]. However in practice, it does not always hold, which would inevitably fail all of the invariant learning solutions [4, 49, 23, 2], including CIGA.

Nevertheless, to examine how CIGA performs under various realistic scenarios, we conduct an additional experiment based on DrugOOD-Assay. We select samples that are from the largest assay group (i.e., the biochemical functionalities of these molecules are tested and reported under the same experimental setup in the lab) [40]. The results are separated and shown in Table 7 and Table 8. Besides the baselines, we also show the "Oracle" performances from the main table, to demonstrate the performance gaps.

From the Table 7 and Table 8, we can see that, both CIGAv1 and CIGAv2 maintain their state-of-the-art performances even in the single training environment setting. We hypothesize that enforcing the mutual information between the estimated $\widehat{G}_c$ also helps to retain the invariance even under the single training environment setting. That may partially explain why CNC can bring some improvements. We believe it is an interesting and promising future direction to develop in-depth understanding and better solutions under this circumstance.

## G.5 Interpretation Visualization

Since we use the interpretable GNN architecture to implement CIGA[7], it brings an additional benefit that provides certain interpretation for the predictions automatically, which may facilitate human understanding in practice.

---

[7]We use the code provided by [64].

First, we provide some interpretation visualizations in SPMotif-Struc and SPMotif-Mixed datasets, under the biases of 0.6 and 0.9. Shown in Fig. 18 to Fig. 21, we use pink to color the ground truth nodes in $G_c$, and denote the relative attention strength with edge color intensities.

Besides, we also provide some interpretation visualization examples in DrugOOD datasets. Shown in Fig. 22 to Fig. 27, we use the edge color intensities to denote the attentions of models that pay to the corresponding edge. Some interesting patterns can be found in the molecules shared with the same label, which could provide insights to the domain experts when developing new drugs. We believe that, because of its superior OOD generalization performance on graphs, CIGA can have high potential to push forward the developments of AI-Assisted Drug Discovery, and enrich the AI tools for facilitating the fundamental practice of science in the future.

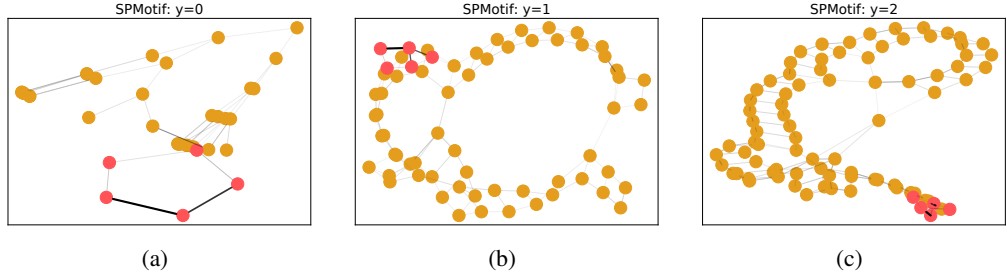

Figure 18: Interpretation visualization of examples from SPMotif-Struc under bias= 0.6.

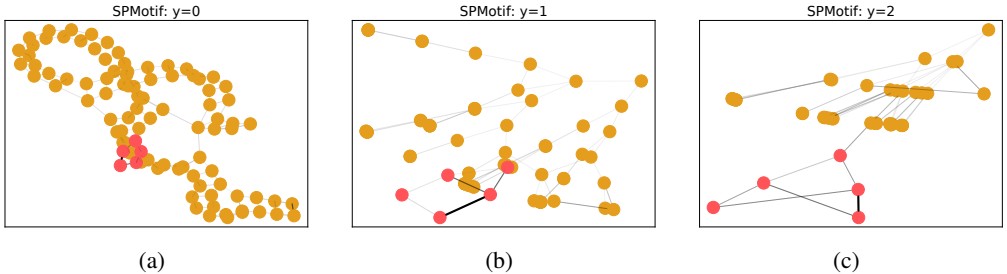

Figure 19: Interpretation visualization of examples from SPMotif-Struc under bias= 0.9.

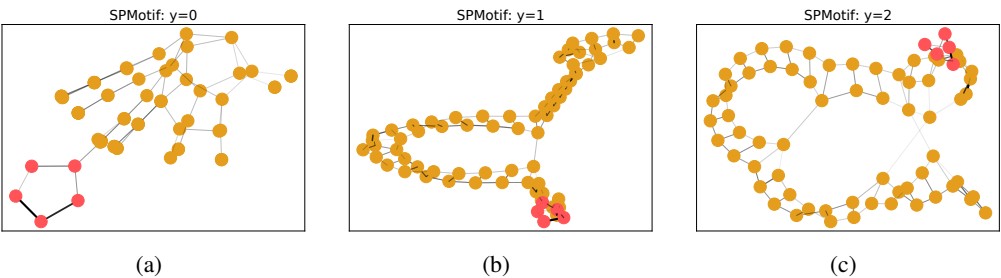

Figure 20: Interpretation visualization of examples from SPMotif-Mixed under bias= 0.6.

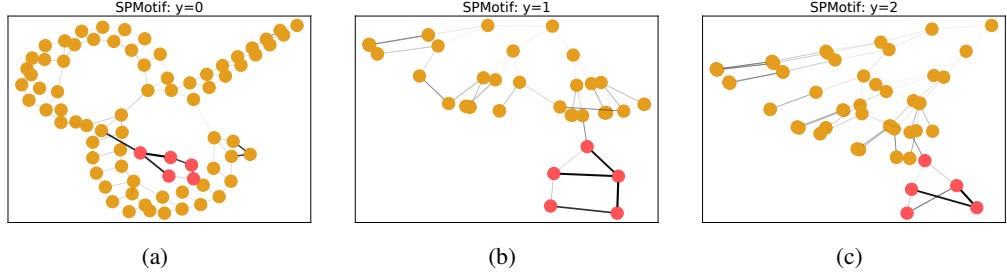

Figure 21: Interpretation visualization of examples from SPMotif-Mixed under bias= 0.9.

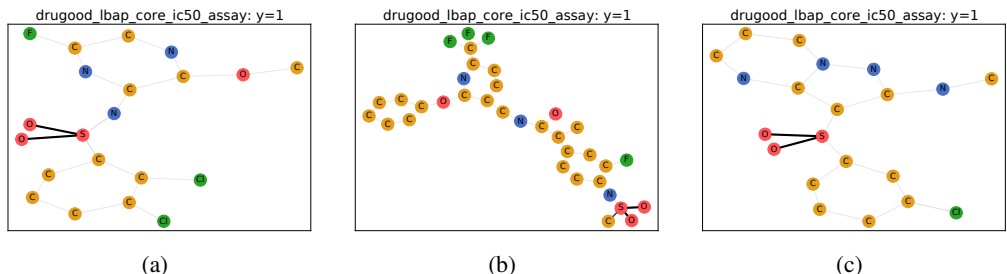

Figure 22: Interpretation visualization of activate examples ($y = 1$) from DrugOOD-Assay.

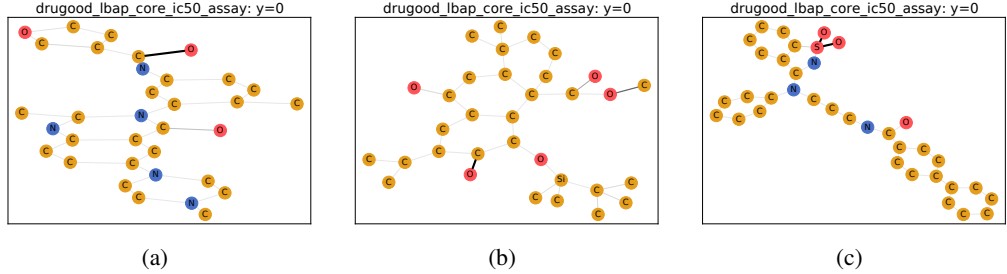

Figure 23: Interpretation visualization of inactivate examples ($y = 0$) from DrugOOD-Assay.

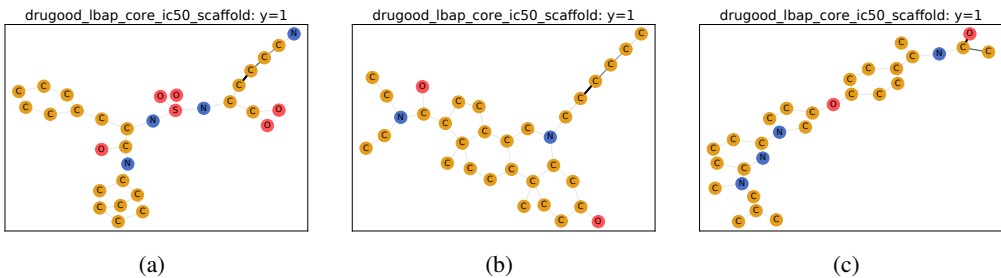

Figure 24: Interpretation visualization of activate examples ($y = 1$) from DrugOOD-Scaffold.

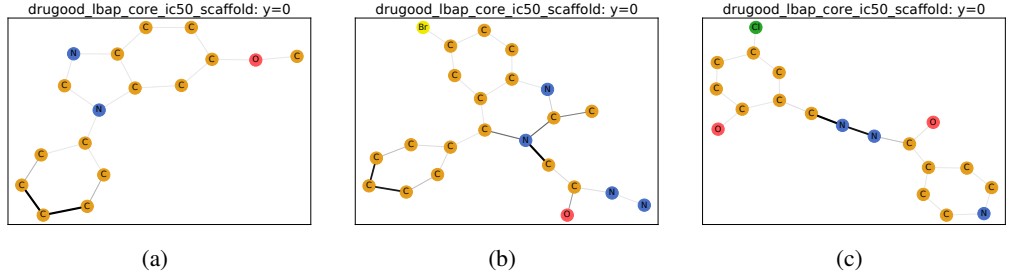

Figure 25: Interpretation visualization of inactivate examples ($y = 0$) from DrugOOD-Scaffold.

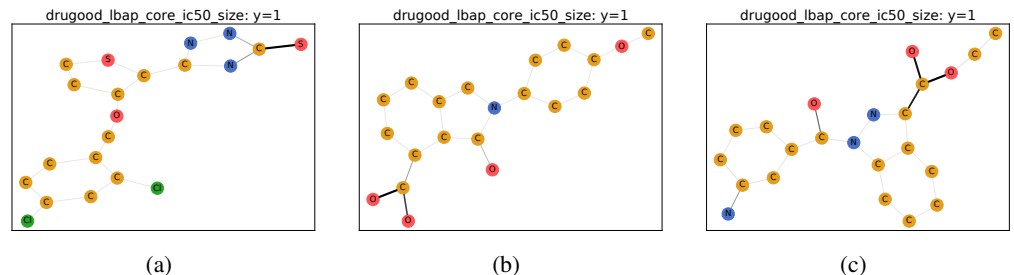

Figure 26: Interpretation visualization of activate examples ($y = 1$) from DrugOOD-Size.

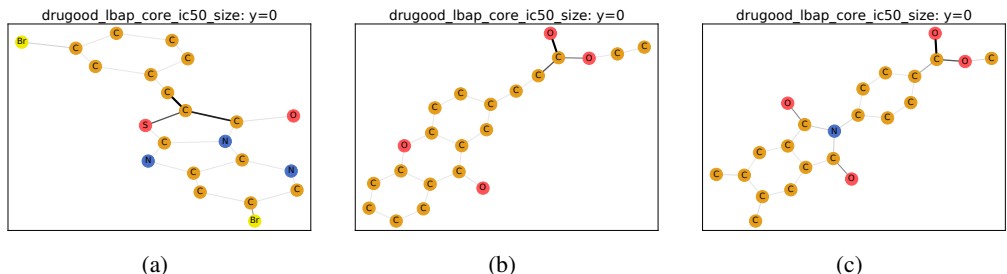

Figure 27: Interpretation visualization of inactivate examples ($y = 0$) from DrugOOD-Size.