# OpenReview forum: "Learning Causally Invariant Representations for Out-of-Distribution Generalization on Graphs"
_NeurIPS.cc/2022/Conference — NeurIPS 2022 Accept_

### Official Review · Reviewer_tMnp · 2022-07-08

**Rating:** 6
**Confidence:** 4
**Soundness:** 3 good
**Presentation:** 3 good
**Contribution:** 2 fair

**Summary:**

The authors consider the problem of out-of-distribution generalization on graphs, where shifts in the distribution can change both the attributes and the structure of the graphs. They propose two causal DAGs describing the shift, and assume the existence of a subgraph which is invariant to the shift and that contains most of the information about the label. Then, they present two learning objectives to identify this critical subgraph which is then used for the label prediction.

**Questions:**

1. What is the relationship between $E_G$ and $E$ ? I can see that $E_G \subseteq E$ but if we replace G in the FIIF SCM with its $\mathcal{G}$-Gen SCM, then: (1) there is no causal relationships between $E_G$ and $E$ and (2) $E_G \perp E$, which is contradictory.
Can't you simply remove the random variable $E_G$?
2. From Assumption 2.1 we have $G = f_{gen}^G(G_C, G_S, E_G)$ and since $E_G \subseteq E$ then the environment also causes G. This obviously would change the causal diagram since then there should be an arrow from the environment to G, changing all the causal models. Please clarify.



**Ethics Review Area:**

["I don’t know"]

**Strengths And Weaknesses:**

#### Strenghts
1. The assumption of the presence of an invariant subgraph is interesting and reasonable, and it is in line with previous work.
2. The empirical evaluation shows the effectiveness of the proposed approach.

#### Weaknesses
1. The learning objective assumes the presence of multiple training environments, and thus the proposed method would not work in the single environment setting.
2.  The relationship between $E_G$ and $E$ is unclear, since they are not causally related. Also, since $E_G$ is an input of $ f_{gen}^G$, and $E_G \subseteq E$ there should be an arrow from $E$ to $G$.

---

> ### Author Response · Authors · 2022-08-02
> **Response to Reviewer tMnp**
>
> Thank you for taking the time to review our paper. Please see our detailed responses to your questions and suggestions below.
>
> **1. The learning objective assumes the presence of multiple training environments, and thus the proposed method would not work in the single environment setting (weakenss 1).**
>
> When developing CIGA, we follow the common assumptions of the invariant learning literature that assumes the existence of multiple training environments [1,2]. Therefore, it’s a common issue for all of the existing invariant learning solutions, including CIGA, that they can’t work in the single environment setting.
>
> We have also conducted preliminary experiments to look into the capabilities of CIGA in single environment OOD generalization, with the hope of providing more guidance for future developments under this circumstance. Specifically,  in the experiments,  we selected samples that are from the largest assay group in the training data (i.e., the biochemical functionalities of these molecules are tested and reported under the same laboratory setup [3]) as the new training set, while keeping the validation and test data the same. The results are shown in the table below:
> | Methods | OOD Performances |  Rank |
> |:-------:|:----------------:|:-----:|
> |   ERM   |    63.28(2.67)   |   5   |
> |   ASAP  |    63.41(0.70)   |   4   |
> |   GIB   |    62.72(0.59)   |   8   |
> |   DIR   |    62.56(0.79)   |   9   |
> |   IRM   |    63.25(1.45)   |   6   |
> |  V-REX  |    62.18(1.71)   |   10  |
> |   EIIL  |    62.95(1.37)   |   7   |
> |  IB-IRM |    61.95(1.72)   |   11  |
> |   CNC   |    63.61(0.96)   |   3   |
> |  CIGAv1 |  **63.86(0.57)** | **2** |
> |  CIGAV2 |  **64.31(0.92)** | **1** |
>
> Interestingly, while invariant learning methods (e.g., IRM, V-REX, etc.) fail as expected, CIGA remains its good performance. We hypothesize that enforcing the mutual information between the estimated $\widehat{G}_c$ also helps to retain the invariances even under the single training environment setting. That may partially explain why CNC can bring some improvements.
>
> We believe it is interesting and promising to develop an in-depth understanding of the observed phenomenon, and explore better solutions under single domain/environment OOD generalization in future work, with the guidance of the invariance principle and causal theory [1,2,3,4].
>
>
>
> **2. The relationship between $E_G$ and $E$ is unclear (weakness 2 and Q1). $E_G$ in $\mathcal{G}$-Gen SCM would change the causal diagram of FIIF SCM and PIIF SCM (Q2).**
>
> Thanks for pointing out this potentially confusing point. We intended to note the influences of the environment $E$ on the graphs. Nevertheless, we agree that it may cause readers confusion. Hence we removed $E_G$ in the revised version in order to improve the clarity, but it won’t affect the other results.
>
> More concretely, the removal/existence of $E_G$ won’t affect the application of the invariance principle and hence the derivation of our solutions. The working rationale of CIGA mainly rely on the invariant relationship $P(Y \|C)$, which is independent to the changes of the environments $E$ and $E_G$, according to the Independent Causal Mechanism (ICM) assumption [1,3,4]. In other words, $P(Y\|C)$ remains invariant, no matter the removal/existence of $E_G$. Therefore, $P(Y\|G_c)$ remains invariant, and it follows that  Eq. 1 still holds. Then, as the followup derivations of CIGA solutions mainly aim to realize Eq. 1, the followup results won’t be affected by the removal of $E_G$.
>
> We hope our responses could clarify your concerns. Please let us know if you have further questions.
>
> ---
> [1] Peters et al., Causal inference using invariant prediction: identification and confidence intervals, Journal of the Royal Statistical Society, 2016.
>
> [2] Arjovsky et al., Invariant Risk Minimization, arXiv 2020.
>
> [3] Ji et al., DrugOOD: Out-of-Distribution (OOD) Dataset Curator and Benchmark for AI-aided Drug Discovery -- A Focus on Affinity Prediction Problems with Noise Annotations, arXiv 2022.
>
> [4] Judea Pearl, Causality, Cambridge University Press, 2 edition, 2009.
>
> [5] Peters et al., Elements of Causal Inference: Foundations and Learning Algorithms, The MIT Press, 2017.

---

> ### Author Response · Authors · 2022-08-06
> **Welcome for more discussions**
>
> Dear reviewer tMnp,
>
> Thank you again for your efforts in reviewing our paper and your valuable comments. We’d be grateful if you can confirm whether our response has addressed your concerns. We’d be glad to answer any outstanding questions and look forward to any further discussions.

---

> ### Author Response · Authors · 2022-08-07
> **Window for discussion and draft updating is closing**
>
> Dear Reviewer tMnp,
>
> We have updated our draft following your suggestions. The changes were listed in this [reply](https://openreview.net/forum?id=A6AFK_JwrIW&noteId=KiTJrAO0O7y). As the window for discussion and draft updating is closing, please let us know ASAP if you feel other changes are needed to address your concerns.
>
> Thanks, The Anonymous Authors.

---

> > ### Comment · Reviewer_tMnp · 2022-08-08
> > **Post Rebuttal Comment**
> >
> > I thank the author for their reply. My concern regarding the presence of $E_G$ has been addressed, and the SCM now seems correct. I will keep my score unchanged.

---

> > > ### Author Response · Authors · 2022-08-08
> > > **Thank you**
> > >
> > > Dear Reviewer tMnp,
> > >
> > > Thank you again for your time and efforts in reviewing our paper.
> > >
> > > The Anonymous Authors.

---

### Official Review · Reviewer_QGqs · 2022-07-10

**Rating:** 6
**Confidence:** 2
**Soundness:** 3 good
**Presentation:** 2 fair
**Contribution:** 3 good

**Summary:**

This paper studies the OOD generalization problem for graphs. It takes a causal view at the problem and proposes four assumptions to describe the graph generation process. Specifically, three factors C (invariant), S (varying), and E (environment) control the graph generation process. The assumptions include two types of distribution shifts in which 1) graph label Y is fully dependent on C; and 2) Y is spuriously correlated with environment E.

Based on the assumptions, the authors decompose the graph learning problem into finding the an invariant subgraph $\hat{G}_c$ and minimizing the risks on $\hat{G}_c$, i.e., Equation (1). The remaining efforts are dedicated to resolve Equation (1)'s intractability. The main idea is to use supervised contrastive learning to maximize the agreement between the invariant part of graphs with the same label .


**Questions:**

Q1: since theoretical results mainly address the connection between the proposed method and definition2.5, it is crucial to understand the significance of definition 2.5. How is the minimax objective in definition2.5 relevant to finding the causal factor C in Figure2?

Q2: in Line189-190, graph's complex properties of structures and features are identified as a key challenge for OOD generalization. It is unclear how does the proposed method resolve this challenge?


**Strengths And Weaknesses:**

**Strengths:**

- OOD generalization for graphs is of high interest for the machine learning community. This paper present a thorough study from the causal view, which should enlighten future researchs in this field.
- The idea is novel and motivated by theory.



**Weaknesses:**

- The method seems complex that includes two GNNs and three loss functions. It is unclear about the practical value of this method.

- The presentation can be improved.
    - Section3.2 is hard to parse. The reasoning process between the multiple objective functions is poorly organized and hard to evaluate.
    - Some equations can be made clearer. For example, what does the C, C', c, c' in Equation (2) mean? Why is the risk term in Equation (1) replaced by mutual information in Equation (3)?

---

> ### Author Response · Authors · 2022-08-02
> **Response to Reviewer QGqs [5/5]**
>
> We’d be grateful if you could take the above responses into consideration when making the final evaluation of our work. Please let us know if there are any outstanding questions.
>
> ---
>
> [1] Ji et al., DrugOOD: Out-of-Distribution (OOD) Dataset Curator and Benchmark for AI-aided Drug Discovery -- A Focus on Affinity Prediction Problems with Noise Annotations, arXiv 2022.
>
> [2] Xu et al., What can neural networks reason about? ICLR 2020.
>
> [3] Xu et al., How neural networks extrapolate: From feedforward to graph neural networks, ICLR 2021.
>
> [4] Ganin et al., Domain-Adversarial Training of Neural Networks, Journal of Machine Learning Research, 2016.
>
> [5] Arjovsky et al., Invariant Risk Minimization, arXiv 2020.
>
> [6] Chang et al., Invariant Rationalization, ICML 2020.
>
> [7] Yu et al., Graph Information Bottleneck for Subgraph Recognition, ICLR 2021.
>
> [8] Wu et al., Discovering Invariant Rationales for Graph Neural Networks, ICLR 2022.
>
> [9] Goodfellow et al., Generative Adversarial Networks, NIPS 2014.
>
> [10] Monti et al., Geometric Matrix Completion with Recurrent Multi-Graph Neural Networks, NIPS 2017.
>
> [11] Dai et al., Learning Transferable Graph Exploration, NeurIPS 2019.
>
> [12] Creager et al., Environment Inference for Invariant Learning, ICML 2021.
>
> [13] Ahuja et al., Invariance Principle Meets Information Bottleneck for Out-of-Distribution Generalization, NeurIPS 2021.
>
> [14] Bevilacqua et al., Size-Invariant Graph Representations for Graph Classiﬁcation Extrapolations, ICML 2021.
>
> [15] Miao et al., Interpretable and Generalizable Graph Learning via Stochastic Attention Mechanism, ICML 2022.
>
> [16] Sener and Koltun, Multi-Task Learning as Multi-Objective Optimization, NeurIPS 2017.
>
> [17] Chen et al., Pareto Invariant Risk minimization, arXiv 2022.
>
> [18] Peters et al., Causal inference using invariant prediction: identification and confidence intervals, Journal of the Royal Statistical Society, 2016.
>
> [19] Wu et al., Graph Information Bottleneck, NeurIPS 2020.
>
> [20] Alemi et al., Deep Variational Information Bottleneck, ICLR 2017.
>
> [21] Ahmad and Lin. A nonparametric estimation of the entropy for absolutely continuous distributions (corresp.), IEEE Transactions on Information Theory, 1976.
>
> [22] Kandasamy et al., Nonparametric von mises estimators for entropies, divergences and mutual informations, NIPS 2015.
>
> [23] Wang and Isola, Understanding contrastive representation learning through alignment and uniformity on the hypersphere, ICML 2020.
>
> [24] Bevilacqua et al., Size-Invariant Graph Representations for Graph Classiﬁcation Extrapolations, ICML 2021.
>
> [25] Lovasz and Szegedy, Limits of dense graph sequences, Journal of Combinatorial Theory, 2016.
>
> [26] Snijders and Nowicki, Estimation and prediction for stochastic blockmodels for graphs with latent block structure, Journal of classiﬁcation, 1997.
>
> [27] Yehudai et al., From local structures to size generalization in graph neural networks, ICML 2021.

---

> > ### Comment · Reviewer_QGqs · 2022-08-08
> > **Comment on the Rebuttal**
> >
> > Thanks authors for the detailed rebuttal. I appreciate the efforts in explaining the minimax objective of Def2.5. However, I would like to point out that the two following questions still remain to be solved.
> >
> > 1. **Graph's complex properties of structures and features**: From the rebuttal, it seems that the effort to resolve this challenge is to apply a GNN encoder. The following steps of finding an invariant subgraph is not specific to graphs, but can be applied to vision and language as well. The proposed method does not treat the spurious correlations in node features and graph structures separately. Is that correct?
> >
> > 2. **The complexity of the proposed method**: I appreciate the authors' efforts in presenting experiments for running time and the sensitivity to hyperparameters. However, implementing the proposed strategy still requires more work than others. It also requires more tests to understand the effectiveness of each module in the methodology.
> >
> > Due to the issues above, I reserve the previous rating.

---

> > > ### Author Response · Authors · 2022-08-08
> > > **Follow-up response to Reviewer QGqs [2/2]**
> > >
> > >
> > > Regarding the complexity of the solution:
> > > We’d like to highlight that, **CIGA is the best method that trades off the number of assumptions between the complexity of the solution for OOD generalization on graphs**.
> > >
> > > > Implementing the proposed strategy still requires more work than others.
> > >
> > > Yes, but **we would like to emphasize that all existing OOD methods, hence including our CIGA, require more works**. Specifically, more works are indeed needed for all methods that try to mitigate more distribution shifts, or discard the environment labels, or handle the graph’s complexity, as shown in the following table (modified from Figure 1 (b) in the paper).
> > >
> > > |                 | More distribution shifts | Known environment labels | Involved graph's complexity | Networks/Procedures | Objectives/Constraints |
> > > |-----------------|--------------------------|--------------------------|-----------------------------|:-------------------:|:----------------------:|
> > > | ERM             | N/A                      | N/A                      | No                          |          1          |            1           |
> > > | IRM             | No                       | Yes                      | No                          |          1          |            2           |
> > > | IB-IRM          | Yes                      | Yes                      | No                          |          1          |            3           |
> > > | EIIL            | No                       | No                       | No                          |          2          |            2           |
> > > | GIB             | No                       | No                       | Yes                         |          2          |            3           |
> > > | DIR             | No                       | No                       | Yes                         |          2          |            3           |
> > > | **CIGA (Ours)** | **Yes**                  | **No**                   | **Yes**                     |        **2**        |          **3**         |
> > >
> > > However, in contrast to the previous solutions, we resolve the OOD generalization on graphs (more distribution shifts, no environment labels, graph’s complexity) with minimal extra efforts.
> > >
> > > > It also requires more tests to understand the effectiveness of each module in the methodology.
> > >
> > > No, no additional tests are required to understand the effectiveness of each module. As also acknowledged in your review, that CIGA is motivated by theory, we understand the functionalities of each component/objective at the beginning without more tests.
> > >
> > > We look forward to more discussion with you if you have any further concerns or questions, and we sincerely hope you could take our above responses into consideration when you consider the rating. Thank you.

---

> > > ### Author Response · Authors · 2022-08-08
> > > **Follow-up response to Reviewer QGqs [1/2]**
> > >
> > > Thanks for your follow-up reply. We’d like to address your concerns in the following response (We will use the references from the previous [reply](https://openreview.net/forum?id=A6AFK_JwrIW&noteId=Z2qV6MFcwex)):
> > >
> > > Regarding the efforts to resolve the graph’s complexity:
> > >
> > > > From the rebuttal, it seems that the effort to resolve this challenge is to apply a GNN encoder. The proposed method does not treat the spurious correlations in node features and graph structures separately. Is that correct?
> > >
> > > Our resolution is not simply applying a GNN encoder. In fact, we are aware of the separation of nodes and edges, and we put a lot of efforts in both theoretical analysis and implementation to resolve the graph’s complexity. We summarize our efforts as follows:
> > >
> > > In theory,
> > >
> > > i). When deriving the SCMs, we discuss that spurious correlation could exist at both nodes and edges. That motivates us to take only a subset of both the node features and edges (resulting in a subgraph) for making predictions.
> > >
> > > ii). As the spurious correlations in node features and the edges can have different correlation modes with the label $Y$, we are motivated to propose a solution that works for the mixed spurious correlations. Combining both (i) and (ii),  we arrive at the invariant graph learning objective Eq. 1.
> > >
> > > iii). When pursuing realizable alternatives of Eq. 1, we adopt several mutual information measures that are estimated based on the extracted subsets of node features and edges in the subgraph.
> > >
> > > In the implementation, the learning objectives derived from the theory require us to estimate the mutual information of the subgraphs that include both nodes and edges. Separately estimating the mutual information based solely on nodes or solely on edges could suffer from larger biases. Therefore, we instead estimate the mutual information based on the learned graph representations that encode the information of the subgraphs (including both nodes and edges) to obtain a more accurate estimation.
> > >
> > > If the reviewer would like to see *more explicit and specialized connections between our methods and graph properties*, we have provided an [example](https://openreview.net/forum?id=A6AFK_JwrIW&noteId=2A0iG8sKLK0) involved with [graphon](https://en.wikipedia.org/wiki/Graphon) which is a typical concept from graph theory. We show that, by assuming $C$ as a graphon, the whole framework of CIGA can be generalized to the previous SOTA solutions [24, 27] that resolve only the graph size shifts.
> > >
> > > > The following steps of finding an invariant subgraph is not specific to graphs, but can be applied to vision and language as well.
> > >
> > > Thanks for pointing out an important aspect concerning the relationship between regular data (vision, languages) and irregular data (graphs).
> > >
> > > - In general, many solutions to graphs can be generalized to vision and languages, as both images and natural languages can be considered as specific variants of graphs. For example, an image can be considered as a single node graph,  and a sentence can be considered as a line graph.
> > > - However, the complexity of graphs prohibits the adoption of the methods developed for regular data. As we discussed in Section 2.3, previous OOD methods developed for Euclidean data all fail to resolve OOD generalization on graphs.
> > > - In addition, the theoretical analysis and the mutual information estimation would be much easier if we do not consider the complexity of graphs.

---

> > > ### Author Response · Authors · 2022-08-09
> > > **Window for discussion is closing**
> > >
> > > Dear Reviewer QGqs,
> > >
> > > As the window for discussion is closing, we’d be grateful if you can confirm whether our follow-up response below has addressed your concerns. We look forward to more discussions with you if you have any outstanding concerns or questions.
> > >
> > > Thanks, The Anonymous Authors.

---

> ### Author Response · Authors · 2022-08-02
> **Response to Reviewer QGqs [4/5]**
>
> Besides, as the first work that aims to handle the comprehensive graph distribution shifts with theoretical guarantees, **we aim at a general solution guided by general causal models under minimal prior knowledge. However, our CIGA is compatible and open to combining more knowledge about the graph properties.** Here are two examples:
>
> i). If we assume that the graphs come from the graphon family [25]. Our SCMs ($\mathcal{G}$-Gen SCM and FIIF SCM) generalize to the graph generative SCM studied in [24], as well as the Erdos-Renyi graphs and Stochastic Block Model graphs [26], which are two classical and widely studied random graph families.  In this example, the edge connection patterns controlled by the graphon $C$, e.g., motif appearance frequency or subgraph densities, act as an informative and invariant indicator for the label $Y$. In contrast, the node attributes and the number of nodes (or graph sizes) are controlled by the spurious factor $S$. Consequently, the invariant subgraph $G_c$ in this example can be regarded as the informative edges indicating the underlying patterns (e.g., edges across two communities). CIGA is expected to leverage these informative edges to predict $Y$, hence can generalize across different environments $E$, which converges to the rationales of the previous state-of-the-art solutions [24,27].
>
> ii). In a more realistic example, i.e., DrugOOD [1], it’s usually the case that a small functional group in a molecule will control the interested biochemical property prediction, or the binding affinity of a molecule to a protein. The relationships between these functional groups and the activate level of the interested biochemical property (e.g., molecule solubility), or binding affinity to the interested protein (e.g., the COVID-19 viral receptor ACE2 protein), are invariant to the changes of examination environments (assays) or the scaffolds of the molecules. Therefore, these functional groups can act as the invariant subgraph $G_c$ which is expected to be identified by CIGA, to make stable predictions about the activate level.

---

> ### Author Response · Authors · 2022-08-02
> **Response to Reviewer QGqs [3/5]**
>
> **3. How is the minmax objective in Def. 2.5 relevant to finding the causal factor $C$ in Figure 2? (Q1)**
>
> We agree that it is crucial to understand the significance of Def. 2.5, hence in the revised version, we added more discussions about Def. 2.5 and in Appendix E.1. Here is a brief discussion of the relatedness between Def. 2.5 and finding the causal factor C to make predictions.
>
> Definition 2.5 basically follows the literature of invariant learning [18,5]. In particular, the minmax formulation is motivated by the causal analysis of the SCMs we built to characterize the potential distribution shifts on graphs. Given a GNN $f$, Def. 2.5 requires $f$ to satisfy the following two requirements:
>
> i). $f$ cannot rely on any parts of $G_s$ or $S$ to make predictions. Otherwise, as $G_s$ and $S$ can be arbitrarily changed under the changes of $E$, thus $f$ can fail catastrophically on the graphs from the specific environment. In other words, the dependence of $f$ on any parts of $S$ would enlarge the maximal possible error in any environment.
>
> ii). $f$ needs to fully use the available $G_c$ or $C$ to make predictions. Otherwise, if there are any parts of $G_c$ or $C$ not leveraged by $f$, including those parts in $f$ to make predictions can always improve the predictive power of $f$ about $Y$. In other words, including more information from $C$ would minimize the maximal possible error in any environment.
>
> Combining i) and ii), it suffices to know that the minmax objective in Def. 2.5 needs $f$ to identify and fully use the available information about $C$ presented in the input graph $G$, which lays the foundation for the inductions of Eq. 1 as well as the followup CIGA solutions.
>
>
> **4. Graph’s complex properties are identified as key challenges, but it is unclear how the proposed method resolve the graph’s complexity (Q2).**
>
> Thanks for your insightful comments. As one of the major barriers in OOD generalization on graphs (the other is the lack of environment labeling), graph’s complex properties of structure and features can raise two specific challenges:
>
> i).  Graphs can contain different levels of distribution shifts, e.g., structure-level shifts or node feature-level shifts. Information theory provides useful tools to handle the complexity of graph structure which is also widely adopted in the literature [20, 7]. The overall rationale is that, we can use the information theoretic tools to derive the desired learning objectives, and then use the proper tools to implement the objectives [20,7]. More concretely in CIGA, based on Eq. 1, we derive two variants of learning objectives that mainly enforce the informativeness and invariance of the estimated invariant subgraph $\widehat{G}_c$. To implement these objectives, i.e., estimating the corresponding information theoretic objectives, we apply the GNN encoders to translate the complex structure-level and node attribute-level information into the learned graph representations. Then, we use the learned graph representations to approach the maximization of $I(\widehat{G}_c; Y)$ via a variational characterization [7, 21], and the maximization of $I(\widehat{G}_c;\widetilde{G}_c\|Y)$ via a non-parameteric resubstitution entropy estimator [21,22,23]. The two approximations lead us to the empirical risks of using the estimated $\widehat{G}_c$ to predict $Y$, and the contrastive loss.
>
> ii). The other challenge is that different levels of shifts on graphs can spuriously correlate with the labels in different modes, i.e., FIIF or PIIF. Our causal analysis allows us to leverage the invariant relationship between $G_c$ and $Y$ to make stable predictions, under both FIIF and PIIF spurious correlation modes, which induces Eq. 1 and hence the followup CIGAv1 and CIGAv2 objectives.
>
> *(To avoid tedious replications in the reply, we kindly refer you to the similar [reply](https://openreview.net/forum?id=A6AFK_JwrIW&noteId=GeZpsENaiNN) to Reviewer zrMs for a more detailed discussion of the two aforementioned points.)*

---

> ### Author Response · Authors · 2022-08-02
> **Response to Reviewer QGqs [2/5]**
>
> **b). CIGA introduces 3 losses, but requires little extra tuning efforts, compared with existing methods:**
> - First, we’d like to note that adopting multiple losses or even additional optimization subprocedure is common in the literature of both invariant learning and graph neural networks in practice [4,5,6,7,8,12,13].
> - In the experiments, we didn't tune hyperparameters exhaustively, but CIGA still maintains high performances. To examine the sensitivity of CIGA to the coefficients of the two additional objectives in practice, i.e., $\alpha$  for $I(\widehat{G_c};\widetilde{G}_c\|Y)$ implemented as the contrastive loss, and $\beta$ for $I(\widehat{G}_s; Y)$ implemented as the hinge loss, we conduct the following ablation study on the most difficult datasets, SPMotif-Mixed with a bias of 0.9, DrugOOD-Scaffold, and NCI109, where we vary the values of $\alpha$, and the values of $\beta$ under a fixed $\alpha$ that yields relatively good performance in CIGAv1. Here we provide the results on DrugOOD-Scaffold, where we vary $\beta$ under a fixed $\alpha=1$, the other results are added to Appendix G.4 in the revised paper:
>
> | Coefficients |         0.1 |         0.5 |           1 |      2      |      3      |      4      |      5      |      6      |      7      |      8      |
> |-------------:|------------:|------------:|------------:|:-----------:|:-----------:|:-----------:|:-----------:|:-----------:|:-----------:|:-----------:|
> |       $\alpha$ for $I(\widehat{G_c};\widetilde{G}_c\|Y)$ | 68.13(0.81) | 68.15(1.02) | 69.13(1.38) | 69.19(0.80) | 68.95(1.11) | 69.09(0.47) | 69.25(0.95) | 68.75(0.95) | 68.86(1.22) | 68.35(0.98) |
> |       $\beta$ for $I(\widehat{G}_s;Y)$ | 68.98(0.47) | 68.49(0.50) | 69.43(1.04) | 69.48(0.21) | 69.30(0.81) | 69.51(0.57) | 69.32(0.87) | 69.15(0.48) | 69.02(0.37) | 69.42(0.26) |
>
> From the results, we can see that both CIGAv1 and CIGAv2 maintain non-trivial OOD performance improvements across various choices of the hyperparameters, demonstrating their robustness and usefulness in practice.
>
> In philosophy, we all aim for a simple and principal solution. However, it is usual to require more objectives/constraints when the inductive bias about the graph generation process is lacking. Nevertheless, we believe it is promising to reduce the number of losses by combining more external knowledge and more advanced architecture [14,15], or automatically tuning solvers [16,17].
>
> **2. Regarding the presentation of Section 3.2 (weakness 2).**
>
> We have reorganized the contents and notations to improve the clarity. Specifically:
> - When deriving the solutions, we added a beginning sentence in italic for each paragraph to demonstrate the purpose of the corresponding paragraphs.
> - We made the formulas in Sec. 3.2 more consistent, switched the expression of empirical risk in Eq. 1 to mutual information, and added an explanation after Eq. 1 to show how it relates to the empirical risk.
> - We added an explanation for the causal factors $c$ and $c’$ after Eq. 2.
> - We improved the readability of the notations(e.g., \hat, \tilde), so that readers can recognize the variables used in the analysis more clearly.
>
> Please let us know if you have any further comments about the updated versions.

---

> ### Author Response · Authors · 2022-08-02
> **Response to Reviewer QGqs [1/5]**
>
> Thank you for your detailed comments and suggestions! Please see our responses to your questions and suggestions below. (We reorganize the weakness and questions a bit to better clarify the concerns).
>
> **1. Regarding the practical value of CIGA, two GNNs and three loss functions. (weakness 1).**
>
> In the below we provide a discussion about the practical values of CIGA (coupled with corresponding revisions in the paper):
>
> To begin with, we’d like to highlight that, **CIGA can provably handle a variety of potential graph distribution shifts that may appear in multiple application scenarios.** This is owe to its solid theoretical foundations built upon the causal analysis. Thus in experiments, CIGA is able to significantly outperforms the previous state-of-the-art methods under different graph distribution shifts. Notably, **CIGA is the only method that consistently improves the OOD generalization performance than ERM on the industry-provided realistic graph OOD benchmark, i.e., DrugOOD [1].** The superiority of CIGA demonstrates its high potential to push forward the developments of AI-assisted Drug Discovery, and enrich AI tools for facilitating the fundamental practice of science.
>
> In the following, we also carefully discuss the complexity in the design of CIGA and provide more evidences to address your concerns.
>
> **a). CIGA adopts the 2-GNN architecture, but brings little additional overhead, comparing with existing methods in terms of the performance improvements and running time:**
> - In this work, we adopt interpretable GNN architectures primarily for the purpose of prototype verification, motivated by the algorithmic reasoning results that a neural network can learn a reasoning process better if its computation structure aligns with the process better [2,3]. Moreover, decomposing the model into and stacking up two neural networks is common and widely adopted in the literature of both OOD generalization, invariant learning and graph neural networks [4,5,6,7,8] and largely used in practice [9,10,11].
> - Nevertheless, to examine how much computational overhead is induced by the architecture and the additional objectives in CIGA, we analyze and compare the averaged training time (seconds per epoch) of different methods on the realistic benchmark DrugOOD-Scaffold, where we omit vrex and IB-IRM as their performances and computational costs are similar to those of IRM.
>
>     |                 |  ERM  |  ASAP  |   GIB   |   DIR   |  IRM  |  EIIL  |  CNC  | CIGAv1 | CIGAv2 |
>     |-----------------|:-----:|:------:|:-------:|:-------:|:-----:|:------:|:-----:|:------:|:------:|
>     |   Running time  | 8.055 | 15.578 | 300.304 | 106.919 |  8.73 | 69.664 | 9.795 | 40.065 | 46.181 |
>     | OOD performance | 68.85 |  66.19 |  62.01  |  63.91  | 68.69 |  68.45 | 67.24 |  69.04 |  69.7  |
>     |     AVG Rank    |   2   |   5.5  |    9    |    8    |   3   |    6   |  4.5  |   3.5  |   3.5  |
>
>     The results show that CIGA enjoys less computational overhead than other interpretable GNNs (i.e., GIB, DIR). Meanwhile, CIGA is the only OOD method that outperforms ERM by a non-trivial margin with a relatively low additional computational overhead, demonstrating its practical potential.
> - Besides, the adopted interpretable GNN architecture also offers an additional benefit, i.e., interpretability, which can further facilitate human understanding of CIGA’s predictions in practice. We provide the interpretability visualization examples of both SPMotif and DrugOOD datasets in Appendix G.5 of the revised paper. Notably, CIGA is able to find interesting substructures in the molecules from DrugOOD, which may provide new insights to human experts during the design of novel drugs in practice.
>
> Nevertheless, concerning the time cost and the estimation of mutual information, we believe it’s a promising future direction to improve the architecture used in CIGA to better suit different needs that could appear in real world such as Edge AI.

---

> ### Author Response · Authors · 2022-08-06
> **Welcome for more discussions**
>
> Dear reviewer QGqs,
>
> Thanks again for your time and efforts in reviewing our paper. Here is a summary of our detailed response below. We humbly expect you could check it and confirm whether our response has addressed your concerns:
> 1. We established comprehensive discussions about the practical value of CIGA:
> - From the performance perspective, CIGA can serve as a general solution to tackle a variety of graph distribution shifts that could appear in different practical scenarios, due to its solid theoretical foundations.
> - From the architecture perspective, the architecture used in CIGA is shown to bring little computational overhead, while bringing more benefits such as interpretability.
> - From the objective perspective, the learning objectives of CIGA is shown to be robust to the coefficients and require little extra tunning efforts.
>
> 2. We improved the readability and clarity of Section 3.2 following your suggestions.
>
> 3. We added a discussion of the equivalence between the minmax objective in Definition 2.5 and finding the causal factor C to make predictions motivated by your suggestions.
>
> 4. We provided a comprehensive discussion about how CIGA resolves the graph’s complexity:
> - Our causal analysis motivates the invariant learning objective of identifying the invariant subgraph (a subset of edges and node attributes) for making stable predictions.
> - To identify the invariant subgraph under the unavailability of environment labels, we adopt the information-theoretic tools to derive the realizable graph learning objectives. Learning with these objectives can identify the informative and invariant subgraphs as required by the invariance principle.
> - To better illustrate the relatedness, we provided two concrete examples of OOD generalization with the graphon model, and in molecular property prediction.
>
> Our response might be a bit long. We’d appreciate your patience and welcome any further discussions or questions!

---

> ### Author Response · Authors · 2022-08-07
> **Window for discussion and draft updating is closing**
>
> Dear Reviewer QGqs,
>
> We have updated our draft following your suggestions. The changes were listed in this [reply](https://openreview.net/forum?id=A6AFK_JwrIW&noteId=KiTJrAO0O7y). As the window for discussion and draft updating is closing, please let us know ASAP if you feel other changes are needed to address your concerns.
>
> Thanks, The Anonymous Authors.

---

> > ### Comment · Reviewer_QGqs · 2022-08-09
> > **Thanks for the Response**
> >
> > I have raised my rating from 5 to 6.

---

> > > ### Author Response · Authors · 2022-08-09
> > > **Thank you for your time and efforts**
> > >
> > > Thank you again for your time and efforts in reviewing our paper, and for your valuable comments that helped us to strengthen the paper!

---

### Official Review · Reviewer_zrMs · 2022-07-11

**Rating:** 6
**Confidence:** 3
**Soundness:** 3 good
**Presentation:** 2 fair
**Contribution:** 3 good

**Summary:**

This paper discusses graph-level distribution shifts on graph neural networks. It propose a causal-based invariant learning objective based on contrastive learning and prove its equivalence/approximation of the designed invariance principles. Various experiments on synthetic and real-world shift demonstrate the effectiveness of CIGA.

**Questions:**

1. What's the accuracy or F1 measure on real-world graphs? It is hard to recognize the performance gap caused by distribution shifts. In other words, the negative effect of distribution shifts are not revealed in current table.
2. What's the relatedness of graph property and proposed method / proof, maybe I am missing something during reading ( please refer to weakness 2 )

**Limitations:**

N.A.


**Strengths And Weaknesses:**

**Strengths**
1. This paper poses a great challenge towards existing OOD generalization algorithms on graphs and provide some sort explanation.
3. The connection between semi-supervised contrastive learning and invariant in Eq.5 is sound.

**Weaknesses**
1. The results in the experiments are sometimes misleading. For example, in Table 3, when baselines are better, the bond font is still on proposed two methods.
2. I read through main theorem 3.1 and its proof, most of the induction is around mutual information or information bottleneck, while the property of graph is rarely used. The author claims challenge of graph OOD is not environment label and convoluted causes of shift, while the proposed method seem to not that relevant / benefit from the nature of the graph.

---

> ### Author Response · Authors · 2022-08-02
> **Response to Reviewer zrMs [3/3]**
>
>
> Besides, as the first work that aims to handle the comprehensive graph distribution shifts with theoretical guarantees, **we start with general causal models as well as its induced solutions** (that is also partially why we use the information-theoretic tools since we aim for a general solution under minimal prior knowledge). **However, CIGA is compatible and open to incorporating more additional knowledge about the graph properties.** For example, if we assume that the graphs come from the graphon family [9]. Our SCMs ($\mathcal{G}$-Gen SCM and FIIF SCM) are generalized to the graph generative SCM studied in [10]. Specifically:
> - We can additionally assume that the causal factor $C$ is the corresponding graphon in the SCM of [10], which controls the generation of the edges between different nodes [9], and determines the label $Y$ [10]. Thus, the edge connection patterns, e.g., motif appearance frequency and subgraph densities, act as an informative and invariant indicator for the label $Y$. In this case, the invariant subgraph $G_c$ can be regarded as the informative edges indicating the underlying patterns (e.g., edges across two communities), which essentially converges to the rationales of the solutions by [10,11].
> - On the other hand, the environment $E$ and the graphon $C$ further control the generation of the node attributes (including the number of nodes and the attribute values). Therefore, a GNN model is prone to the changes of the environments if it overfits to some spurious patterns about the graph sizes or the attributes, which is consistent with the observations in the literature [10,11,12] as well as our experimental results.
>
> Given the two aforementioned additional assumptions, our SCMs are generalized to that of [10]. Moreover, following the discussion of [10], our SCMs can also be generalized to the Erdos-Renyi graphs and Stochastic Block Model graphs [13], which are two classical and widely studied random graph families. It also partially explains why CIGA can perform well under various graphs and distribution shifts, and even close the performance gaps caused by the distribution shifts in some circumstances. We provide a more detailed discussion in Appendix C.1 of the revised paper.
>
> Please let us know if you have any further questions. We’d be grateful if you could take the above responses into consideration when making the final evaluation of our work.
>
> ---
>
> [1] Peters et al., Causal inference using invariant prediction: identification and confidence intervals, Journal of the Royal Statistical Society, 2016.
>
> [2] Arjovsky et al., Invariant Risk Minimization, arXiv 2020.
>
> [3] Wu et al., Graph Information Bottleneck, NeurIPS 2020.
>
> [4] Yu et al., Graph Information Bottleneck for Subgraph Recognition, ICLR 2021.
>
> [5] Alemi et al., Deep Variational Information Bottleneck, ICLR 2017.
>
> [6] Ahmad and Lin. A nonparametric estimation of the entropy for absolutely continuous distributions (corresp.), IEEE Transactions on Information Theory, 1976.
>
> [7] Kandasamy et al., Nonparametric von mises estimators for entropies, divergences and mutual informations, NIPS 2015.
>
> [8] Wang and Isola, Understanding contrastive representation learning through alignment and uniformity on the hypersphere, ICML 2020.
>
> [9] Lovasz and Szegedy, Limits of dense graph sequences, Journal of Combinatorial Theory, 2016.
>
> [10] Bevilacqua et al., Size-Invariant Graph Representations for Graph Classiﬁcation Extrapolations, ICML 2021.
>
> [11] Yehudai et al., From local structures to size generalization in graph neural networks, ICML 2021.
>
> [12] Knyazev et al., Understanding Attention and Generalization in Graph Neural Networks, NeurIPS 2019.
>
> [13] Snijders and Nowicki, Estimation and prediction for stochastic blockmodels for graphs with latent block structure, Journal of classiﬁcation, 1997.

---

> > ### Comment · Reviewer_zrMs · 2022-08-09
> > **Thanks for the detailed rebuttals**
> >
> > I would like to appreciate the author's rebuttal to help me better understand the merits of the paper. Hence, I would like to raise my score from 5 to 6.

---

> > > ### Author Response · Authors · 2022-08-09
> > > **Thanks for your insightful comments**
> > >
> > > Dear Reviewer zrMs,
> > >
> > > We'd like to thank you again for your time and efforts in reviewing our paper. Your insightful comments have helped to further strengthen the paper a lot!
> > >
> > > The Anonymous Authors.

---

> ### Author Response · Authors · 2022-08-02
> **Response to Reviewer zrMs [2/3]**
>
> **The causal analysis lays the foundation for adopting the invariance principle [1,2] and the induction of CIGA solutions (i.e., Eq. 1, that aims to identify the underlying invariant subgraph for making predictions).** Specifically, we model the complex graph generation process and characterize the potential distribution shifts with three SCMs:
> - The causal relationship among the environment $E$ and the spurious subgraphs $G_s$ (the set of edges and node attributes that are spuriously correlated with $E$) enables us to identify and understand the key challenges in OOD generalization on graphs, e.g., the convoluted distribution shifts at different levels and from different causes.
> - Meanwhile, the causal relationship between the latent causal factor $C$, the label $Y$ and the invariant subgraph $G_c$ (the set of edges and node attributes that are invariant to the changes of $E$) provides us the possibility to approach the OOD generalization on graphs with the invariance principle from causality.
> - More formally, the invariant relationship between the invariant subgraph $G_c$ and the label $Y$ fundamentally motivates us to derive the learning objective Eq. 1. Once we identify the underlying invariant subgraph $G_c$ and let the GNN focus only on $G_c$ to predict $Y$, it is expected to be an invariant GNN (Def. 2.5), and be able to generalize to OOD graphs under different graph distribution shifts.
>
> When solving for Eq. 1 under the unavailability of environment label, **we adopt the tools from information theory as a proxy to mitigate the complexity of graphs, and to enforce the informativeness and invariance of the extracted subgraphs,** which derives two implementations as CIGAv1 (Eq. 3) and CIGAv2 (Eq. 4). In fact, the information theory is also widely adopted in the literature to handle the complexity in graphs during the objective induction [3,4]. Following the common practice in the literature, once we derive the proper information-theoretic learning objective, we can use various tools to implement it. Similarly and more concretely in CIGA:
> - Through the maximization of the mutual information between the extracted subgraph and the label, i.e., $I(\widehat{G}_c; Y)$, we can extract the most informative structural and node attributes from the original graph. The extraction is guided via a variational characterization of the mutual information between the subgraph $G_c$ and $Y$ [4,5], which yields a lower bound of $I(\widehat{G}_c; Y)$ with respect to the negative empirical risk of using $\widehat{G}_c$ to predict $Y$, i.e., $I(\widehat{G}_c; Y)\geq -R(G_c)$. Hence we can maximize the mutual information between $\widehat{G}_c$ and $Y$ by minimizing the empirical risk of using the estimated $\widehat{G}_c$ to predict $Y$, where the graph encoder in the GNN model will encode the information of the structures and the node attributes in the learned graph representations that are used to predict $Y$.
> - Besides, to avoid extracting some edges and nodes from the spurious subgraph $G_s$ into the $\widehat{G}_c$, we simultaneously maximize the mutual information of the extracted subgraphs from the same class, i.e., $I(\widehat{G}_c;\widetilde{G}_c \|Y)$. Intuitively, it enforces the estimated invariant subgraphs from the same class to share high mutual information. However, estimating the mutual information between graphs is difficult [3,4]. We thus adopt the contrastive loss to approximate $I(\widehat{G}_c;\widetilde{G}_c \|Y)$, which can be regarded as a non-parameteric resubstitution entropy estimator via the von Mises-Fisher kernel density [6,7,8]. In the implementation, we also adopt the GNN encoder to encode the information of the structures and the node attributes in the learned graph representations for calculating the losses.
> - Theorem 3.1 (i) proves the equivalence of learning with the above two objectives to identifying the underlying invariant subgraph, while under the assumption that the sizes of $G_c$ is known and fixed. To further mititgate the size constraints, we introduce another objective based on the spurious subgraph $G_s$ and  $Y$. If we can simutaneously maximize the mutual information between the estimated spurious subgraph $\widehat{G}_s=G-\widehat{G}_c$ and $Y$, i.e., $I(\widehat{G}_s; Y)$, intuitively it can "absorb" the edges and node attributes that are also (spuriously) correlated with $Y$ into the counterpart $\widehat{G}_s$ of $\widehat{G}_c$. Theorem 3.2 (ii) proves the usefulness of the incorporation of $I(\widehat{G}_s; Y)$, and its implementation is also similar to the estimation of $I(\widehat{G}_c; Y)$ except the additional hinge loss on $R(\widehat{G}_s)\geq R(\widehat{G}_c)$.

---

> ### Author Response · Authors · 2022-08-02
> **Response to Reviewer zrMs [1/3]**
>
> Thank you for taking time to carefully review our paper and for your positive feedbacks about our work. Please see our detailed responses to your comments and suggestions below. We reorganize the weakness and questions a bit to better clarify your concerns.
>
> **1. Regarding the experiment tables: some baselines are not in bold fonts (weakness 1); It is hard to recognize the performance gap caused by distribution shifts from the table (questions 1).**
>
> Thanks for pointing out the bold typo in our paper, and for your constructive suggestion about the presented tables. We have checked and corrected the bold highlights in all tables in the revised version, where CIGA remains the best OOD method on graphs.
>
> To show the performance gaps caused by the distribution shifts, we run an "Oracle" experiment for each dataset, where we run ERM using the same GNN backbone on the randomly shuffled data. In this way, the training and test data will be empirically made as "IID". The "Oracle" performances have been updated in the revised version. Here we report some of the results in the table below, where we can observe large performance gaps caused by various graph distribution shifts. In some datasets such as Twitter and Proteins, surprisingly, CIGA can perform on par with the "Oracle". Nevertheless, for some more difficult datasets such as DrugOOD datasets, there remains large space for future improvements between CIGA and the "Oracle".
>
> |              | SPMotif-Struc 0.9 | SPMotif-Mixed 0.9 |  DrugOOD-Sca |    Twitter   |   Proteins  |
> |--------------|:-----------------:|:-----------------:|:------------:|:------------:|:-----------:|
> |          ERM |    49.64 (4.63)   |    41.36 (3.29)   | 68.85 (0.62) | 60.81 (2.05) | 0.22 (0.09) |
> |   Prev. SOTA |    57.29 (14.5)   |    50.45 (4.90)   | 68.92 (0.98) | 63.50 (1.23) | 0.29 (0.00) |
> |       CIGAv1 |    51.78 (7.29)   |    49.01 (9.92)   | 69.04 (0.86) | 63.66 (0.84) | 0.40 (0.06) |
> |       CIGAv2 |    63.41 (7.38)   |    54.25 (5.38)   | 69.70 (0.27) | 64.45 (1.99) | 0.31 (0.12) |
> | Orcale (IID) |    88.70(0.17)    |    88.73(0.25)    |  84.71(1.60) |  64.21(1.77) |  0.39(0.09) |
>
> **2. Relatedness of graph property and the proposed method/proof (weakness 2 and question 2).**
>
> Indeed,  the complexity in graphs brings more challenges instead of benefits to the OOD generalization on graphs. Graph properties are fundamentally involved during the causal modeling of the challenges raised by the graph’s complexity, as well as during the induction of the CIGA solutions. We provide more detailed discussions as follows.

---

> ### Author Response · Authors · 2022-08-06
> **Welcome for more discussions**
>
> Dear reviewer zrMs,
>
> Thank you again for your time in reviewing our paper and your valuable comments on our work. We’d be grateful if you can confirm whether our response has addressed your concerns. Here is a short summary:
>
> 1. Motivated by your comment, we conducted experiments to show the OOD and IID performance gaps in the experiment tables, and found CIGA can close the performance gaps caused by the distribution shifts in some datasets.
>
> 2. We summarized how our theory and solution relate to the graph properties:
> - The causal analysis of graphs motivates the invariant learning objective of identifying the invariant subgraph for making stable predictions.
> - The information-theoretic tools help the induction of realizable invariant graph learning objectives under the unavailability of environment labels, by using and estimating mutual information among different subgraphs and labels.
> - Our analysis and results can generalize to a broad class of graph families such as graphon graphs, Erdos-Renyi graphs, and Stochastic Block Model graphs, when incorporating further assumptions.
>
> We’d be glad to answer any outstanding questions and look forward to any further discussions.

---

> ### Author Response · Authors · 2022-08-07
> **Window for discussion and draft updating is closing**
>
> Dear Reviewer zrMs,
>
> We have updated our draft following your suggestions. The changes were listed in this [reply](https://openreview.net/forum?id=A6AFK_JwrIW&noteId=KiTJrAO0O7y). As the window for discussion and draft updating is closing, please let us know ASAP if you feel other changes are needed to address your concerns.
>
> Thanks, The Anonymous Authors.

---

> ### Author Response · Authors · 2022-08-08
> **Window for discussion is closing**
>
> Dear Reviewer zrMs,
>
> Thanks again for your time and efforts in reviewing our paper. As the window for discussion is closing, we’d be grateful if you can confirm whether our response has addressed your concerns. We look forward to more discussions with you if you have any further concerns or questions.
>
> Thanks, The Anonymous Authors.

---

### Author Response · Authors · 2022-08-02
**We have uploaded a revised version [Updated on 2nd Aug.]**

Dear reviewers,

We have revised our paper following the suggestions/comments from all the reviewers. The revision is in blue color in the paper.

Specifically, we have revised our paper to improve its clarity and readability:
- We revised the SCMs and removed $E_G$ in Sec. 2.2 and in Appendix C to improve the clarity (**tMnp**), while it doesn’t change the other results as the invariant relationships between $C$, $Y$ and $G_c$ remain unchanged.
- We added a discussion about Def. 2.5 (invariant GNN) in Appendix E.1 to show how the minmax objective corresponds to identifying the causal factor $C$ (**QGqs**).
- In Sec. 3.2, we reorganized the notations, added subtitles for each paragraph, explanations to the risk term and mutual information in Eq. 1, and explanations to $C$, $C’$, $c$, $c’$ in Eq. 2 (**QGqs**).
- In experiment tables 1-3, to show the performance gaps caused by the distribution shifts (**zrMs**), we added one line at the bottom to show the "Oracle" performance of each dataset, which is obtained by running ERM based on the randomly shuffled (empirically made IID) data of each dataset.

Concerning the common question raised by Reviewer **zrMs** and Reviewer **QGqs** about how CIGA resolves the complex graph properties, we’d like to highlight that:
- The causal analysis in Sec. 2 not only provides a lens to understand the key challenges in OOD generalization on graphs, but also motivates the ultimate learning objective of CIGA (i.e., Eq. 1). By identifying the invariant subgraph $G_c$ for making predictions about $Y$, the GNN model is expected to be able to generalize to OOD graphs under various distribution shifts.
- When approaching Eq. 1 under the unavailability of environment label $E$, we leverage the information theory as a proxy to reason and derive the information-theoretic learning objectives (Eq. 3 and Eq. 4), prove their soundness (Theorem 3.1), and implement them using various tools such as variational characterization and mutual information estimator, based on the learned graph representations. This procedure follows the common practice of tackling graph related problems with information theory in the literature.
- Throughout the paper, we aim for a general characterization and solution under minimal prior knowledge about the OOD generalization on graphs. Nevertheless, our solution is compatible and open for incorporating more graph related inductive biases. In the [reply](https://openreview.net/forum?id=A6AFK_JwrIW&noteId=2A0iG8sKLK0) to Reviewer **zrMs** and the [reply](https://openreview.net/forum?id=A6AFK_JwrIW&noteId=3VpgsTh_GfJ) toReviewer **QGqs**, we show concrete examples of how our theories and solutions generalize to previous state-of-the-art when incorporating the graphon assumption on the graph family.

Concerning the practical value of CIGA, we’d like to highlight that:
- As in practice, graph distribution shifts can appear in a variety of forms. Owing to the generality of our theory, CIGA is shown to be able to generalize under various distribution shifts and graphs, and achieve new state-of-the-art OOD generalization performances.
- Notably, CIGA is the only method that consistently outperforms ERM in the industry-provided realistic OOD benchmark, i.e., DrugOOD, demonstrating its high potential to push forward the developments of AI-Assisted Drug Discovery, and enrich the AI tools for facilitating the fundamental practice of science.
- Moreover, the interpretable GNN architecture used in the current version of CIGA also brings an extra benefit, i.e., interpretability of the results. Hence we also provide interpretability visualization examples in Appendix G.5. From the results we find CIGA can discover interesting patterns, which may provide new insights to human experts in drug discovery.
- Furthermore, in the replies ([part i](https://openreview.net/forum?id=A6AFK_JwrIW&noteId=naQjdPVFgi), [part ii](https://openreview.net/forum?id=A6AFK_JwrIW&noteId=WGhKB9x2oxJ)) to Reviewer **QGqs**, we provide empirical evidences showing that the architecture and the objectives of CIGA require little additional computational overhead and tuning efforts. In the [reply](https://openreview.net/forum?id=A6AFK_JwrIW&noteId=C0Z0Hh7AGCQ) to Reviewer **tMnp**, we find CIGA remains the state-of-the-art OOD method and brings non-trivial improvements under the single training environment setting. These empirical results could serve as the strong evidence for the high potential of CIGA that can be applied to various application scenarios.

Besides, we also provide a link of our codes for reproducing the results in our paper: https://anonymous.4open.science/r/CIGA-6B5F/  .

We again thank all reviewers for their efforts and many helpful comments/suggestions.

---

### Meta-Review · Area_Chair_jyCP · 2022-08-26

**Recommendation:** Accept
**Confidence:** Certain

**Metareview:**



Graph NNs have proven to work considerably well in the in-distribution setting. However, they fail when test data come from a different distribution than test, as shown by previous work. This paper aligns with recent works, and aims to study how to obtain invariance to shifts described by the assumed causal model. The assumed causal model is reasonable, and the solution is novel.
There is consensus among the referees, as evidenced by the score of 6 from each of them, that these results could be of interest to Neurips.

**Award:**

No

---

### Decision · Program_Chairs · 2022-09-14

Accept